# Communication trade-offs for Local-SGD with large step size

**Kumar Kshitij PATEL**
MLO, EPFL, Lausanne, Switzerland
TTIC-Toyota Technological Institute Chicago
kkpatel@ttic.edu

**Aymeric DIEULEVEUT**
MLO, EPFL, Lausanne, Switzerland
CMAP, Ecole Polytechnique, Palaiseau, France
aymeric.dieuleveut@polytechnique.edu

## Abstract

Synchronous mini-batch SGD is state-of-the-art for large-scale distributed machine learning. However, in practice, its convergence is bottlenecked by slow communication rounds between worker nodes. A natural solution to reduce communication is to use the *"local-SGD"* model in which the workers train their model independently and synchronize every once in a while. This algorithm improves the computation-communication trade-off but its convergence is not understood very well. We propose a non-asymptotic error analysis, which enables comparison to *one-shot averaging* i.e., a single communication round among independent workers, and *mini-batch averaging* i.e., communicating at every step. We also provide adaptive lower bounds on the communication frequency for large step-sizes ($t^{-\alpha}$, $\alpha \in (1/2, 1)$) and show that *local-SGD* reduces communication by a factor of $O\left(\frac{\sqrt{T}}{P^{3/2}}\right)$, with $T$ the total number of gradients and $P$ machines.

## 1 Introduction

We consider the minimization of an objective function which is accessible through unbiased independent and identically distributed estimates of its gradients. This problem has received attention from various communities over the last fifty years in optimization, stochastic approximation, and machine learning [1–7]. The most widely used algorithms are stochastic gradient descent (SGD), a.k.a. Robbins-Monro algorithm [8], and some of its modifications based on averaging of the iterates [1, 2, 9]. For a convex differentiable function $F : \mathbb{R}^d \to \mathbb{R}$, SGD iteratively updates an estimator $(\boldsymbol{v}_t)_{t \geq 0}$ for any $t \geq 1$

$$\boldsymbol{v}_t = \boldsymbol{v}_{t-1} - \eta_t \boldsymbol{g}_t(\boldsymbol{v}_{t-1}), \tag{1}$$

where $(\eta_t)_{t \geq 0}$ is a deterministic sequence of positive scalars, referred to as the *learning rate* and $\boldsymbol{g}_t(\boldsymbol{v}_{t-1})$ is an oracle on the gradient of the function $F$ at $\boldsymbol{v}_{t-1}$. We focus on objective functions that are both smooth and strongly convex [10]. While these assumptions might be restrictive in practice, they enable to provide a tight analysis of the error of SGD. In such a setting, two types of proofs have been used traditionally. On one hand, *Lyapunov*-type proofs rely on controlling the expected squared distance to the optimal point [11]. Such analysis suggests using *small* decaying steps, inversely proportional to the number of iterations ($t^{-1}$). On the other hand, studying the recursion as a stochastic process [1] enables to better capture the reduction of the noise through averaging. It results in optimal convergence rates for larger steps, typically scaling as $t^{-\alpha}$, $\alpha \in (1/2, 1)$ [10].

Over the past decade, the amount of available data has steadily increased: to adapt SGD to such situations, it has become necessary to *distribute* the workload between several machines, also referred to as *workers* [12–14]. For SGD, two extreme approaches have received attention: 1) workers run SGD independently and at the end aggregate their results, called *one-shot averaging* (**OSA**) [13, 15] or *parameter mixing*, and 2) *mini-batch averaging* (**MBA**) [16–20], where workers communicate after every iteration: all gradients are thus computed at the *same* support point (iterate) and the algorithm is equivalent to using mini-batches of size $P$, with $P$ the number of workers. While OSA

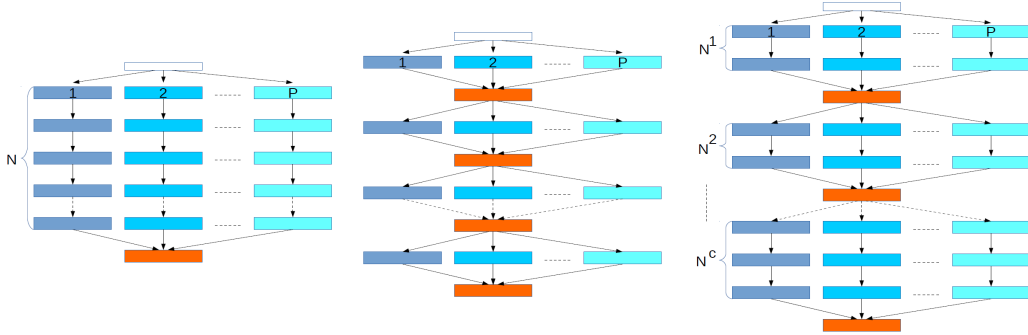

Figure 1: Schematic representation of one-shot averaging (left), mini-batch averaging (middle) and local-SGD (right). Vertical threads correspond to machines and orange boxes to communication rounds.

requires only a single communication step, it typically does not perform very well in practice [21]. At the other extreme, MBA performs better in practice, but the number of communications equals the number of steps, which is a major burden, as communication is highly time consuming [22]. To optimize this computation-communication-convergence trade-off, we consider the *local-SGD* framework: $P$ workers run SGD iterations in parallel and communicate periodically. This framework encompasses *one-shot averaging* and *mini-batch averaging* as special cases (see Figure 1). We make the following contributions:

1) We provide the first non-asymptotic analysis for local-SGD with large step sizes (typically scaling as $t^{-\alpha}$, for $\alpha \in (1/2; 1)$), in both on-line and finite horizon settings. Our assumptions encompass the ubiquitous *least-squares regression* and *logistic regression*.

2) Our comparison of the two extreme cases, OSA and MBA, underlines the communication trade-offs. While both of these algorithms are asymptotically equivalent for a fixed number of machines, mini-batch theoretically outperforms one-shot averaging when we consider the precise bias-variance split. In the regime where both the number of machines and gradients grow simultaneously we show that mini-batch SGD outperforms one-shot averaging.

3) Under three different sets of assumptions, we quantify the *frequency of communication* necessary for local-SGD to be optimal (i.e., as good as mini-batch). Precisely, we show that the communication frequency can be reduced by as much as $O\left(\frac{\sqrt{T}}{P^{3/2}}\right)$, with $T$ gradients and $P$ workers. Moreover, our bounds suggest an adaptive communication frequency for logistic regression, which depending on the expected distance to the optimal point (a phenomenon observed by Zhang et al. [21]).

4) We support our analysis by experiments illustrating the behavior of the algorithms.

The paper is organized as follows: in Section 2.1, we introduce the general setting, notations and algorithms, then in Section 2.2, we describe the related literature. Next, in Section 2.3, we describe assumptions made on the objective function. In Section 3, we provide our main results, their interpretation, consequence and comparison with other results. Results in the on-line setting and experiments are presented in the Appendix A.2 and Appendix B.

## 2 Algorithms and setting

We first introduce a couple of notations. We consider the finite dimensional Euclidean space $\mathbb{R}^d$ embedded with its canonical inner product $\langle \cdot, \cdot \rangle$. For any integer $\ell \in \mathbb{N}^*$, we denote by $[\ell]$ the set $\{1, \ldots, \ell\}$. We consider a strongly-convex differentiable function $F : \mathbb{R}^d \to \mathbb{R}$. We denote $\boldsymbol{w}^\star := \operatorname{argmin}_{\boldsymbol{w}} F(\boldsymbol{w})$. With only one machine, *Serial-SGD* performs a sequence of updates according to Equation (1). In the next section, we describe local-SGD, the object of this study.

### 2.1 Local-SGD algorithm

We consider $P$ machines, each of them running SGD. Periodically, workers aggregate (i.e., average) their models and restart from the resulting model. We denote by $C$ the number of communication steps. We define a *phase* as the time between two communication rounds. At *phase* $t \in [C]$, for any worker $p \in [P]$, we perform $N^t$ *local steps* of SGD. Iterations are thus naturally indexed by $(t, k) \in [C] \times [N^t]$. We consider the lexicographic order $\preccurlyeq$ on such pairs, which matches the order in which iterations are processed. Note that we assume the number of local steps to be the same over

all machines $p$. While this assumption can be relaxed in practice, is facilitates our proof technique and notation. At any $k \in [N^t]$, we denote by $\boldsymbol{w}_{p,k}^t$ the model proposed by worker $p$, at phase $t$, after $k$ local iterations. All machines initially start from the same point $\boldsymbol{w}_0$, that is for any $p \in [P]$, $\boldsymbol{w}_{p,0}^1 = \boldsymbol{w}_0$. The update rule is thus the following, for any $p \in [P], t \in [C], k \in [N^t]$:

$$\boldsymbol{w}_{p,k}^t = \boldsymbol{w}_{p,k-1}^t - \eta_k^t g_{p,k}^t(\boldsymbol{w}_{p,k-1}^t). \tag{2}$$

Aggregation steps consist in averaging the final local iterates of a phase: for any $t \in [C]$, $\hat{\boldsymbol{w}}^t = \frac{1}{P}\sum_{p=1}^P \boldsymbol{w}_{p,N^t}^t$. At phase $t+1$, every worker $p \in [P]$ restarts from the averaged model: $\boldsymbol{w}_{p,0}^{t+1} := \hat{\boldsymbol{w}}^t$. Eventually, we want to control the excess risk of the Polyak-Ruppert (PR) averaged iterate:

$$\overline{\overline{\boldsymbol{w}}}^C = \frac{1}{\sum_{t=1}^C N^t} \sum_{t=1}^C N^t \overline{\boldsymbol{w}}^t = \frac{1}{P\sum_{t=1}^C N^t} \sum_{t=1}^C \sum_{p=1}^P \sum_{k=1}^{N^t} \boldsymbol{w}_{p,k}^t,$$

with $\overline{\boldsymbol{w}}^t = \frac{1}{PN^t}\sum_{k=1}^{N^t}\sum_{p=1}^P \boldsymbol{w}_{p,k}^t$. We use the notation $\overline{\boldsymbol{w}}$ to underline the fact that iterates are averaged over one phase and $\overline{\overline{\boldsymbol{w}}}$ when averaging is made over all iterations. All averaged iterates can be computed on-line.

The algorithm, called *local-SGD*, is thus parameterized by the number of machines $P$, communication steps $C$, local iterations $(N^t)_{t\in[C]}$, the starting point $\boldsymbol{w}_0$, the learning rate $(\eta_k^t)_{(t,k)\in[C]\times[N^t]}$, and the first order oracle on the gradient. Pseudo-code of the algorithm is given in the Appendix, in Fig. S5.

**Link with classical algorithms.** Special cases of local-SGD correspond to *one-shot averaging* or *mini-batch averaging*. More precisely, for a total number of gradients $T$, with $P$ workers, $C = T/P$ communication rounds, and $(N^t)_{t\in[C]} = (1, \ldots, 1)$, we realize an instance of P-mini-batch averaging (P-MBA). On the other hand, with $P$ workers, $C = 1$ communication, and $(N^1) = T/P$, we realize an instance of one shot-averaging. Our goal is to get general convergence bounds for local-SGD that recover classical bounds for both these settings when we choose the correct parameters. While comparing to Serial-SGD (which is also a particular case of the algorithm), would also be interesting, we focus here on the comparison between local-SGD, *one-shot averaging* and *mini-batch averaging*. Indeed, the step size is generally increased for mini-batch with respect to Serial SGD, and the running efficiency of algorithms is harder to compare: we only focus on different algorithms that use the *same number of machines*.

## 2.2 Related Work

**On Stochastic Gradient Descent.** Bounds on the excess risk of SGD for convex functions have been widely studied: most proofs rely on controlling the decay of the mean squared distance $\mathbb{E}[\|\boldsymbol{v}_t - \boldsymbol{w}^\star\|^2]$, which results in an upper bound on the mean excess of risk $\mathbb{E}[F(\bar{\boldsymbol{v}}_t) - F(\boldsymbol{w}^\star)]$ [23, 24]. This upper bound is composed of a "bias" term that depends on the initial condition, and a "variance" term that involves either an upper bound on the *norm* of the noisy gradient (in the non-smooth case), or an upper bound on the *variance* of the noisy gradient in the smooth case [5, 11]. In the strongly convex case such an approach advocates for the use of *small* step sizes, scaling as $(\mu t)^{-1}$. However, in practice, this is not a very satisfying result, as the constant $\mu$ is typically unknown, and convergence is very sensitive to ill-conditioning. On the other hand, in the smooth and strongly-convex case, the classical analysis by Polyak and Juditsky [1], relies on an explicit decomposition of the stochastic process $(\bar{\boldsymbol{v}}_t - \boldsymbol{w}^\star)_{t\geq 1}$: the effect of averaging on the noise term is better taken into account, and this analysis thus suggests to use larger steps, and results in the optimal rate for $\eta_t \propto t^{-\alpha}$, with $\alpha \in (0; 1)$. This type of analysis has been successfully used recently [10, 15, 25, 26].

For quadratic functions, larger steps can be used, as pointed by Bach and Moulines [27]. Indeed, even with *non-decaying* step size, the averaged process converges to the optimal point. Several studies focus on understanding properties of SGD for quadratic functions: a detailed non-asymptotic analysis is provided by Défossez and Bach [28], acceleration under the additive noise oracle (see Assumption A4 below) is studied by Dieuleveut et al. [29] (without this assumption by Jain et al. [30]), and Jain et al. [20] analyze the effects of mini-batch and tail averaging.

**One shot averaging.** In this approach, the $P$-independent workers compute several steps of stochastic gradient descent, and a unique communication step is used to average the different models [13, 31, 32]. Zinkevich et al. [13] show a reduction of the variance when multiple workers are used, but neither consider the Polyak-Ruppert averaged iterate as the final output, nor provide non-asymptotic rates.

Zhang et al. [33] provide the first non-asymptotic results for OSA but their dependence on constants (like strong convexity constant $\mu$, moment bounds, etc.) is worse; as well as their single machine convergence bound [34] is not truly non-asymptotic (like for e.g., Bach and Moulines [10]). More importantly, their results hold only for small learning rates like $\frac{c}{\mu t}$. Rosenblatt and Nadler [35] have also discussed the asymptotic equivalence of OSA with vanilla-SGD by providing an analysis up to the second order terms. Further, Jain et al. [20] have provided non-asymptotic results for least-square regression using similar Polyak-Juditsky analysis of the stochastic process, while our results apply to more general problems. Their approach encompasses one shot averaging and the effect of tail averaging, that we do not consider here. Recently, Godichon and Saadane [15] proposed an approach similar to ours (but only for one shot averaging). However, their result relies on an asymptotic bound, namely $\mathbb{E}[\|\boldsymbol{w}_t - \boldsymbol{w}^\star\|^2] \leq C_1 \eta_t$ (as in Rakhlin et al. [34]), while our analysis is purely non-asymptotic and we also improve the upper bound on the noise term which results from the analysis.

**Mini-batch averaging.** Mini-batch averaging has been studied by Dekel et al. [16], Takáč et al. [17]. These papers show an improvement in the variance of the process, and make comparisons to SGD. It has been found that increasing the mini-batch size often leads to increasing generalization errors, which limits their distributivity [36]. Jain et al. [20] have provided upper bounds on learning-rate and mini-batch size for optimal performance. Recently, large mini-batches have been leveraged successfully in deep learning as in [37–39] by properly tuning learning rates, etc.

**Local-SGD.** Zhang et al. [21] empirically show that local-SGD performs well. They also provide a theoretical guarantee on the variance of the process, however, they assume the variance of the estimated gradients to be uniformly upper bounded (Assumption A4 below). Such an assumption is restrictive in practice, for example it is not satisfied for least squares regression. In a simultaneous work, Stich [40] has provided an analysis for local-SGD. The limitation with their analysis is that they also assume bounded gradients and use a small step size scaling as $\frac{c}{\mu t}$. More importantly, their analysis doesn't extend to the extreme case of one-shot averaging like ours. Lin et al. [41] have experimentally shown that local-SGD is better than the synchronous mini-batch techniques, in terms of overcoming the large communication bottleneck. Recently, Yu et al. [42] have given convergence rates for the non-convex synchronous and a stale synchronous settings.

We have summarized the major limitations of some of these analyses in Table S3, given in Appendix I. Our motivation is to get away with some of these restrictive assumptions, and provide tight upper bounds for the above three averaging schemes. In the following section, we present the set of assumptions under which our analysis is conducted.

### 2.3 Assumptions

We first make the following classical assumptions on the objective function $F : \mathbb{R}^d \to \mathbb{R}$. In the following, we use different subsets of these assumptions:

**A1 (Strong convexity)** *The function $F$ is strongly-convex with convexity constant $\mu > 0$.*

**A2 (Smoothness and regularity)** *The function $F$ is three times continuously differentiable with second and third uniformly bounded derivatives:* $\sup_{\boldsymbol{w} \in \mathbb{R}^d} \left\| \left| F^{(2)}(\boldsymbol{w}) \right| \right\| < L$, *and* $\sup_{\boldsymbol{w} \in \mathbb{R}^d} \left\| \left| F^{(3)}(\boldsymbol{w}) \right| \right\| < M$. *Especially $F$ is $L$-smooth.*

**Q1 (Quadratic function)** *There exists a positive definite matrix $\Sigma \in \mathbb{R}^{d \times d}$, such that the function $F$ is the quadratic function $\boldsymbol{w} \mapsto \|\Sigma^{1/2}(\boldsymbol{w} - \boldsymbol{w}^\star)\|^2 / 2$.*

If **Q1** is satisfied, then Assumptions **A1**, **A2** are satisfied, and $L$ and $\mu$ are respectively the largest and smallest eigenvalues of $\Sigma$. At any iteration $(t, k) \in [C] \times [N^t]$, any machine can query an unbiased estimator of the gradient $g_{p,k}^t(\boldsymbol{w})$ at a point $\boldsymbol{w}$. Formally, we make the following assumption:

**A3 (Oracle on the gradient)** *We observe unbiased estimators of the gradient $g_{p,k+1}^t(\boldsymbol{w})$: for any $(t, k) \in [C] \times [N^t]$ and $\boldsymbol{w} \in \mathbb{R}^d$, $\mathbb{E}[g_{p,k+1}^t(\boldsymbol{w}_{p,k}^t)|\boldsymbol{w}_{p,k}^t] = F'(\boldsymbol{w}_{p,k}^t)$. Moreover, for any fixed $\boldsymbol{w}$ the functions $(g_{p,k}^t)_{(t,k)}(\boldsymbol{w})$ are i.i.d. . (See Appendix A.1 for a more formal statement.)*

In Proposition 3, we make the additional, stronger assumption that the variance of gradient estimates is uniformly upper bounded, a standard assumption in the SGD literature, see e.g. Zhang et al. [21]:

**A4 (Uniformly bounded variance)** *The variance of the error, $\mathbb{E}[\|g_{p,k}^t(\boldsymbol{w}_{p,k}^t) - F'(\boldsymbol{w}_{p,k}^t)\|^2]$ is uniformly upper bounded by $\sigma_\infty^2$, a constant which does not depend on the iteration.*

Assumption **A4** is for example true if the sequence of random vectors $(g_{p,k+1}^t(\boldsymbol{w}_{p,k}^t) - F'(\boldsymbol{w}_{p,k}^t))_{t \in [C], k \in [N^t], p \in [P]}$ is i.i.d.. This setting is referred to as the semi-stochastic setting [29].

We also consider the following conditions on the regularity of the gradients, for $p \geq 2$:

**A5 (Cocoercivity of the random gradients)** *For any $p \in [P]$, $t \in [C]$, $k \in [N^t]$, $g_{p,k}^t$ is almost surely $L$-co-coercive (with the same constant as in A2): that is, for any $w_1, w_2 \in \mathbb{R}^d$, $L\langle g_{p,k}^t(w_1) - g_{p,k}^t(w_2)w_1 - w_2\rangle \geq \|g_{p,k}^t(w_1) - g_{p,k}^t(w_2)\|^2$.*

Almost sure $L$-co-coercivity [43] is for example satisfied if for any $(p, k, t)$, there exist a random function $f_{p,k}^t$ which is a.s. convex and $L$-smooth and such that $g_{p,k}^t = (f_{p,k}^t)'$. Finally, we assume the fourth order moment of the random gradients at $w^\star$ to be well defined:

**A6 (Finite variance at $w^\star$)** $\exists \sigma \geq 0$, *s.t. for any $t, k, p \in [C] \times [N^t] \times [P]$, $\mathbb{E}[\|g_{p,k}^t(w^\star)\|^4] \leq \sigma^4$.*

It must be noted that **A**6 is a much weaker assumption than **A**4, for e.g., least-square regression satisfies former but not latter. Most of these assumptions are classical in machine learning. SGD for least squares regression satisfies **Q**1, **A**3, **A**5 and **A**6. On the other hand, SGD for logistic regression satisfies **A**1, **A**2, **A**3 and **A**4. Our main result Theorem 6 (lower bounding the frequency of communications) applies to both these sets of assumptions. In Appendix C.3 we further detail how these assumptions apply in machine learning.

**Learning rate.** We always assume that for any $t \in [C], k \in [N^t]$, the learning rate satisfies $2\eta_k^t L \leq 1$. We consider two different types of learning rates:
1) in the *finite horizon* (FH) case, the step size $(\eta_k^t)_{(t,k)\in[C]\times[N^t]}$ is a constant $\eta$, that can depend on the number of iterations eventually performed by the algorithm; 2) in the *on-line* case, the sequence of step size is a subsequence of a universal sequence $(\tilde{\eta}_\ell)_{\ell \geq 0}$. Moreover, in our analysis, when using decaying learning rate, the step size only depends on the number of iterations processed in the past: $\eta_k^t = \tilde{\eta}_{\{\sum_{t'=1}^{t-1} N^{t'} + k\}}$. Especially, the step size at iteration $(t, k)$ does not depend on the machine.

Though both of these approaches are often considered to be nearly equivalent [44, 45], fundamental differences exist in their convergence properties. The *on-line* case is harder to analyze, but ultimately provides a better convergence rate. However as the behavior is easier to interpret in the finite horizon case, we postpone results for on-line setting to Appendix A.2. In the following section, we present our main results.

# 3 Main Results

**Sketch of the proof.** We follow the approach by Polyak and Juditsky, which relies on the following decomposition: for any $p \in [P], t \in [C], k \in [N^t]$, Equation (2) is trivially equivalent to: $\eta_k^t F''(w^\star)(w_{p,k-1}^t - w^\star) = w_{p,k-1}^t - w_{p,k}^t - \eta_k^t[g_{p,k}^t(w_{p,k-1}^t) - F'(w_{p,k-1}^t)] - \eta_k^t[F'(w_{p,k-1}^t) - F''(w^\star)(w_{p,k-1}^t - w^\star)]$. We have added and subtracted a first order Taylor expansion around the optimal value $w^\star$ of the gradient. Thus, using the definition of $\overline{\overline{w}}^C$:

$$F''(w^\star)\left(\overline{\overline{w}}^C - w^\star\right) = \frac{1}{P\sum_{t=1}^C N^t} \sum_{t=1}^C \sum_{p=1}^P \sum_{k=1}^{N^t} \left(\frac{w_{p,k-1}^t - w_{p,k}^t}{\eta_k^t} - \left[g_{p,k}^t(w_{p,k-1}^t) - F'(w_{p,k-1}^t)\right]\right.$$
$$\left. - \left[F'(w_{p,k-1}^t) - F''(w^\star)(w_{p,k-1}^t - w^\star)\right]\right). \tag{3}$$

In other words, the error can be decomposed into three terms: the first one mainly depends on the *initial condition*, the second one is a *noise term*: it is the mean of centered random variables (as $\mathbb{E}[g_{p,k}^t(w_{p,k-1}^t) - F'(w_{p,k-1}^t)] = 0$), and the third is a *residual term* that accounts for the fact that the function is not quadratic (if $F$ is quadratic, then $F'(w_{p,k-1}^t) - F''(w^\star)(w_{p,k-1}^t - w^\star) = 0$).

**Controlling different terms in Equation** (3). The variance of the noise $g_{p,k}^t(w_{p,k-1}^t) - F'(w_{p,k-1}^t)$ and the residual term both directly depend on the distance $\|w_{p,k-1}^t - w^\star\|^2$. The proof is thus composed of two aspects: (1) we first provide a tight control for this quantity, with or without communication: in the following propositions, this corresponds to an upper bound on $\mathbb{E}[\|w_{p,k}^t - w^\star\|^2]$ [1], (2) we provide the subsequent upper bound on $\mathbb{E}[\|F''(w^\star)(\overline{\overline{w}}^C - w^\star)\|^2]$.

We first compare the convergence in the two extreme situations, *i.e.*, for *Mini-batch averaging* (MBA) and *One-shot averaging* (OSA) for *finite horizon* setting, and then provide these results for local-SGD.

## 3.1 Results for MBA and OSA, Finite Horizon setting

First we assume the step size $\eta_k^t$ to be a constant $\eta$ at every iteration for any $t \in [C], k \in [N^t]$. Our first contribution is to provide *non-asymptotic* convergence rates for *MBA* and *OSA*, that allow a simple comparison. For the benefit of presentation, we define following quantities: $Q_{bias} = 1 + \frac{M^2\eta}{\mu}\|w^0 - w^\star\|^2 + \frac{L^2\eta}{\mu P}$, $Q_{1,var}(X) = \frac{L^2\eta}{\mu} + \frac{P}{X\eta\mu}$, $Q_{2,var}(X) = \frac{M^2XP\eta^2\sigma^2}{\mu^2}$.

In the following, we use the $\precsim$ notation to denote inequality up to an absolute constant. Recall that for MBA, the total number of gradients processed is $T = PC$, while it is $T = PN$ for OSA. We have the following results respectively for MBA and OSA:

**Proposition 1 (Mini-batch Averaging)** *Under Assumptions A1, A2, A3, A5, A6, we have the following bound for mini-batch SGD: for any $t \in [C]$,*

$$\mathbb{E}\left[\left\|\hat{w}^t - w^\star\right\|^2\right] \leq (1 - \eta\mu)^t \|w_0 - w^\star\|^2 + \frac{2\sigma^2\eta}{P}\frac{1 - (1 - \eta\mu)^t}{\mu}, \tag{4}$$

$$\mathbb{E}\left[\left\|F''(w^\star)(\overline{\overline{w}}^C - w^\star)\right\|^2\right] \precsim \frac{\|w^0 - w^\star\|^2}{\eta^2 C^2}Q_{bias} + \frac{\sigma^2}{T}\left(1 + \frac{Q_{1,var}(C)}{P} + \frac{Q_{2,var}(C)}{P^2}\right). \tag{5}$$

**Proposition 2 (One-shot Averaging)** *Under Assumptions A1, A2, A3, A5, A6, we have the following bound for one shot averaging: $p \in [P], t = 1, k \in [N]$,*

$$\mathbb{E}\left[\left\|w_{p,k}^1 - w^\star\right\|^2\right] \leq (1 - \eta\mu)^k \|w_0 - w^\star\|^2 + 2\sigma^2\eta\frac{1 - (1 - \eta\mu)^k}{\mu}, \tag{6}$$

$$\mathbb{E}\left[\left\|F''(w^\star)(\overline{\overline{w}}^C - w^\star)\right\|^2\right] \precsim \frac{\|w^0 - w^\star\|^2}{\eta^2 N^2}Q_{bias} + \frac{\sigma^2}{T}\left(1 + Q_{1,var}(N) + Q_{2,var}(N)\right). \tag{7}$$

**Interpretation, fixed $P$.** Using mini-batch naturally reduces the variance of the process $(w_{p,k}^t)_{p \in [P], t \in [C], k \in [N^t]}$. Equations (4) and (6) show that the speed at which the initial condition is forgotten remains the same, but that the variance of the local process is reduced by a factor $P$.

Equations (5) and (7) show that the convergence depends on an *initial condition* term and a *variance term*. For a fixed number of machines $P$, and a step size scaling as $\eta = X^{-\alpha}$, $0.5 < \alpha < 1$, $X \in \{N, C\}$, the speed at which the *initial condition* is forgotten is asymptotically dictated by $Q_{bias}/(\eta X)^2$ where $X \in \{N, C\}$, for *both algorithms* (if we use the same number of gradients for both algorithms, naturally, $N = C$.) As for the variance term, it scales as $\sigma^2 T^{-1}$ as $T \to \infty$, as the remaining terms $Q_{var}(X)$ asymptotically vanish for $\eta = X^{-\alpha}$. It reduces with the total number $T$ of gradients used in the process. Interestingly, this term is *the same* for the two extreme cases (MBA and OSA): it does not depend on the number of communication rounds. This phenomenon is often described as *"the noise is the noise and SGD doesn't care"* (for asynchronous SGD, [46]). Though we recover this asymptotic equivalence here, our belief is that this asymptotic point of view is typically misleading as the asymptotic regime is not always reached, and the residual terms do then matter.

Indeed, the lower order terms do have a dependence on the number of communication rounds: when the number of communications increases, the overall effect of the noise is reduced. More precisely, since $Q_{var}(N) = Q_{var}(C)$ the remaining terms are respectively $P$ or $P^2$ times smaller for mini-batch. This provides a theoretical explanation of why mini-batch SGD outperforms one shot averaging in practice. It also highlights the weakness of an asymptotic analysis: the dominant term might be equivalent, without reflecting the actual behavior of the algorithm. Disregarding communication aspects, mini-batch SGD is in that sense *optimal*.

Note that for quadratic functions, $Q_{2,var} = 0$ as $M = 0$. The conditions on the step size can thus be relaxed, and the asymptotic rates described above would be valid for any step size satisfying $\eta \leq \mu$ [20]. Extension to the on-line setting, eventually leading to a better convergence rate, is given in Proposition S7 in AppendixA.2.

**Interpretation, $P, T \to \infty$.** When both the total number of gradients used $T$ and the number of machines $P$ are allowed to grow simultaneously, the asymptotic regime is not necessarily the same for MBA and OSA, as remaining terms are not always negligible. For example, if fixing $\eta = X^{-2/3}$, $X \in \{N, C\}$ (we chose $\alpha = 2/3$ to balance $Q_{1,var}$ and $Q_{2,var}$), the variance term

would be controlled by $\sigma^2 T^{-1}(1 + \frac{P}{\mu C^{1/3}})$. Thus, unless $P \leq \mu C^{1/3}$, MBA could outperform OSA by a factor as large as $P$.

**Novelty and proofs.** Both Propositions 1 and 2 are proved in the Appendix G. Importantly, Equations (4) and (6) respectively imply Equations (5) and (7) under the stated conditions: this is the reason why we only focus on proving equations similar to Equations (4) and (6) for local-SGD.

Proposition 1 is similar to the analysis of *Serial-SGD* for large step size, but with a reduction in the variance proportional to the number of machines. Such a result is derived from the analysis by Dieuleveut et al. [25], combining the approach of Bach and Moulines [27] with the correct upper bound for smooth strongly convex SGD [47], and controlling similarly higher order moments. While this result is expected, we have not found it under such a simple form in the literature. Proposition 2 follows a similar approach, we combine the proof for mini-batch with a control of the iterates of each of the machines. This is closely related to Godichon and Saadane [15], but we preserve a non-asymptotic approach.

**Remark: link with convergence in function values.** As we use Equation (3) as a starting point, we provide convergence results on the Mahalanobis distance $\|F''(\boldsymbol{w}^\star)(\overline{\overline{\boldsymbol{w}}}^C - \boldsymbol{w}^\star)\|^2$: it is the natural quantity in such a setting [10, 15, 27]. These results could be translated into function value convergence $F(\overline{\overline{\boldsymbol{w}}}^C) - F(\boldsymbol{w}^\star)$, using the inequality $F(\overline{\overline{\boldsymbol{w}}}^C) - F(\boldsymbol{w}^\star) \leq L\mu^{-2}\|F''(\boldsymbol{w}^\star)(\overline{\overline{\boldsymbol{w}}}^C - \boldsymbol{w}^\star)\|^2$ but the dependence on $\mu$ would be pessimistic and sub-optimal. However, a similar approach has been used by Bach [44], under a slightly different set of assumptions (including self-concordance, e.g., for logistic regression), recovering optimal rates. Extension to such a set of assumptions, which relies on tracking other quantities, is an important direction.

While the "classical proof", which provides rates for function values directly (with smoothness, or with uniformly bounded gradients) has a better dependence on $\mu$, one cannot easily obtain a noise reduction when averaging between machines. Similarly, there is no proof showing that one-shot averaging is asymptotically optimal that relies only on function values. In other words, these proofs do not adequately capture the noise reduction due to averaging. Moreover, such proof techniques relying on function values typically involve a small step size $1/(\mu t)$ (because the noise reduction is captured inefficiently). Such step size performs poorly in practice (initial condition is forgotten slowly), and $\mu$ is unknown.

In conclusion, though they do not directly result in optimal dependence on $\mu$ for function values, we believe our approach allows to correctly capture the effect of the noise, and is thus suitable for capturing the effect of local-SGD.

**Comparing upper bounds:** Our analysis relies on upper bounds: one should handle comparison with cautions. Nevertheless, we think our analysis is tight enough to provide good insights, especially because the bound for OSA averaging nearly matches the bound for MBA (contrary to Stich [40]). Moreover, the bounds given above are tight in the following senses, see Appendix A.3 for details:
*(i)* the bias term in equations (5) and (7) is clearly *exact* in the simple case of a quadratic one dimensional function, in the absence of noise: it is normal that in such a situation, MBA and OSA converge similarly: each of the $P$ independent machines computes the same recursion!
*(ii)* the bound for the variance, scaling as $(PN)^{-1}$ for any $\eta \propto N^{-\alpha}, 0.5 < \alpha < 1$, matches the statistical minimax rate [48] for least squares regression: from the statistical point of view, if we are only given $NP$ independent observations, then no estimator can have an error uniformly lower than $\sigma^2(PN)^{-1}$.

Optimizing over the step size in Eqs (5) and (7) results in a somehow disappointing observation: the rate for $\eta \propto N^{-\alpha}, 0.5 < \alpha < 1$[2] is dictated by the bias and scales as $O((\eta N)^{-2})$, which is slow (but tight, see point (i) above). This is unfortunately unavoidable *with constant step sizes*: the convergence rate with decaying steps is much faster in the on-line setting[3], but bounds are much harder to read see Sec. A.2. In other words, bounds in Propositions 1 and 2 are *tight*, but *slower* than in on-line setting. As *all the trade-offs regarding communications are preserved* (our main focus), we chose to highlight the results in finite horizon in the main text.

**Conclusion:** for a fixed or limited number of machines, asymptotically, the convergence rate is similar for OSA and MBA. However, non-asymptotically, or when the number of machines also increases, the dominant terms can be as much as $P^2$ times smaller for MBA. In the following we provide conditions for local-SGD to perform as well as MBA (while requiring much fewer communication rounds).

## 3.2 Convergence of Local-SGD, Finite Horizon setting

For local-SGD we first consider the case of a quadratic function, under the assumption that the noise has a uniformly upper bounded variance. While this set of assumptions is not realistic, it allows an intuitive presentation of the results. Similar results for settings encompassing LSR and LR follow. We provide a bound on the moment of an iterate after the communication step $\hat{\boldsymbol{w}}^t$ (i.e., the restart point of the next phase), and on the second order moment of any iterate. For $t \in [C]$, we denote $\boldsymbol{N}_1^t := \sum_{t'=1}^{t} N^{t'}$.

**Proposition 3 (Local-SGD: Quadratic Functions with Bounded Noise)** *Under Assumptions **Q** 1, A3, A4, we have the following bound for local-SGD: for any $p \in [P], t \in [C], k \in [N^t]$,*

$$\mathbb{E}\left[\left\|\hat{\boldsymbol{w}}^{t-1} - \boldsymbol{w}^{\star}\right\|^2\right] \leq (1 - \eta\mu)^{\boldsymbol{N}_1^{t-1}} \left\|\boldsymbol{w}_0 - \boldsymbol{w}^{\star}\right\|^2 + \frac{\sigma_\infty^2 \eta}{P} \frac{1 - (1 - \eta\mu)^{\boldsymbol{N}_1^{t-1}}}{\mu}$$

$$\mathbb{E}\left[\left\|\boldsymbol{w}_{p,k}^t - \boldsymbol{w}^{\star}\right\|^2\right] \leq (1 - \eta\mu)^{\boldsymbol{N}_1^{t-1}+k} \left\|\boldsymbol{w}_0 - \boldsymbol{w}^{\star}\right\|^2 + \sigma_\infty^2 \eta \left(\underbrace{\frac{1 - (1 - \eta\mu)^{\boldsymbol{N}_1^{t-1}}}{P\mu}}_{\text{long term reduced variance}} + \underbrace{\frac{1 - (1 - \eta\mu)^{k}}{\mu}}_{\text{local iteration variance}}\right).$$

To prove such a result, we use the classical technique, and introduce a *ghost* sequence $\breve{\boldsymbol{w}}_k^t := \frac{1}{P}\sum_{p=1}^{P} \boldsymbol{w}_{p,k}^t$, and recursively control $\left\|\breve{\boldsymbol{w}}_k^t - \boldsymbol{w}^{\star}\right\|^2$. We conclude by remarking that $\breve{\boldsymbol{w}}_{N^t}^t = \hat{\boldsymbol{w}}^t$. This proof is given in Appendix D.2.

**Interpretation.** The variance bound for the iterates "just after" communication, $\hat{w}^t$ exactly behaves as in mini-batch case: the initialization term decays linearly with the number of local steps, and the variance is reduced proportionally to the number of workers $P$. On the other hand, the bound on the iterates $\boldsymbol{w}_{p,k}^t$ shows that the variance of this process is composed of a "long term" reduced variance, that accumulates through phases, and is increasingly converging to $\frac{\sigma_\infty^2 \eta}{P\mu}$ and of an extra variance $\eta\sigma_\infty^2 \frac{1-(1-\eta\mu)^k}{\mu}$, that increases within the phase, and is upper bounded by $\sigma_\infty^2 \eta^2 k$.

In the case of constant step size, the iterates of serial SGD converge to a limit distribution $\pi_\eta$ that depends on the step size [25]. Here, the iterates after communication (or the mini-batch iterates) converge to a distribution with reduced variance $\pi_{\eta/P}$, thus local iterates periodically restart from a distribution with reduced variance, then slowly "diverge" to the distribution with large variance. If the number of local iterations is small enough, the iterates keep a reduced variance. More precisely, we have the following result.

**Corollary 4** *If for all $t \in [C]$, $N^t \leq (\mu\eta P)^{-1}$, then the second order moment of $\boldsymbol{w}_{p,k}^t$ admits the same upper bound as the mini-batch iterate $\hat{\boldsymbol{w}}_{MB}^{\boldsymbol{N}_1^{t-1}+k}$ (Equation (4)) up to a constant factor of 2. As a consequence, Equation (5) is still valid, and local-SGD performs "optimally".*

**Interpretation.** This result shows that if the algorithm communicates often enough, the convergence of the Polyak Ruppert iterate $\overline{\overline{\boldsymbol{w}}}^C$ is as good as in the mini-batch case, thus it is "optimal". Moreover, the minimal number of communication rounds is easy to define: the maximal number of local steps $N^t$ decays as the number of workers and the step size increases. This bound implies that more communication steps are necessary when more machines are used. Note that $(\eta P)^{-1}$ is a large number, as a typical value for $\eta$ is inversely proportional to (a power of) the number of local steps for e.g., $(\sum_{t'=1}^{t} N^{t'})^{-\alpha}$, $\alpha \in (1/2; 1)$.

**Example 5** *With constant number of local steps $N^t = N$, and learning rate $\eta = c(NC)^{-1/2}$ in order to obtain an optimal $O(\sigma^2 T^{-1})$ parallel variance[4] rate, local-SGD communicates $O(\sqrt{NC}/(P\mu))$ times less as compared to mini-batch averaging.*

We believe that this is the first result (with Stich [40]) that shows a communication reduction proportional to a power of the number of local steps of a local solver (i.e., $O(\sqrt{NC})$), compared to mini-batch averaging. In the following, we alternatively relax the bounded variance assumption **A**4 and the quadratic assumption **Q**1, and show similar results for local-SGD. This allows us to successively cover the cases of least squares regression (LSR) and logistic regression (LR).

**Theorem 6** *Under either of the following sets of assumptions, the convergence of the Polyak Ruppert iterate $\overline{\overline{w}}^C$ is as good as in the mini-batch case, up to a constant:*
*(i) Assume **Q**1, **A**3, **A**5, **A**6, and for any $t \in [C]$, $N^t \leq (\mu\eta P)^{-1}$ and $\mu\eta^2 \mathbf{N}_1^t = O(1)$.*
*(ii) Assume **A**1, **A**2, **A**3, **A**4, and for any $t \in [C]$, $N^t \leq \inf\left((\eta PM\mathbb{E}[\|\hat{w}^t - w^\star\|])^{-1}, (\mu\eta P)^{-1}\right)$.*

These results are derived from Proposition S16 and Proposition S20 which generalize Proposition 3. Those results are proved in Appendix D and E and constitute the main technical challenge of the paper.

**Interpretation.** We note that in both of these situations, the optimal rates can be achieved if the communications happen often enough, and beyond such a number of communication rounds, there is no substantial improvement in the convergence. This result corresponds to the effect observed in practice [21]. The first set of assumption is valid for LSR, the second for LR. In the first case, the maximal number of local steps before communication is upper bounded by the same ratio as in Corollary 4, but the "constant" that appears is $\exp(\mu\eta^2\mathbf{N}_1^t)$, so we need this quantity to be small (which is typically always satisfied in practice) in order to be optimal w.r.t. mini-batch averaging. A similar result as Theorem 5 can be provided reducing the communication by a factor of $O(\frac{\sqrt{NC}}{P\mu})$.

In the second case, the maximal number of local steps is smaller than before, by a factor $\mu^{-1}$, but the allowed maximal number of local steps can increase along with the epochs, as $\mathbb{E}[\|\hat{w}^t - w^\star\|]$ is typically decaying. This adaptive communication frequency has been observed to work well in practice [21] and also explored in [49], in a setting without PR averaging. Assuming optimization on a compact space with radius $R$ for instance, one can obtain a $O(\frac{\sqrt{NC}}{P^2})$ times improvement in communication, similar to Theorem 5.

Though they may reflect the actual behavior of the algorithm, such results might be difficult to use directly in practice, as $\mu$ is unknown. However, as it is not the limiting factor in Theorem 6.2, an estimation of $\mathbb{E}[\|\hat{w}^t - w^\star\|]$ could allow us to use adaptive phases lengths to minimize communications.

## 4 Conclusion

Stochastic approximation and distributed optimization are both very densely studied research areas. However, in practice most distributed applications stick to bulk synchronous mini-batch SGD. While the algorithm has desirable convergence properties, it suffers from a huge communication bottleneck. In this paper we have analyzed a natural generalization of mini-batch averaging, local-SGD. Our analysis is non-asymptotic, which helps us to better understand the exact communication trade-offs. We give feasible lower bounds on communication frequency which significantly reduce the need for communication, while providing similar non-asymptotic convergence as mini-batch averaging. Our results apply to common loss functions, and use large step sizes. Further, our analysis unifies and extends all the scattered results for one-shot averaging, mini-batch averaging and local-SGD, providing an intuitive understanding of their behavior.

While they provide some intuition and are believed to be tight, our comparisons are based on upper bounds. Proving corresponding lower bounds is an interesting and important open direction. Also, it would also be interesting to study observable quantities to predict an adaptive communication frequency and to relax some of the technical assumptions required by the analysis. The on-line case, experiments, proofs, additional materials and a review of distributed optimization follow in the appendix.

**Acknowledgements**

We would like to acknowledge Sai Praneeth Reddy, Sebastian Stich, Martin Jaggi and Nathan Srebro for helpful comments and discussions at various stages of this project.

## Footnotes

[1] more precisely, on $\mathbb{E}[\|\hat{w}^t - w^\star\|^2]$ and $\mathbb{E}[\|w_{p,k}^1 - w^\star\|^2]$ for MBA and OSA respectively.

[2]A good step size is unlikely to be larger than $1/\sqrt{N}$: such "very large" LR (which is rarely used in practice) does not perform well for non-quadratic functions (note that for quadratic, the $NP\eta^2$ vanishes, and a constant $\eta$ would get a rate $1/N^2 + 1/PN$).

[3]the bias decreases as $1/N^2$ instead of $1/(\eta N)^2$ (see Prop.S7).

[4] in online setting, the same example would hold, resulting in a $O(\frac{\sigma^2}{T})$ convergence **rate** (not only variance).

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
