[Supplementary Material]

# Communication trade-offs for synchronized distributed SGD (Local-SGD) with large step size (with Appendix)

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

[5]Data available at https://www.csie.ntu.edu.tw/ cjlin/libsvmtools/datasets/.

[6]In the following, $\square, \diamond, \clubsuit$, etc. are used as symbolic notations to ease presentation.

[7]Note that after the final iteration of the phase the learning rate (which the algorithm uses nowhere) corresponds to the first learning rate for the next phase. This anomaly in notation is a direct result of us considering the ghost process, which runs continuously till the end.

[8]Note that we ignore t=1 in second inequality for second term as we have already incorporated it in the first term

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

# Communication trade-offs for synchronized distributed SGD with large step size

## SUPPLEMENTARY MATERIAL

In this Appendix, we give the proofs of our main results, and auxiliary elements. In Section A.2, we provide results in the on-line setting where we consider the particular case of a decaying sequence $\eta_k^t = (\sum_{t'=1}^{t-1} N^{t'} + k)^{-\alpha}$, for some $\alpha \in (\frac{1}{2}, 1)$. In Appendix B, we describe the experimental evaluations that illustrate the behavior of the different processes. In Appendix C we provide some additional material (Tables, interpretations, etc.) which may help the reader navigate through our results. In Appendix D, we prove contraction Lemmas for $\mathbb{E}[\|\boldsymbol{w}_{p,k}^t - \boldsymbol{w}^\star\|^2]$. In Appendix E, we prove similar guarantees for moment of order 4. In Appendix G, we give the proof of the main results on $\|F''(\boldsymbol{w}^\star)(\overline{\overline{\boldsymbol{w}}}^C - ws)\|^2$ for mini-batch, one-shot averaging, and Local-SGD in the Finite Horizon setting. In Appendix H we give similar results in the online setting (for decaying step size). Finally, we provide a brief survey of distributed optimization techniques in Appendix I.

## Contents

# A  Main results in the on-line Setting and tightness of Proposition 1

## A.1  Most general assumption

Assumption 3 should be formally written as follows:

**A3 (Oracle on the gradient)** *There exists a filtration $(\mathcal{H}_k^t)_{(t,k)\in[C]\times[N^t]}$ on some probability space $(\Omega, \mathcal{F}, \mathbb{P})$ such that for any $(t,k) \in [C] \times [N^t]$ and $\boldsymbol{w} \in \mathbb{R}^d$, $g_{p,k+1}^t(\boldsymbol{w})$ is a $\mathcal{H}_{k+1}^t$-measurable random variable and $\mathbb{E}\left[g_{p,k+1}^t(\boldsymbol{w})|\mathcal{H}_k^t\right] = F'(\boldsymbol{w})$. In addition, we assume the functions $(g_{p,k}^t)_{(t,k)\in[C]\times[N^t]}$ to be independent and identically distributed (i.i.d.) random fields.*

A filtration is an increasing (i.e., for all $(t,k) \preccurlyeq (t',k')$, $\mathcal{H}_k^t \subset \mathcal{H}_{k'}^{t'}$), sequence of $\sigma$-algebras. **A**3 expresses that we have access to an i.i.d. sequence $(g_{p,k}^t)_{(t,k)\in[C]\times[N^t]}$ of unbiased estimators of $F'$. Remark that with such notations, for any $t \in [C], k \in [N^t], p \in [P]$, $\boldsymbol{w}_{p,k}^t$ is $\mathcal{H}_k^t$-measurable.

## A.2  Main results: On-line Setting

In the on-line setting we consider the particular case of a decaying sequence $\eta_k^t = (\sum_{t'=1}^{t-1} N^{t'}+k)^{-\alpha}$, for some $\alpha \in (\frac{1}{2}, 1)$. The analysis is slightly more involved as Equation (3) results in more terms than in the finite horizon setting (sums do not directly telescope). While the decaying step-size case enables to improve some terms with respect to the finite horizon case (*e.g.* the speed at which one forgets the initial condition), the trade-offs concerning communication remain unchanged. We define the following constants to make the presentation clear, for $\alpha \in (1/2; 1)$:

$$R_{bias}(X) = 1 + X^{2\alpha} \exp\left(-\mu c_\eta X^{1-\alpha}\right) + \frac{1}{(\mu c_\eta)^{\frac{1}{1-\alpha}}} + \frac{M^2 c_\eta^2 \left\|\boldsymbol{w}^0 - \boldsymbol{w}^\star\right\|^2}{(\mu c_\eta)^{\frac{2}{1-\alpha}}} + \frac{2L^2 c_\eta^2}{P(\mu c_\eta)^{\frac{1}{1-\alpha}}},$$

$$R_{1,var}(X) = \frac{X^{2\alpha-1}P}{2\alpha-1} \exp\left(-\frac{\mu X^{1-\alpha}}{2(1-\alpha)}\right) + \frac{P}{X^{1-\alpha}}\frac{1}{c_\eta\mu} + \frac{P}{X\mu^{\frac{2\alpha}{1-\alpha}}c_\eta^{\frac{2}{1-\alpha}}} + \frac{L^2 P c_\eta^2}{X^\alpha \mu^2}$$

$$R_{2,var}(X) = \frac{M^2\sigma^2 P c_\eta^2}{\mu^2 X^{2\alpha-1}}.$$

Now we present a result similar to Proposition 1 for mini-batch averaging and one shot averaging:

**Proposition S7 (On-line Mini-batch Averaging and One-shot averaging)** *Under the Assumptions **A**1, **A**2, **A**3, **A**5, **A**6 using $\eta_k^t = (\sum_{t'=1}^{t-1} N^{t'} + k)^{-\alpha}$ we have for respectively mini-batch averaging and one-shot averaging:*

$$\mathbb{E}\left[\left\|\nabla^2 F(\boldsymbol{w}^\star)(\boldsymbol{w} - \boldsymbol{w}^\star)\right\|^2\right] \precsim \frac{\left\|\boldsymbol{w}^0 - \boldsymbol{w}^\star\right\|^2}{X^2 c_\eta^2} R_{bias}(X) + \frac{2\sigma^2}{T}\left(1 + \frac{R_{1,var}(X)}{\kappa} + \frac{R_{2,var}(X)}{\kappa^2}\right),$$

*with respectively $\kappa = 1$ and $X = N$ for one-shot averaging, and $\kappa = P$ and $X = C$ for mini-batch averaging.*

**Interpretation and comparison.** This proposition is directly derived from Lemma S59 in Appendix H. This proposition is similar to Propositions 1 and 2, but the overall convergence rate is better as using decaying step size eventually performs better. For example, the bias term mainly decays as $1/X^2$ instead of $1/(\eta X)^2$. This underlines why in practice decaying step size is often preferable. Asymptotically, the variance term is now dominant, and as before, MBA and OSA have similar performance as $\sigma^2 T^{-1}$.

**Optimal step size and asymptotic regimes for $P, T$** For a fixed number of machine $P$, the bias is asymptotically vanishing, and if we ignore the linearly decaying terms and the dependence on $\mu$, the resulting dominating term in $R_{1,2,var}$ is controlled by $X^{-\min\{(1-\alpha),\alpha,2\alpha-1\}}$, which would result in an optimal choice of $\alpha = 2/3$.

In the non asymptotic regime, where the total number of iterations and $P$ grow simultaneously, the variance of OSA scales as $T^{-1}$ as long as $PX^{-\min\{(1-\alpha),\alpha,2\alpha-1\}} = O(1)$. In other words, for $\alpha = 2/3$, we need $P \leq X^{1/3}$: the number of machines as to be smaller than the number of iterations to the power $1/3$, in other words, for 1000 iterations, one could only use 10 machines to reach the asymptotic regime where OSA performs similarly to MBA.

## A.3 Tight bias term for finite horizon setting

For a simple 1-dimensional quadratic function $F(w) = h(w - \boldsymbol{w}^\star)^2$, with $h > 0$, without any noise (we observe $y_i = w_0 x_i + \varepsilon_i$, with $\varepsilon_i \equiv 0$, and $x_i \equiv \sqrt{h}$), we have for a step size $\eta$, for any $p \in \{1, \ldots, P\}, k \leq N$:

$$\boldsymbol{w}_{p,k}^1 - \boldsymbol{w}^\star = (1 - \eta h)^k (w_0 - \boldsymbol{w}^\star) \tag{S1}$$

$$h^2 \left( N^{-1} \sum_{k=0}^{N-1} \boldsymbol{w}_{p,k}^1 - \boldsymbol{w}^\star \right)^2 = \frac{1 - (1 - \eta h)^N}{(\eta N)^2} (w_0 - \boldsymbol{w}^\star)^2, \tag{S2}$$

$$\implies h^2 \left( \overline{\overline{\boldsymbol{w}}}^C - \boldsymbol{w}^\star \right)^2 = \frac{1 - (1 - \eta h)^N}{(\eta N)^2} (w_0 - \boldsymbol{w}^\star)^2, \tag{S3}$$

which exactly matches the Bias term in (5) (for a quadratic, $M = 0$) and $L^2 \eta / hP \leq 1/P$ is a small constant ($\eta L \leq 1$ and $L = \mu = h$).

# B  Experimental results

Table S1: Data-sets for experimentation.

| Name of the Data-set | Task | Algorithm | Number of Samples | Number of Features |
|---|---|---|---|---|
| Epsilon | Classification | Logistic | 400000 | 2000 |
| Year Prediction MSD | Regression | Least-Squares | 463715 | 90 |
| CPU Stall | Regression | Least-Squares | 8192 | 12 |

Figure S1: Performance of Local SGD

We perform experiments for three different data-sets[5], two for least-squares regression and one for logistic regression Table S1. For all the curves we use $\log(y)$ v/s $\log(x)$ plots unless explicitly mentioned. Moreover, to elucidate the theory we use the same learning rates for all the algorithms in an experiment. The number of workers is set to $P = 32$ every where, and plots are labeled w.r.t.

the number of local steps $N$ which we don't change along the different phases. We do the following experiments:

Figure S2: Performance of Local SGD at the optimal

Figure S3: Variance of the loss function compared to MBA

Figure S4: Iterate Convergence of a single process against SGD and the ghost process.

1. Performance of local SGD with different number of local steps spanning OSA to MBA (Figure S1). We globally find MBA to perform the best. Besides, as we increase the number of local steps $N$ the performance gets closer to OSA. This observation aligns with our theoretical guarantees. We use the averaged iterate (i.e., $\overline{\overline{w}}$, the average over all the iterates

till that point) for reporting the performance. The current iterate (i.e., $\breve{\boldsymbol{w}}_k^t$, the ghost iterate for the current iteration) is omitted as the graphs are too noisy to be interpreted, and a variance of the loss is used instead.

2. Performance of local SGD with different number of local steps when started at the optimal point (Figure S2). We expect that if we start at $\boldsymbol{w}^\star$ then the bias term goes to zero and the difference between the algorithms becomes sharper. This is because our results predict that for constant learning rate, the initial conditions are forgotten at the same rate. We see that mini-batch outperforms OSA no the first iterations, but not asymptotically.

3. Variance of the estimators, for loss (Figure S3) and iterate values (Figure S4). We expect that a larger mini-batch size predicts a lower variance for these cases, and we observe the same through our experiments. In fact, the mean squared error of the parameters at the optimal is observed to be following a periodic curve. The value on an individual worker rises until it communicates, but always remains lower than a single SGD process run for the same number of iterations. This, verifies our theory and results for iterate convergence. Moreover, the variance at the loss function follows a similar pattern which elucidates the fact the intuitions developed in the paper also hold for functional convergence.

# C  Some Additional Material

## C.1  Pseudo codes

Pseudo codes of both algorithms are given in Figure S5.

```
1: procedure SGD
2:     Input: F : ℝᵈ → ℝ
3:     v₀ ← Initialize
4:     for t = 0,1,2,...T do
5:         gₜ(vₜ₋₁) ← SFO(F, vₜ₋₁)
6:         vₜ ← vₜ₋₁ − ηₜgₜ(vₜ₋₁)
7:     Output:
        S(v₀, v₁, .., v_{T−1}, v_T) ∈ ℝᵈ
```

Algorithm 1: Vanilla-SGD

```
1: procedure LOCAL-SGD
2:     Input: F : ℝᵈ → ℝ
3:     ŵ⁰ = w⁰ ← Initialize
4:     for t = 1, 2, ...C do
5:         parfor i=1,2,...P do
6:             wᵗᵢ,₀ ← ŵᵗ⁻¹
7:             for k=0,1,2,...Nᵗ do
8:                 gᵗᵢ,ₖ(wᵗᵢ,ₖ₋₁) ← SFO(F, wᵗᵢ,ₖ₋₁)
9:                 wᵗᵢ,ₖ ← wᵗᵢ,ₖ₋₁ − ηᵗₖgᵗᵢ,ₖ(wᵗᵢ,ₖ₋₁)
10:            w̄ᵗᵢ ← (1/Nₜ) Σ_{k=1}^{Nₜ} wᵗᵢ,ₖ
11:        end parfor
12:        w̄ᵗ ← (1/P) Σ_{i=1}^{P} w̄ᵗᵢ
13:        ŵᵗ ← (1/p) Σ_{i=1}^{P} wᵗᵢ,Nₜ
14:    Output: w̄̄ᵀ = (1/C) Σ_{t=1}^{C} w̄ᵗ ∈ ℝᵈ
```

Algorithm 2: Local-SGD

Figure S5: Serial and parallel SGD algorithms. **SFO** stands for the stochastic first order oracle. Note that every node has access to the full function i.e., the data is not distributed across nodes.

## C.2  Summary of Results

In the table below, we specify for which algorithm our results apply (mini batch, one shot, or local SGD), under which assumptions they are proved and if they apply to the on-line setting(OL) or just the finite horizon(FH) case.

## C.3  Example: Learning from i.i.d. observations

Our main motivation comes from machine learning; consider two sets $\mathcal{X}, \mathcal{Y}$ and a convex loss function $\ell : \mathcal{X} \times \mathcal{Y} \times \mathbb{R}^d \to \mathbb{R}$. The generalization error is defined as $F_\ell(\boldsymbol{w}) = \mathbb{E}_{X,Y}[\ell(X, Y, \boldsymbol{w})]$, where

| | | Assumptions | | | | | | | Setting | |
|---|---|---|---|---|---|---|---|---|---|---|
| Proposition | Algorithm | **A**1 | **A**2 | **Q**1 | **A**3 | **A**4 | **A**5 | **A**6 | FH | OL |
| Proposition 1 | Mini-Batch | ✓ | ✓ | | ✓ | | ✓ | ✓ | ✓ | |
| Proposition 2 | One-shot averaging | ✓ | ✓ | | ✓ | | ✓ | ✓ | ✓ | |
| Proposition S7 | Mini-Batch &OS | ✓ | ✓ | | ✓ | | ✓ | ✓ | | ✓ |
| Proposition 3 | Local SGD | | | ✓ | ✓ | ✓ | | | ✓ | |
| Corollary S17 | Local SGD | | | ✓ | ✓ | | ✓ | ✓ | ✓ | ✓ |
| Corollary S21 | Local SGD | ✓ | ✓ | | ✓ | ✓ | | | ✓ | ✓ |
| Theorem 6 1. | Local SGD | | | ✓ | ✓ | | ✓ | ✓ | ✓ | |
| Theorem 6 2. | Local SGD | ✓ | ✓ | | ✓ | ✓ | | | ✓ | |

Table S2: Summary of results

$(X, Y)$ are some random variables. Given i.i.d. observations $(X_k, Y_k)_{k \in \mathbb{N}^*}$ with the same distribution as $(X, Y)$, for any $k \in \mathbb{N}^*$, we define $f_k(\cdot) = \ell(X_k, Y_k, \cdot)$ the loss with respect to observation $k$. SGD can be used in two contexts:

1. *Stochastic Approximation*: We use *independent* observations at each iteration. The total number of iterations is thus at most the number of observations we access. SGD then corresponds to following the gradient of the loss $f_k$ on a single independent observation $(X_k, Y_k)$. As the gradients we use are then unbiased gradients of the generalization error, this means that SGD directly minimizes this (unknown) function.

2. *Empirical Risk Minimization*: We define the empirical risk as $\hat{F}_\ell(\boldsymbol{w}) = N^{-1} \sum_{k=1}^{N} [\ell(X_k, Y_k, \boldsymbol{w})]$. At each step $t$, we sample an index $i_t$ *uniformly on* $[N]$, and use the gradient of the loss $f_{i_t}$. Here the number of iterations is not limited, but the algorithm will converge to the minimum of the empirical risk.

In practice, this means that in the first situation, we want to optimize the precision of the algorithm for a limited number of oracle calls, while in the second situation one would rather optimize the number of outer iterations of the algorithm (*i.e.* its running time). In both these assumptions, Assumption **A**3 is satisfied for the filtration generated by all the observations before time $(t, k)$ (respectively all the indices sampled before time $(t, k)$).

Two typical situations regarding loss functions are worth mentioning. On the first hand, in *least-squares regression*, $\mathcal{X} = \mathbb{R}^d$, $\mathcal{Y} = \mathbb{R}$, and the loss function is $\ell(X, Y, \boldsymbol{w}) = (\langle X, \boldsymbol{w} \rangle - Y)^2$. Then $F_\Sigma$ is the quadratic function $\boldsymbol{w} \mapsto \left\| \Sigma^{1/2}(\boldsymbol{w} - \boldsymbol{w}^\star) \right\|^2 / 2$, with $\Sigma = \mathbb{E}[X X^\top]$, which satisfies Assumption **Q**1. For any $\boldsymbol{w} \in \mathbb{R}^d$,

$$f_k'(\boldsymbol{w}) - F_\Sigma'(\boldsymbol{w}) = (X_k X_k^\top - \Sigma)(\boldsymbol{w} - \boldsymbol{w}^\star) - (X_k^\top \boldsymbol{w}^\star - Y_k) X_k \tag{S4}$$

Then, Assumption **A**5 and **A**6 are satisfied, if $X$ is bounded and $Y$ has finite variance.

On the other hand, in *logistic regression*, where $\ell(X, Y, \boldsymbol{w}) = \log(1 + \exp(-Y \langle X, \boldsymbol{w} \rangle))$. Assumptions **A**2 and **A**4 are then satisfied [44], as is Assumption **A**1 under an additional restriction to a compact set or if an extra regularization is added.

SGD for least squares regression typically satisfies **Q**1, **A**3, **A**5 and **A**6. On the other hand, SGD for logistic regression satisfies **A**1, **A**2, **A**3 and **A**4.

# D Convergence guaranties for the second order moment.

In this section, we prove several Lemmas that allow to control the second order moment for the iterate. We first recall a few useful inequalities that will be used in the following. See for example [50].

If $F$ is convex and smooth (*e.g.* satisfies **A**2), the gradient of $F$ is cocoercive, thus for any $\boldsymbol{w} \in \mathbb{R}^d$:

$$L \langle F'(\boldsymbol{w}), \boldsymbol{w} - \boldsymbol{w}^\star \rangle \geq \|F'(\boldsymbol{w})\|. \tag{S5}$$

If the function is strongly-convex (Assumption **A**1), then for any $\boldsymbol{w} \in \mathbb{R}^d$:

$$\langle F'(\boldsymbol{w}), \boldsymbol{w} - \boldsymbol{w}^\star \rangle \geq \mu \|\boldsymbol{w} - \boldsymbol{w}^\star\|^2. \tag{S6}$$

## D.1 Inner iteration Lemma

We first recall the proof of the convergence for inner iterates. This proof corresponds to what happens on one machine, and can be found in the literature [10, 25] for example.

For any $p \in [P], t \in [C], k \in [N^t]$, under Assumptions **A**1, **A**2, **A**3, **A**5, **A**6, we have

$$\mathbb{E}\left[\left\|\boldsymbol{w}_{p,k}^t - \boldsymbol{w}^\star\right\|^2\right] \leq \mathbb{E}\left[\left\|\boldsymbol{w}_{p,k-1}^t - \boldsymbol{w}^\star\right\|^2\right] - \eta_k^t \left\langle F'(\boldsymbol{w}_{p,k-1}^t), \boldsymbol{w}_{p,k-1}^t - \boldsymbol{w}^\star \right\rangle + 2(\eta_k^t)^2 \sigma^2 \tag{S7}$$

$$\mathbb{E}\left[\left\|\boldsymbol{w}_{p,k}^t - \boldsymbol{w}^\star\right\|^2\right] \leq (1 - \eta_k^t \mu)\mathbb{E}\left[\left\|\boldsymbol{w}_{p,k-1}^t - \boldsymbol{w}^\star\right\|^2\right] + 2\eta_k^t \sigma^2.$$

Using the second equation recursively results in:

$$\mathbb{E}\left[\left\|\boldsymbol{w}_{p,k}^t - \boldsymbol{w}^\star\right\|^2\right] \leq \prod_{m=1}^{k} (1 - \eta_m^t \mu)\mathbb{E}\left[\left\|\boldsymbol{w}_{p,0}^t - \boldsymbol{w}^\star\right\|^2\right] + 2\sigma^2 \sum_{m=1}^{k} (\eta_m^t)^2 \prod_{l=m+1}^{k} (1 - \eta_l^t \mu). \tag{S8}$$

More precisely, for precise reference in the following proofs, we referenced this inequality with the following specific cases:

**Lemma S8** *Under Assumptions A1, A2, A3, A5, A6, for mini-batch SGD with batch-size $P$ and step-size $\eta$ we have,*

$$\mathbb{E}\left[\left\|\boldsymbol{w}_{MB}^t - \boldsymbol{w}^\star\right\|^2\right] \leq \prod_{m=1}^{t} (1 - \mu\eta)\mathbb{E}\left[\left\|\boldsymbol{w}^0 - \boldsymbol{w}^\star\right\|^2\right] + \frac{2\sigma^2 \eta^2}{P} \sum_{m=1}^{t} \prod_{l=m+1}^{t} (1 - \mu\eta).$$

Such a result on reduced variance for mini-batch SGD ($\frac{\sigma^2}{P}$) can be found in many previous works like [51]. Since mini-batch SGD is trivial to parallelize, this result also holds for the averaged iterate for outer iteration $t$ while using mini-batch averaging. Similarly, for decaying step sizes,

**Lemma S9** *Under Assumptions A1, A2, A3, A5, A6, and $\tilde{\eta}_t = \frac{c_\eta}{t^\alpha}$ for mini-batch SGD, for any $t \in [C]$ we have,*

$$\mathbb{E}\left[\left\|\boldsymbol{w}_{MB}^t - \boldsymbol{w}^\star\right\|^2\right] \leq \prod_{m=1}^{t} (1 - \mu\tilde{\eta}_m) \left\|\boldsymbol{w}^0 - \boldsymbol{w}^\star\right\|^2 + 2\sigma^2 \frac{1}{P} \sum_{m=1}^{t} (\tilde{\eta}_m)^2 \prod_{l=m+1}^{t} (1 - \mu\tilde{\eta}_l).$$

Similarly, in the case of one-shot averaging,

**Lemma S10** *Under Assumptions A1, A2, A3, A5, A6 and a constant step-size $\eta$ using one-shot averaging, for any $K \in [N^1]$ and $i \in [P]$ we have,*

$$\mathbb{E}\left[\left\|\boldsymbol{w}_{i,K}^1 - \boldsymbol{w}^\star\right\|^2\right] \leq \prod_{m=1}^{K} (1 - \mu\eta) \left\|\boldsymbol{w}^0 - \boldsymbol{w}^\star\right\|^2 + 2\sigma^2 \eta^2 \sum_{m=1}^{K} \prod_{l=m+1}^{K} (1 - \mu\eta_l^1).$$

**Lemma S11** *Under Assumptions A1, A2, A3, A5, A6, and $\eta_k^1 = \tilde{\eta}_k = \frac{c_\eta}{k^\alpha}$ using one-shot averaging for any $K \in [N^1]$ and $i \in [P]$ we have,*

$$\mathbb{E}\left[\left\|\boldsymbol{w}_{i,K}^1 - \boldsymbol{w}^\star\right\|^2\right] \leq \prod_{m=1}^{K} (1 - \mu\eta_m^1) \left\|\boldsymbol{w}^0 - \boldsymbol{w}^\star\right\|^2 + 2\sigma^2 \sum_{m=1}^{K} (\eta_m^1)^2 \prod_{l=m+1}^{K} (1 - \mu\eta_l^1).$$

## D.2 Proof of Proposition 3

In this Section we prove Proposition 3. In order to provide a bound on the mean squared distance to the optimum of the outer iterates, we introduce a *ghost* sequence [52], *i.e.*, a sequence of iterates which is not actually computed. For any $t \in [C], k \in [N^t]$, we define

$$\breve{\boldsymbol{w}}_k^t := \frac{1}{P} \sum_{i=1}^P \boldsymbol{w}_{i,k}^t. \tag{S9}$$

We prove the following Lemma:

**Lemma S12** *Under Assumptions **Q**1, **A**3 and **A**4, for any $t \in [C], K \in [N^t]$, we have:*

$$\mathbb{E}\left[\left\|\breve{\boldsymbol{w}}_K^t - \boldsymbol{w}^\star\right\|^2\right] \leq \prod_{m=1}^K (1 - \mu \eta_m^t) \left\|\breve{\boldsymbol{w}}_0^t - \boldsymbol{w}^\star\right\|^2 + \frac{\sigma_\infty^2}{P} \sum_{m=1}^K (\eta_m^t)^2 \prod_{l=m+1}^K (1 - \mu \eta_l^t). \tag{S10}$$

Remarking that for any $t \in [C]$, $\breve{\boldsymbol{w}}_{N^t}^t = \hat{\boldsymbol{w}}^t$ this implies the *first inequality* of Proposition 3. Note that this Lemma is valid for both decaying steps and and a constant learning rate. Especially, for a constant step size $\eta$, and $K = N^t$:

$$\mathbb{E}\left[\left\|\hat{\boldsymbol{w}}^t - \boldsymbol{w}^\star\right\|^2\right] \leq (1 - \mu \eta)^{N^t} \left\|\hat{\boldsymbol{w}}^{t-1} - \boldsymbol{w}^\star\right\|^2 + \frac{\sigma_\infty^2}{P} \eta \frac{1 - (1 - \mu \eta)^{N^t}}{\mu}.$$

More generally, we also have the following corollary, if we denote $(\tilde{\eta}_k)_{k \geq 0}$ the sequence such that $\eta_k^t = \tilde{\eta}_{\{\sum_{t'=1}^{t-1} N^{t'} + k\}}$ (this just corresponds to re-indexing the sequence):

**Corollary S13** *Under Assumptions **Q**1, **A**3 and **A**4, for any $T \in [C]$, we have:*

$$\mathbb{E}\left[\left\|\hat{\boldsymbol{w}}^T - \boldsymbol{w}^\star\right\|^2\right] \leq \prod_{k=1}^{\sum_{t=1}^T N^t} (1 - \mu \tilde{\eta}_k) \|\boldsymbol{w}_0 - \boldsymbol{w}^\star\|^2 + \frac{\sigma_\infty^2}{P} \sum_{t=1}^{\sum_{t=1}^T N^t} \tilde{\eta}_k^2 \prod_{j=k+1}^{\sum_{t=1}^T N^t} (1 - \mu \tilde{\eta}_j). \tag{S11}$$

**Proof 14 (Proof of Corollary S13)** *By induction, Lemma S12 implies that for any $T \in [C]$*

$$\mathbb{E}\left[\left\|\hat{\boldsymbol{w}}^T - \boldsymbol{w}^\star\right\|^2\right] \leq \prod_{t=1}^T \prod_{k=1}^{N^t} (1 - \mu \eta_k^t) \|\boldsymbol{w}_0 - \boldsymbol{w}^\star\|^2 + \frac{\sigma_\infty^2}{P} \sum_{t=1}^T \prod_{t'=t+1}^T \prod_{k=1}^{N^{t'}} (1 - \mu \eta_k^t) \sum_{k=1}^{N^t} (\eta_k^t)^2 \prod_{j=k+1}^{N^t} (1 - \mu \eta_j^t). \tag{S12}$$

*Then using $\eta_k^t = \tilde{\eta}_{\{\sum_{t'=1}^{t-1} N^{t'} + k\}}$, the corollary is just re-writing of Equation (S12).*

To prove the second inequality of Proposition 3, we combine Lemma S12 and Equation (S8), using the fact that $\boldsymbol{w}_{p,0}^t = \hat{\boldsymbol{w}}^{t-1}$.

This results means that for a quadratic function with gradients having uniformly bounded variance, the outer iteration decay is the same as for mini-batch iterations (but for mini-batch, it is true under the weaker set of Assumptions **A**1, **A**2, **A**3, **A**5, **A**6).

### D.2.1 Proof

**Proof 15 (Proof of Lemma S12)** *By definition of $\breve{\boldsymbol{w}}_k^t$, we have for any $t \in [C], k \in [N^t]$, using the linearity of $F'$ (Assumption **Q**1):*

$$\frac{1}{P} \sum_{i=1}^P \boldsymbol{w}_{i,k+1}^t = \frac{1}{P} \sum_{i=1}^P \boldsymbol{w}_{i,k}^t - \frac{1}{P} \sum_{i=1}^P \eta_{k+1}^t g_{i,k+1}^t (\boldsymbol{w}_{i,k}^t)$$

$$\breve{\boldsymbol{w}}_{k+1}^t - \boldsymbol{w}^\star = \breve{\boldsymbol{w}}_k^t - \boldsymbol{w}^\star - \frac{1}{P} \sum_{i=1}^P \eta_{k+1}^t g_{i,k+1}^t (\boldsymbol{w}_{i,k}^t)$$

$$\mathbb{E}\left[\left\|\breve{\boldsymbol{w}}_{k+1}^t - \boldsymbol{w}^\star\right\|^2 | \mathcal{H}_{k,t}\right] \leq \left\|\breve{\boldsymbol{w}}_k^t - \boldsymbol{w}^\star\right\|^2 - 2\eta_{k+1}^t \langle \breve{\boldsymbol{w}}_k^t - \boldsymbol{w}^\star, F'(\breve{\boldsymbol{w}}_k^t) \rangle$$

$$+ (\eta_{k+1}^t)^2 \mathbb{E}\left[\left\|\frac{1}{P}\sum_{i=1}^P g_{i,k+1}^t(\boldsymbol{w}_{i,k}^t)\right\|^2 |\mathcal{H}_{k,t}\right] . \qquad \text{(S13)}$$

*Now analyzing just the last term,*

$$(\eta_{k+1}^t)^2 \mathbb{E}\left[\left\|\frac{1}{P}\sum_{i=1}^P g_{i,k+1}^t(\boldsymbol{w}_{i,k}^t)\right\|^2 |\mathcal{H}_{k,t}\right]$$

$$= (\eta_{k+1}^t)^2 \mathbb{E}\left[\left\|\frac{1}{P}\sum_{i=1}^P \left(g_{i,k+1}^t(\boldsymbol{w}_{i,k}^t) - F'(\boldsymbol{w}_{i,k}^t)\right)\right\|^2 |\mathcal{H}_{k,t}\right] + (\eta_{k+1}^t)^2 \left\|F'(\breve{\boldsymbol{w}}_k^t)\right\|^2 . \qquad \text{(S14)}$$

*Under the independence of the noises (Assumption A3), then the uniform upper bound on the variance (Assumption A4), we have the following upper bound :*

$$\mathbb{E}\left[\left\|\frac{1}{P}\sum_{i=1}^P \left(g_{i,k+1}^t(\boldsymbol{w}_{i,k}^t) - F'(\boldsymbol{w}_{i,k}^t)\right)\right\|^2 |\mathcal{H}_{k,t}\right] = \frac{1}{P^2}\sum_{i=1}^P \mathbb{E}\left[\left\|\left(g_{i,k+1}^t(\boldsymbol{w}_{i,k}^t) - F'(\boldsymbol{w}_{i,k}^t)\right)\right\|^2 |\mathcal{H}_{k,t}\right]$$

$$\leq \frac{1}{P}\sigma_\infty^2 .$$

*Under Assumption Q1, $F'$ is co-coercive, thus using Equation (S5), we have the following upper bound:*

$$\mathbb{E}\left[\left\|\breve{\boldsymbol{w}}_{k+1}^t - \boldsymbol{w}^\star\right\|^2 |\mathcal{H}_{k,t}\right] \leq \left\|\breve{\boldsymbol{w}}_k^t - \boldsymbol{w}^\star\right\|^2 - 2\eta_{k+1}^t(1 - \eta_{k+1}^t L)\langle\breve{\boldsymbol{w}}_k^t - \boldsymbol{w}^\star, F'(\breve{\boldsymbol{w}}_k^t)\rangle + \frac{(\eta_{k+1}^t)^2\sigma_\infty^2}{P} .$$

*And using strong convexity (esp. Equation (S6)), and the fact that $\eta_{k+1}^t L \leq \frac{1}{2}$:*

$$\mathbb{E}\left[\left\|\breve{\boldsymbol{w}}_{k+1}^t - \boldsymbol{w}^\star\right\|^2 |\mathcal{H}_{k,t}\right] \leq (1 - 2\mu\eta_{k+1}^t(1 - \eta_{k+1}^t L))\left\|\breve{\boldsymbol{w}}_k^t - \boldsymbol{w}^\star\right\|^2 + \frac{(\eta_{k+1}^t)^2\sigma_\infty^2}{P}$$

$$\leq (1 - \mu\eta_{k+1}^t)\left\|\breve{\boldsymbol{w}}_k^t - \boldsymbol{w}^\star\right\|^2 + \frac{(\eta_{k+1}^t)^2\sigma_\infty^2}{P} . \qquad \text{(S15)}$$

*By recursion, we then have, for any $K \in [N^t]$:*

$$\mathbb{E}\left[\left\|\breve{\boldsymbol{w}}_K^t - \boldsymbol{w}^\star\right\|^2\right] \leq \prod_{k=1}^K (1 - \mu\eta_k^t)\left\|\breve{\boldsymbol{w}}_0^t - \boldsymbol{w}^\star\right\|^2 + \frac{\sigma_\infty^2}{P}\sum_{k=1}^K (\eta_k^t)^2 \prod_{j=k}^K (1 - \mu\eta_j^t) .$$

*This concludes the proof.*

### D.3 Proof of Proposition S16

In this Section we prove Proposition S16.

#### D.3.1 Statement of Proposition S16

**Proposition S16 (Local-SGD: Quadratic Functions)** *Under Assumptions Q1,A3,A5,A6, we have the following bound for one shot averaging: $p \in [P], t \in [C], k \in [N^t]$,*

$$\mathbb{E}\left[\left\|\hat{\boldsymbol{w}}^t - \boldsymbol{w}^\star\right\|^2\right] \leq \kappa_2^t \prod_{k=1}^{\sum_{t'=1}^t N^{t'}} (1 - \mu\tilde{\eta}_k)\left\|\boldsymbol{w}_0 - \boldsymbol{w}^\star\right\|^2 + 2\kappa_1^t\kappa_2^t\frac{\sigma^2}{P}\sum_{t=1}^{\sum_{k=1}^t N^t} \tilde{\eta}_k^2 \prod_{j=k+1}^{\sum_{t'=1}^t N^{t'}} (1 - \mu\tilde{\eta}_j)$$

$$\text{(S16)}$$

$$\mathbb{E}\left[\left\|\boldsymbol{w}_{p,k}^t - \boldsymbol{w}^\star\right\|^2\right] \leq \kappa_2^t \prod_{k=1}^{\sum_{t'=1}^t N^{t'}+k} (1 - \mu\tilde{\eta}_k)\left\|\boldsymbol{w}_0 - \boldsymbol{w}^\star\right\|^2 + 2\kappa_1^t\kappa_2^t\frac{\sigma^2}{P}\sum_{u=1}^{\sum_{t'=1}^t N^{t'}} \tilde{\eta}_u^2 \prod_{j=k+1}^{\sum_{t'=1}^t N^{t'}+k} (1 - \mu\tilde{\eta}_j)$$

$$+ 2\frac{\sigma^2}{P} \sum_{u=\sum_{t'=1}^{t} N^{t'}}^{\sum_{t'=1}^{t} N^{t'}+k} \tilde{\eta}_u^2 \prod_{j=u+1}^{\sum_{t'=1}^{t} N^{t'}+k} (1 - \mu\tilde{\eta}_j), \tag{S17}$$

with, for $t \in [C]$, $\kappa_1^t = \left(4 + \mu \sum_{k=1}^{N^t}(\eta_k^t)^2\right)$, and $\kappa_2^t := \exp\left(\mu \sum_{t'=0}^{t} \sum_{k=1}^{N^t}(\eta_k^t)^2\right)$.

When considering a constant step size $\eta$, we have the following corollary.

**Corollary S17 (Local-SGD: Quadratic Functions)** *Under Assumptions **Q**1,**A**3,**A**5,**A**6, we have the following bound for one shot averaging: $p \in [P], t \in [C], k \in [N^t]$, constant learning rate $\eta$,*

$$\mathbb{E}\left[\left\|\hat{\boldsymbol{w}}^{t-1} - \boldsymbol{w}^\star\right\|^2\right] \leq \tau_2^t (1 - \eta\mu)^{\boldsymbol{N}_1^{t-1}} \|\boldsymbol{w}_0 - \boldsymbol{w}^\star\|^2 + 2\tau_1^t \tau_2^t \frac{\sigma^2 \eta}{P} \frac{1 - (1 - \eta\mu)^{\boldsymbol{N}_1^{t-1}}}{\mu} \tag{S18}$$

$$\mathbb{E}\left[\left\|\boldsymbol{w}_{p,k}^t - \boldsymbol{w}^\star\right\|^2\right] \leq \tau_2^t (1 - \eta\mu)^{\boldsymbol{N}_1^{t-1}+k} \|\boldsymbol{w}_0 - \boldsymbol{w}^\star\|^2$$

$$+ 2\sigma^2 \eta \left(\sup_{t'=1...t}(\tau_1^t)\tau_2^t \frac{1 - (1 - \eta\mu)^{\boldsymbol{N}_1^{t-1}}}{P\mu} + \frac{1 - (1 - \eta\mu)^k}{\mu}\right). \tag{S19}$$

Where we have $\tau_1^t = 4 + \mu\boldsymbol{N}_1^t \eta^2$ and $\tau_2^t = \exp\left(\mu\boldsymbol{N}_1^t \eta^2\right)$. Under the latter requirement (for optimality) that for any $t$, $\boldsymbol{N}^t \mu P \eta \leq 1$, we have $\mu\boldsymbol{N}_1^t \eta^2 \leq C\eta P^{-1}$, thus this is generally a small constant. This result is a consequence of Lemma S18.

**Interpretation.** As before, the first bound shows that the variance of the iterates *after communication* is reduced by a factor of $P$ w.r.t. the serial case, thus almost as good as mini-batch averaging. However, the constants involved are worse than in the additive noise setting (Proposition 3). Consequently, and similarly to Proposition 3, the bound for the current iterates is composed of two terms for the variance: a "reduced variance" coming from the communication step, and a "inner loop" variance, that does not benefit from the number of machines.

Finally, we provide a convergence result in the most general case, removing the quadratic assumption. For the sake of concision, we skip the bound for the averaged iterate after a communication round, and directly give the result for the inner process.

### D.3.2 Proof

This result is a consequence of Lemma S18, which implies Equation (S18). Indeed, using it recursively, and using $(1 + x) \leq \exp(x)$, we get:

$$\mathbb{E}\left[\left\|\hat{\boldsymbol{w}}^T - \boldsymbol{w}^\star\right\|^2\right] \leq \exp\left(\mu \sum_{t'=0}^{T}\sum_{k=1}^{N^t}(\eta_k^t)^2\right) \prod_{t'=1}^{T}\prod_{k=1}^{N^{t'}}(1 - \mu\tilde{\eta}_k^t)\mathbb{E}\left[\|\boldsymbol{w}_0 - \boldsymbol{w}^\star\|^2\right]$$

$$+ 2\kappa_1 \exp\left(\mu \sum_{t'=0}^{t}\sum_{k=1}^{N^t}(\eta_k^t)^2\right) \frac{\sigma^2}{P} \sum_{t=1}^{T} \prod_{t'=t+1}^{T}\prod_{k=1}^{N^{t'}}(1 - \mu\eta_k^t)\sum_{k=1}^{N^t}(\eta_k^t)^2 \prod_{j=k+1}^{N^t}(1 - \mu\eta_j^t)$$

With, for $t \in [C]$, $\kappa_1^t = \left(4 + \mu\sum_{k=1}^{N^t}(\eta_k^t)^2\right)$, and $\kappa_2^t := \exp\left(\mu\sum_{t'=0}^{t}\sum_{k=1}^{N^t}(\eta_k^t)^2\right)$, and rewriting everything in terms of the sequence $\tilde{\eta}_k$, it gives Equation (S16). The second inequality naturally follows.

**Lemma S18** *Under Assumptions **Q**1, **A**3, **A**5, **A**6, for any $t \in [C], K \in [N^t]$, we have:*

$$\mathbb{E}\left[\left\|\hat{\boldsymbol{w}}^t - \boldsymbol{w}^\star\right\|^2\right] \leq \left(1 + \mu\sum_{k=1}^{N^t}(\eta_k^t)^2\right) \prod_{k=1}^{N^t}(1 - \mu\eta_k)\mathbb{E}\left[\left\|\hat{\boldsymbol{w}}^{t-1} - \boldsymbol{w}^\star\right\|^2\right] \tag{S20}$$

$$+ 2\left(4 + \mu\sum_{k=1}^{N^t}(\eta_k^t)^2\right)\frac{\sigma^2}{P}\sum_{k=0}^{N^t}(\eta_k^t)^2 \prod_{j=k+1}^{N^t}(1 - \mu\eta_j^t). \tag{S21}$$

The proof is a bit technical, so we summarize here the 2 main steps:

1. We prove an inequality (namely Equation (S23)) that is comparable to Equation (S15), but with an extra term.
2. We use the control on the inner process (Appendix D.1) to control the extra term.

**Proof 19** *We consider again the ghost process defined at Equation (S9). Equations (S13) and (S14) are still valid. We now use the following decomposition[6]:*

$$\square := (\eta_{k+1}^t)^2 \mathbb{E}\left[\left\|\frac{1}{P}\sum_{i=1}^P g_{i,k+1}^t(\boldsymbol{w}_{i,k}^t)\right\|^2 |\mathcal{H}_{k,t}\right]$$

$$= (\eta_{k+1}^t)^2 \mathbb{E}\left[\left\|\frac{1}{P}\sum_{i=1}^P \left(g_{i,k+1}^t(\boldsymbol{w}_{i,k}^t) - F'(\boldsymbol{w}_{i,k}^t)\right)\right\|^2 |\mathcal{H}_{k,t}\right] + (\eta_{k+1}^t)^2 \left\|F'(\breve{\boldsymbol{w}}_k^t)\right\|^2$$

$$\leq 2(\eta_{k+1}^t)^2 \mathbb{E}\left[\left\|\frac{1}{P}\sum_{i=1}^P \left(g_{i,k+1}^t(\boldsymbol{w}_{i,k}^t) - F'(\boldsymbol{w}_{i,k}^t) - g_{i,k+1}^t(\boldsymbol{w}^\star)\right)\right\|^2 |\mathcal{H}_{k,t}\right]$$

$$+ 2(\eta_{k+1}^t)^2 \mathbb{E}\left[\left\|\frac{1}{P}\sum_{i=1}^P g_{i,k+1}^t(\boldsymbol{w}^\star)\right\|^2 |\mathcal{H}_{k,t}\right] + (\eta_{k+1}^t)^2 \left\|F'(\breve{\boldsymbol{w}}_k^t)\right\|^2 .$$

*Using the independence of the noises (Assumption A3) we have,*

$$\square \leq \frac{2(\eta_{k+1}^t)^2}{P^2}\sum_{i=1}^P \mathbb{E}\left[\left\|\left(g_{i,k+1}^t(\boldsymbol{w}_{i,k}^t) - F'(\boldsymbol{w}_{i,k}^t) - g_{i,k+1}^t(\boldsymbol{w}^\star)\right)\right\|^2 |\mathcal{H}_{k,t}\right]$$

$$+ \frac{2(\eta_{k+1}^t)^2}{P}\mathbb{E}\left[\left\|g_{i,k+1}^t(\boldsymbol{w}^\star)\right\|^2 |\mathcal{H}_{k,t}\right] + (\eta_{k+1}^t)^2 \left\|F'(\breve{\boldsymbol{w}}_k^t)\right\|^2$$

$$\leq \frac{4(\eta_{k+1}^t)^2}{P^2}\sum_{i=1}^P \left(\mathbb{E}\left[\left\|\left(g_{i,k+1}^t(\boldsymbol{w}_{i,k}^t) - g_{i,k+1}^t(\boldsymbol{w}^\star)\right)\right\|^2 |\mathcal{H}_{k,t}\right] + \mathbb{E}\left[\left\|\left(F'(\boldsymbol{w}_{i,k}^t) - F'(\boldsymbol{w}^\star)\right)\right\|^2 |\mathcal{H}_{k,t}\right]\right)$$

$$+ \frac{2(\eta_{k+1}^t)^2}{P}\mathbb{E}\left[\left\|g_{i,k+1}^t(\boldsymbol{w}^\star)\right\|^2 |\mathcal{H}_{k,t}\right] + (\eta_{k+1}^t)^2 \left\|F'(\breve{\boldsymbol{w}}_k^t)\right\|^2 .$$

*Using Assumption A5 (co-coercivity for $(g_{i,k}^t)$-s and $F$) we obtain,*

$$\square \leq \frac{8L(\eta_{k+1}^t)^2}{P^2}\sum_{i=1}^P \left\langle F'(\boldsymbol{w}_{i,k}^t) - F'(\boldsymbol{w}^\star), \boldsymbol{w}_{i,k}^t - \boldsymbol{w}^\star\right\rangle + \frac{2(\eta_{k+1}^t)^2}{P}\mathbb{E}\left[\left\|g_{i,k+1}^t(\boldsymbol{w}^\star)\right\|^2 |\mathcal{H}_{k,t}\right]$$

$$+ (\eta_{k+1}^t)^2 L\left\langle F'(\breve{\boldsymbol{w}}_k^t), \breve{\boldsymbol{w}}_k^t - \boldsymbol{w}^\star\right\rangle . \tag{S22}$$

*This leads to, combining Equations (S13) and (S22), and the upper bound on the variance of the noise at the optimum (Assumption A6)*

$$\diamond := \mathbb{E}\left[\left\|\breve{\boldsymbol{w}}_{k+1}^t - \boldsymbol{w}^\star\right\|^2 |\mathcal{H}_{k,t}\right]$$

$$\leq \left\|\breve{\boldsymbol{w}}_k^t - \boldsymbol{w}^\star\right\|^2 - 2\eta_{k+1}^t \left\langle \breve{\boldsymbol{w}}_k^t - \boldsymbol{w}^\star, F'(\breve{\boldsymbol{w}}_k^t)\right\rangle + \frac{2(\eta_{k+1}^t)^2}{P}\mathbb{E}\left[\left\|g_{i,k+1}^t(\boldsymbol{w}^\star)\right\|^2 |\mathcal{H}_{k,t}\right]$$

$$+ \frac{8L(\eta_{k+1}^t)^2}{P^2}\sum_{i=1}^P \left\langle F'(\boldsymbol{w}_{i,k}^t) - F'(\boldsymbol{w}^\star), \boldsymbol{w}_{i,k}^t - \boldsymbol{w}^\star\right\rangle + (\eta_{k+1}^t)^2 L\left\langle F'(\breve{\boldsymbol{w}}_k^t), \breve{\boldsymbol{w}}_k^t - \boldsymbol{w}^\star\right\rangle$$

$$\leq \left\|\breve{\boldsymbol{w}}_k^t - \boldsymbol{w}^\star\right\|^2 - 2\eta_{k+1}^t(1 - \eta_{k+1}^t L)\left\langle \breve{\boldsymbol{w}}_k^t - \boldsymbol{w}^\star, F'(\breve{\boldsymbol{w}}_k^t)\right\rangle + 2\frac{(\eta_{k+1}^t)^2}{P}\sigma^2$$

$$+ \frac{8L(\eta_{k+1}^t)^2}{P^2}\sum_{i=1}^P \left\langle F'(\boldsymbol{w}_{i,k}^t) - F'(\boldsymbol{w}^\star), \boldsymbol{w}_{i,k}^t - \boldsymbol{w}^\star\right\rangle .$$

*Using $L\eta_{k+1}^t \leq \frac{1}{2}$, and strong-convexity (Assumption A1)*

$$\mathbb{E}\left[\left\|\breve{\boldsymbol{w}}_{k+1}^t - \boldsymbol{w}^\star\right\|^2 |\mathcal{H}_{k,t}\right] \leq (1 - \mu\eta_{k+1}^t)\left\|\breve{\boldsymbol{w}}_k^t - \boldsymbol{w}^\star\right\|^2 + \frac{2(\eta_{k+1}^t)^2\sigma^2}{P}$$

$$+ \frac{8L(\eta_{k+1}^t)^2}{P^2}\sum_{i=1}^P \left\langle F'(\boldsymbol{w}_{i,k}^t) - F'(\boldsymbol{w}^\star), \boldsymbol{w}_{i,k}^t - \boldsymbol{w}^\star\right\rangle. \qquad \text{(S23)}$$

*This inequality should be compared to Equation (S15). It is interesting to remark that the last term is not an artifact of the proof: this is easy to check for least-squares regression.*

*Using recursively the above inequality and using the definition of $\breve{\boldsymbol{w}}^t$, and taking expectation on the historical randomness we have, for any $N \in [N^t - 1]$*

$$\mathbb{E}\left[\left\|\breve{\boldsymbol{w}}_{N+1}^t - \boldsymbol{w}^\star\right\|^2\right] \leq \prod_{k=0}^N (1 - \mu\eta_{k+1}^t)\mathbb{E}\left[\left\|\breve{\boldsymbol{w}}_0^t - \boldsymbol{w}^\star\right\|^2\right] + 2\frac{\sigma^2}{P}\sum_{k=0}^N (\eta_{k+1}^t)^2 \prod_{j=k+1}^N (1 - \mu\eta_{j+1}^t)$$

$$+ \frac{8L}{P^2}\mathbb{E}\left[\sum_{k=0}^N (\eta_{k+1}^t)^2 \sum_{i=1}^P \left\langle F'(\boldsymbol{w}_{i,k}^t) - F'(\boldsymbol{w}^\star), \boldsymbol{w}_{i,k}^t - \boldsymbol{w}^\star\right\rangle \prod_{j=k+1}^N (1 - \mu\eta_{j+1}^t)\right].$$

*Especially, for $N = N^t - 1$, $\breve{\boldsymbol{w}}_{N^t}^t = \hat{\boldsymbol{w}}^t$, and moreover $\breve{\boldsymbol{w}}_0^t = \hat{\boldsymbol{w}}^{t-1}$:*

$$\mathbb{E}\left[\left\|\hat{\boldsymbol{w}}^t - \boldsymbol{w}^\star\right\|^2\right] \leq \prod_{k=0}^{N^t-1} (1 - \mu\eta_{k+1}^t)\mathbb{E}\left[\left\|\hat{\boldsymbol{w}}^{t-1} - \boldsymbol{w}^\star\right\|^2\right] + 2\frac{\sigma^2}{P}\sum_{k=0}^{N^t-1} (\eta_{k+1}^t)^2 \prod_{j=k+1}^{N^t-1} (1 - \mu\eta_{j+1}^t)$$

$$+ \frac{8L}{P^2}\mathbb{E}\left[\sum_{k=0}^{N^t-1} (\eta_{k+1}^t)^2 \sum_{i=1}^P \left\langle F'(\boldsymbol{w}_{i,k}^t) - F'(\boldsymbol{w}^\star), \boldsymbol{w}_{i,k}^t - \boldsymbol{w}^\star\right\rangle \prod_{j=k+1}^{N^t-1} (1 - \mu\eta_{j+1}^t)\right].$$

$$\text{(S24)}$$

*To upper bound the last term in the above equation, we use Equation (S7),*

$$\clubsuit := \frac{8L}{P^2}\sum_{k=0}^{N^t-1} (\eta_{k+1}^t)^2 \sum_{i=1}^P \left\langle F'(\boldsymbol{w}_{i,k}^t) - F'(\boldsymbol{w}^\star), \boldsymbol{w}_{i,k}^t - \boldsymbol{w}^\star\right\rangle \prod_{j=k+1}^{N^t-1} (1 - \mu\eta_{j+1}^t)$$

$$\leq \frac{8L}{P^2}\sum_{k=0}^{N^t-1} \eta_{k+1}^t \sum_{i=1}^P \left(\mathbb{E}\left[\left\|\boldsymbol{w}_{i,k}^t - \boldsymbol{w}^\star\right\|^2\right] - \mathbb{E}\left[\left\|\boldsymbol{w}_{i,k+1}^t - \boldsymbol{w}^\star\right\|^2\right]\right.$$

$$+ 2(\eta_{k+1}^t)^2\sigma^2\Bigg)\prod_{j=k+1}^{N^t-1} (1 - \mu\eta_{j+1}^t)$$

$$\leq \frac{8L}{P^2}\sum_{k=0}^{N^t-1} \eta_{k+1}^t \sum_{i=1}^P \left(\mathbb{E}\left[\left\|\boldsymbol{w}_{i,k}^t - \boldsymbol{w}^\star\right\|^2\right] - \mathbb{E}\left[\left\|\boldsymbol{w}_{i,k+1}^t - \boldsymbol{w}^\star\right\|^2\right]\right)\prod_{j=k+1}^{N^t-1} (1 - \mu\eta_{j+1}^t)$$

$$+ \frac{16L\sigma^2}{P}\sum_{k=0}^{N^t-1} (\eta_{k+1}^t)^3 \prod_{j=k+1}^{N^t-1} (1 - \mu\eta_{j+1}^t).$$

*Note that since the mean squared distance doesn't depend on the machine, we can assume to be working on machine 1. This leads to, using an Abel transform:*

$$\clubsuit \leq \frac{8L}{P}\sum_{k=0}^{N^t-1} \left(\mathbb{E}\left[\left\|\boldsymbol{w}_{1,k}^t - \boldsymbol{w}^\star\right\|^2\right] - \mathbb{E}\left[\left\|\boldsymbol{w}_{1,k+1}^t - \boldsymbol{w}^\star\right\|^2\right]\right)\prod_{j=k+1}^{N^t-1} (1 - \mu\eta_{j+1}^t)\eta_{k+1}^t$$

$$+ \frac{16L\sigma^2}{P}\sum_{k=0}^{N^t-1} (\eta_{k+1}^t)^3 \prod_{j=k+1}^{N^t-1} (1 - \mu\eta_{j+1}^t)$$

$$\leq \frac{8L}{P}\left(\sum_{k=0}^{N^t-1}\mathbb{E}\left[\|\boldsymbol{w}_{1,k}^t-\boldsymbol{w}^\star\|^2\right]\left(\eta_{k+1}^t\prod_{j=k+1}^{N^t-1}(1-\mu\eta_{j+1}^t)-\eta_k^t\prod_{j=k}^{N^t-1}(1-\mu\eta_{j+1}^t)\right)\right.$$

$$\left.+\mathbb{E}\left[\|\boldsymbol{w}_{1,0}^t-\boldsymbol{w}^\star\|^2\right]\prod_{j=0}^{N^t-1}(1-\mu\eta_{j+1}^t)\eta_0^t-\mathbb{E}\left[\|\boldsymbol{w}_{1,N^t}^t-\boldsymbol{w}^\star\|^2\right]\eta_{N^t}^t\right)$$

$$+\frac{16L\sigma^2}{P}\sum_{k=0}^{N^t-1}(\eta_{k+1}^t)^3\prod_{j=k+1}^{N^t-1}(1-\mu\eta_{j+1}^t).$$

*Finally, using convexity, we have that*

$$\mathbb{E}\left[\|\breve{\boldsymbol{w}}_{N^t}^t-\boldsymbol{w}^\star\|^2\right]\leq\frac{1}{P}\sum_{p=1}^P\mathbb{E}\left[\|\boldsymbol{w}_{p,N^t}^t-\boldsymbol{w}^\star\|^2\right]=\mathbb{E}\left[\|\boldsymbol{w}_{1,N^t}^t-\boldsymbol{w}^\star\|^2\right].$$

*Thus:*

$$\clubsuit\leq\frac{8L}{P}\sum_{k=0}^{N^t-1}\mathbb{E}\left[\|\boldsymbol{w}_{1,k}^t-\boldsymbol{w}^\star\|^2\right]\prod_{j=k+1}^{N^t-1}(1-\mu\eta_{j+1})\big(\eta_{k+1}^t-\eta_k^t(1-\mu\eta_{k+1}^t)\big)$$

$$+\frac{8L}{P}\mathbb{E}\left[\|\hat{\boldsymbol{w}}^{t-1}-\boldsymbol{w}^\star\|^2\right]\prod_{j=0}^{N^t-1}(1-\mu\eta_{j+1}^t)\eta_0^t-\frac{8L}{P}\mathbb{E}\left[\|\hat{\boldsymbol{w}}^t-\boldsymbol{w}^\star\|\right]\eta_{N^t}^t$$

$$+\frac{16L\sigma^2}{P}\sum_{k=0}^{N^t-1}(\eta_{k+1}^t)^3\prod_{j=k+1}^{N^t-1}(1-\mu\eta_{j+1}^t). \tag{S25}$$

*We now use Equation (S8). It leads to the following,*

$$\frac{8L}{P}\sum_{k=0}^{N^t-1}\mathbb{E}\left[\|\boldsymbol{w}_{1,k}^t-\boldsymbol{w}^\star\|^2\right]\prod_{j=k+1}^{N^t-1}(1-\mu\eta_{j+1})\big(\eta_{k+1}^t-\eta_k^t(1-\mu\eta_{k+1}^t)\big)$$

$$\leq\frac{8L}{P}\prod_{j=0}^{N^t-1}(1-\mu\eta_{j+1})\mathbb{E}\left[\|\hat{\boldsymbol{w}}^{t-1}-\boldsymbol{w}^\star\|^2\right]\sum_{k=0}^{N^t-1}\big(\eta_{k+1}^t-\eta_k^t(1-\mu\eta_{k+1}^t)\big)$$

$$+\frac{8L}{P}\sum_{k=0}^{N^t-1}\big(2\sigma^2\sum_{l=1}^k(\eta_l^t)^2\prod_{m=l+1}^k(1-\mu\eta_m^t)\big)\prod_{j=k+1}^{N^t-1}(1-\mu\eta_{j+1})\big(\eta_{k+1}^t-\eta_k^t(1-\mu\eta_{k+1}^t)\big)$$

$$\leq\frac{8L}{P}\prod_{j=0}^{N^t-1}(1-\mu\eta_{j+1})\mathbb{E}\left[\|\hat{\boldsymbol{w}}^{t-1}-\boldsymbol{w}^\star\|^2\right]\sum_{k=0}^{N^t-1}\big(\eta_{k+1}^t-\eta_k^t(1-\mu\eta_{k+1}^t)\big)$$

$$+\frac{16\sigma^2L}{P}\sum_{k=0}^{N^t-1}\sum_{l=1}^k(\eta_l^t)^2\prod_{j=l+1}^{N^t-1}(1-\mu\eta_{j+1})\big(\eta_{k+1}^t-\eta_k^t(1-\mu\eta_{k+1}^t)\big)$$

$$\leq\frac{8L}{P}\prod_{j=0}^{N^t-1}(1-\mu\eta_{j+1})\mathbb{E}\left[\|\hat{\boldsymbol{w}}^{t-1}-\boldsymbol{w}^\star\|^2\right]\left(\eta_{N^t-1}^t-\eta_0^t+\sum_{k=0}^{N^t-1}\mu(\eta_k^t)^2\right)$$

$$+\frac{16\sigma^2L}{P}\sum_{l=1}^{N^t-1}\sum_{k=l}^{N^t-1}(\eta_l^t)^2\prod_{j=l+1}^{N^t-1}(1-\mu\eta_{j+1})\big(\eta_{k+1}^t-\eta_k^t(1-\mu\eta_{k+1}^t)\big)$$

$$\leq\frac{8L}{P}\prod_{j=0}^{N^t-1}(1-\mu\eta_{j+1})\mathbb{E}\left[\|\hat{\boldsymbol{w}}^{t-1}-\boldsymbol{w}^\star\|^2\right]\left(\eta_{N^t-1}^t-\eta_0^t+\sum_{k=0}^{N^t-1}\mu(\eta_k^t)^2\right)$$

$$+\frac{16\sigma^2L}{P}\sum_{l=1}^{N^t-1}(\eta_l^t)^2\prod_{j=l+1}^{N^t-1}(1-\mu\eta_{j+1})\sum_{k=0}^{N^t-1}\big(\eta_{k+1}^t-\eta_k^t(1-\mu\eta_{k+1}^t)\big)$$

$$\leq \frac{8L}{P} \prod_{j=0}^{N^t-1}(1-\mu\eta_{j+1})\mathbb{E}\left[\left\|\hat{\boldsymbol{w}}^{t-1}-\boldsymbol{w}^\star\right\|^2\right]\left(\eta_{N^t}^t-\eta_0^t+\sum_{k=0}^{N^t-1}\mu(\eta_{k+1}^t)^2\right)$$

$$+\frac{16\sigma^2 L}{P}\sum_{k=0}^{N^t-1}(\eta_{k+1}^t)^2\prod_{j=k+1}^{N^t-1}(1-\mu\eta_{j+1})\left(\eta_{N^t}^t-\eta_0^t+\sum_{k=0}^{N^t-1}\mu(\eta_{k+1}^t)^2\right). \tag{S26}$$

*Combining Equations* (S24) *to* (S26)*, we get, denoting* $C_{N^t}=\eta_{N^t}^t+\sum_{k=0}^{N^t-1}\mu(\eta_{k+1}^t)^2$:

$$\mathbb{E}\left[\left\|\hat{\boldsymbol{w}}^t-\boldsymbol{w}^\star\right\|^2\right]\leq\prod_{k=0}^{N^t-1}(1-\mu\eta_{k+1})\mathbb{E}\left[\left\|\hat{\boldsymbol{w}}^{t-1}-\boldsymbol{w}^\star\right\|^2\right]+2\frac{\sigma^2}{P}\sum_{k=0}^{N^t-1}(\eta_{k+1}^t)^2\prod_{j=k+1}^{N^t-1}(1-\mu\eta_{j+1}^t)$$

$$+\frac{8L}{P}\prod_{j=0}^{N^t-1}(1-\mu\eta_{j+1})\mathbb{E}\left[\left\|\hat{\boldsymbol{w}}^{t-1}-\boldsymbol{w}^\star\right\|^2\right](C_{N^t}-\eta_0^t)-\frac{8L}{P}\mathbb{E}\left[\left\|\hat{\boldsymbol{w}}^t-\boldsymbol{w}^\star\right\|\right]\eta_{N^t}^t$$

$$+\frac{16\sigma^2 L}{P}\sum_{k=0}^{N^t-1}(\eta_{k+1}^t)^2\prod_{j=k+1}^{N^t-1}(1-\mu\eta_{j+1})(C_{N^t}-\eta_0^t)$$

$$+\frac{8L}{P}\mathbb{E}\left[\left\|\hat{\boldsymbol{w}}^{t-1}-\boldsymbol{w}^\star\right\|^2\right]\prod_{j=0}^{N^t-1}(1-\mu\eta_{j+1}^t)\eta_0^t+\frac{16L\sigma^2}{P}\sum_{k=0}^{N^t-1}(\eta_{k+1}^t)^3\prod_{j=k+1}^{N^t-1}(1-\mu\eta_{j+1}^t).$$

*Thus, simplifying:*

$$\left(1+\frac{8L}{P}\eta_{N^t}^t\right)\mathbb{E}\left[\left\|\hat{\boldsymbol{w}}^t-\boldsymbol{w}^\star\right\|^2\right]$$

$$\leq\left(1+\frac{8L}{P}\eta_{N^t}^t+\sum_{k=0}^{N^t-1}\mu(\eta_{k+1}^t)^2\right)\prod_{k=0}^{N^t-1}(1-\mu\eta_{k+1})\mathbb{E}\left[\left\|\hat{\boldsymbol{w}}^{t-1}-\boldsymbol{w}^\star\right\|^2\right]$$

$$+2\frac{\sigma^2}{P}\sum_{k=0}^{N^t-1}(\eta_{k+1}^t)^2\left(1+\frac{8L}{P}\eta_{N^t}^t+\sum_{k=0}^{N^t-1}\mu(\eta_{k+1}^t)^2+L\eta_{k+1}^t\right)\prod_{j=k+1}^{N^t-1}(1-\mu\eta_{j+1}^t).$$

*This concludes the proof of the Lemma, using* $L\eta_k^t\leq 1/2$:

$$\mathbb{E}\left[\left\|\hat{\boldsymbol{w}}^t-\boldsymbol{w}^\star\right\|^2\right]\leq\left(1+\mu\sum_{k=1}^{N^t}(\eta_k^t)^2\right)\prod_{k=1}^{N^t}(1-\mu\eta_k)\mathbb{E}\left[\left\|\hat{\boldsymbol{w}}^{t-1}-\boldsymbol{w}^\star\right\|^2\right]$$

$$+2\left(4+\mu\sum_{k=1}^{N^t}(\eta_k^t)^2\right)\frac{\sigma^2}{P}\sum_{k=0}^{N^t}(\eta_k^t)^2\prod_{j=k+1}^{N^t}(1-\mu\eta_j^t).$$

This result can be used recursively. It implies that if $\mu\sum_{t=1}^{C}\sum_{k=1}^{N^t}(\eta_k^t)^2\leq K$, then the upper bound on the outer iterates is as good as the one for mini-batch, up to a constant.

### D.4 Proof of Proposition S20

In this Section we prove the first upper bound of Corollary S21.

#### D.4.1 Statement of Proposition S20

Finally, we provide a convergence result in the most general case, removing the quadratic assumption.

**Proposition S20 (Local-SGD: General Functions)** *Under Assumptions A1, A2, A3, A4 we have:*

$$\mathbb{E}\left[\left\|\boldsymbol{w}_{p,k}^t-\boldsymbol{w}^\star\right\|^2\right]\leq\kappa_2\prod_{k=1}^{\sum_{t'=1}^{t}N^{t'}+k}(1-\mu\tilde{\eta}_k)\left\|\boldsymbol{w}_0-\boldsymbol{w}^\star\right\|^2+2\frac{\sigma^2}{P}\sum_{u=\sum_{t'=1}^{t}N^{t'}}^{\sum_{t'=1}^{t}N^{t'}+k}\tilde{\eta}_u^2\prod_{j=u+1}^{\sum_{t'=1}^{t}N^{t'}+k}(1-\mu\tilde{\eta}_j)$$

$$+ (\sup_{t'=1...t} C_{P,M,K,t'}) \frac{\sigma^2}{P} \sum_{u=1}^{\sum_{t'=1}^{t} N^{t'}} \tilde{\eta}_u^2 \prod_{j=k+1}^{\sum_{t'=1}^{t} N^{t'}+k} (1 - \mu\tilde{\eta}_j),$$

with $C_{P,M,K,t} = 1 + MP \sum_{k=1}^{K} \eta_k^t \left\| \breve{\boldsymbol{w}}_{k-1}^t - \boldsymbol{w}^\star \right\|$.

**Interpretation:**   if $(\sup_{t'=1...t} C_{P,M,K,t})$ is uniformly bounded, we perform as well as minibatch SGD for the outer iterations (up to a constant).

For a constant step size $\eta$, the proposition has the following corollary:

**Corollary S21 (Local-SGD: General Functions)** *Under Assumptions A1, A2, A3, A4 we have:*

$$\mathbb{E}\left[ \left\| \boldsymbol{w}_{p,k}^t - \boldsymbol{w}^\star \right\|^2 \right] \leq \tau_2^t (1 - \eta\mu)^{\boldsymbol{N}_1^{t-1}+k} \left\| \boldsymbol{w}_0 - \boldsymbol{w}^\star \right\|^2$$

$$+ \sigma_\infty^2 \left( \left( \sup_{t'=1...t} C_{P,M,t'} \right) \frac{1 - (1-\eta\mu)^{\boldsymbol{N}_1^{t-1}}}{P\mu} + 2\frac{1 - (1-\eta\mu)^k}{\mu} \right).$$

Where $C_{P,M,t} = 1 + MP\eta \sum_{k=1}^{N^t} \mathbb{E}\left[ \left\| \breve{\boldsymbol{w}}_{k-1}^t - \boldsymbol{w}^\star \right\| \right]$. We prove the on-line case of the result using Lemma S22 in supplementary material.

**Interpretation.**   When communication occurs, averaging the different models over the machines results in a variance reduction, but at each phase, the variance accumulated within the phase is degraded with respect to the simplest setting by at most $C_{P,M,t}$. This constant increases with the number of machines and the step size, and also depends on the mean distance $\sum_{k=1}^{N^t} \mathbb{E}\left[ \left\| \breve{\boldsymbol{w}}_{k-1}^t - \boldsymbol{w}^\star \right\| \right]$ during phase $t$. As a consequence if $C_{P,M,t}$ is uniformly bounded, we perform as well as minibatch SGD. If $\mathbb{E}\left[ \left\| \breve{\boldsymbol{w}}_{k-1}^t - \boldsymbol{w}^\star \right\| \right]$ is assumed to be decaying, this is true if for any $t \in [T]$, $N^t \eta MP \mathbb{E}\left[ \left\| \hat{\boldsymbol{w}}^t - \boldsymbol{w}^\star \right\| \right] \leq O(1)$.

In the following, we alternatively relax the bounded variance assumption A4 and the quadratic assumption Q1, and show similar results for local SGD. This allows us to successively cover the cases of least squares regression (LSR) and logistic regression (LR).

### D.4.2   Proof

Proposition S20 follows from Lemma S22. We have for any $t \in [C], K \in [N^t]$,

$$\mathbb{E} \left\| \breve{\boldsymbol{w}}_K^t - \boldsymbol{w}^\star \right\|^2 \leq \prod_{k=1}^{K} (1 - \mu\eta_k^t) \mathbb{E} \left\| \breve{\boldsymbol{w}}_0^t - \boldsymbol{w}^\star \right\|^2 + C_{P,M,K,t} \frac{\sigma_\infty^2}{P} \sum_{k=1}^{K} (\eta_k^t)^2 \prod_{j=k+1}^{K} (1 - \mu\eta_j^t),$$

with $C_{P,M,K,t} = 1 + MP \sum_{k=1}^{K} \eta_k^t \left\| \breve{\boldsymbol{w}}_{k-1}^t - \boldsymbol{w}^\star \right\|$.

As in the two previous sections, we first focus on upper bounding $\mathbb{E}\left[ \left\| \breve{\boldsymbol{w}}_k^t - \boldsymbol{w}^\star \right\|^2 \right]$. We prove the following Lemma:

**Lemma S22** *For any $t \in [C], K \in [N^t]$, under Assumptions A1, A2, A3, A4 we have:*

$$\mathbb{E} \left\| \breve{\boldsymbol{w}}_K^t - \boldsymbol{w}^\star \right\|^2 \leq \prod_{k=1}^{K} (1 - \mu\eta_k^t) \mathbb{E} \left\| \breve{\boldsymbol{w}}_0^t - \boldsymbol{w}^\star \right\|^2 + C_{P,M,K,t} \frac{\sigma_\infty^2}{P} \sum_{k=1}^{K} (\eta_k^t)^2 \prod_{j=k+1}^{K} (1 - \mu\eta_j^t),$$

with $C_{P,M,K,t} = 1 + MP \sum_{k=1}^{K} \eta_k^t \mathbb{E}\left[ \left\| \breve{\boldsymbol{w}}_{k-1}^t - \boldsymbol{w}^\star \right\| \right]$.

This means, if we have consider an weak upper bound on $\mathbb{E}\left[ \left\| \breve{\boldsymbol{w}}_k^t - \boldsymbol{w}^\star \right\| \right] \leq R$ that the inner loops keeps the same variance as the mini-batch case if $MP \sum_{k=1}^{K} \eta_k^t = O(1)$. For example, for a constant step size $\eta$, it results in $PN^t\eta \leq 1$, *i.e.* $N^t \leq \frac{1}{P\eta}$. Note that the number of inner steps one can make increases with the phases, as $\mathbb{E}\left[ \left\| \hat{w}^t - \boldsymbol{w}^\star \right\| \right]$ decreases.

### D.4.3 Proof of Lemma S22

We rely on the following decomposition. Almost surely, we have:

$$\mathbb{E}\left[\left\|\breve{\boldsymbol{w}}_{k+1}^t - \boldsymbol{w}^\star\right\|^2 |\mathcal{H}_t^k\right] \leq \left\|\breve{\boldsymbol{w}}_k^t - \boldsymbol{w}^\star\right\|^2 - 2\eta_{k+1}^t\langle\breve{\boldsymbol{w}}_k^t - \boldsymbol{w}^\star, F'(\breve{\boldsymbol{w}}_k^t)\rangle$$

$$+ (\eta_{k+1}^t)^2\mathbb{E}\left[\left\|\frac{1}{P}\sum_{i=1}^P g_{i,k+1}^t(\boldsymbol{w}_{i,k}^t)\right\|^2 |\mathcal{H}_{k,t}\right]$$

$$+ 2\eta_{k+1}^t\langle\breve{\boldsymbol{w}}_k^t - \boldsymbol{w}^\star, F'(\breve{\boldsymbol{w}}_k^t) - \frac{1}{P}\sum_{p=1}^P F'(\boldsymbol{w}_{p,k}^t)\rangle. \tag{S27}$$

The first two lines correspond to the quadratic case (Equation (S13)), that has been analyzed in Lemma S18. The third term accounts for the difference between the mean gradient and the gradient at the mean point. We use Assumption A2 to control this term.

We then use the following Lemma, which control how the inner iterates $\boldsymbol{w}_{p,k}^t$ deviate from their average $\breve{\boldsymbol{w}}_k^t$:

**Lemma S23** *For any $t \in [C], k \in [N^t]$, under Assumptions A1, A2, A3, A4 we have a.s.:*

$$\frac{1}{P}\sum_{p=1}^P \mathbb{E}\left[\left\|\boldsymbol{w}_{p,k}^t - \breve{\boldsymbol{w}}_k^t\right\|^2\right] \leq \sigma_\infty^2\sum_{j=1}^k(\eta_j^t)^2\prod_{s=j+1}^k(1 - \eta_s^t\mu).$$

The proof of this Lemma is postponed to Appendix D.4.4.

Using Cauchy-Schwarz inequality and the bound on the third order derivative of $F$, we have:

$$2\eta_{k+1}^t\langle\breve{\boldsymbol{w}}_k^t - \boldsymbol{w}^\star, F'(\breve{\boldsymbol{w}}_k^t) - \frac{1}{P}\sum_{p=1}^P F'(\boldsymbol{w}_{p,k}^t)\rangle \leq 2\eta_{k+1}^t\left\|\breve{\boldsymbol{w}}_k^t - \boldsymbol{w}^\star\right\|\left\|F'(\breve{\boldsymbol{w}}_k^t) - \frac{1}{P}\sum_{p=1}^P F'(\boldsymbol{w}_{p,k}^t)\right\|, \tag{S28}$$

and, using a second order expansion of the gradient at $\breve{\boldsymbol{w}}_k^t$ together with Assumption A2 we have:

$$\left\|F'(\breve{\boldsymbol{w}}_k^t) - \frac{1}{P}\sum_{p=1}^P F'(\boldsymbol{w}_{p,k}^t)\right\| \leq \frac{M}{P}\sum_{p=1}^P\left\|\boldsymbol{w}_{p,k}^t - \breve{\boldsymbol{w}}_k^t\right\|^2. \tag{S29}$$

Using the proof of Equation (S15), and combining Equations (S27) to (S29) and Lemma S23, we have, for any $t \in [C], k \in [N^t]$:

$$\triangle := \mathbb{E}\left[\left\|\breve{\boldsymbol{w}}_{k+1}^t - \boldsymbol{w}^\star\right\|^2 |\mathcal{H}_t^k\right]$$

$$\triangle \leq \left\|\breve{\boldsymbol{w}}_k^t - \boldsymbol{w}^\star\right\|^2 - 2\eta_{k+1}^t\langle\breve{\boldsymbol{w}}_k^t - \boldsymbol{w}^\star, F'(\breve{\boldsymbol{w}}_k^t)\rangle + (\eta_{k+1}^t)^2\mathbb{E}\left[\left\|\frac{1}{P}\sum_{i=1}^P g_{i,k+1}^t(\boldsymbol{w}_{i,k}^t)\right\|^2 |\mathcal{H}_{k,t}\right]$$

$$+ 2\eta_{k+1}^t\langle\breve{\boldsymbol{w}}_k^t - \boldsymbol{w}^\star, F'(\breve{\boldsymbol{w}}_k^t) - \frac{1}{P}\sum_{p=1}^P F'(\boldsymbol{w}_{p,k}^t)\rangle$$

$$\mathbb{E}[\triangle] \leq (1 - \mu\eta_{k+1}^t)\mathbb{E}\left[\left\|\breve{\boldsymbol{w}}_k^t - \boldsymbol{w}^\star\right\|^2\right] + (\eta_{k+1}^t)^2\frac{1}{P}\sigma_\infty^2$$

$$+ 2\eta_{k+1}^t\mathbb{E}\left[\left\|\breve{\boldsymbol{w}}_k^t - \boldsymbol{w}^\star\right\|\right]M\sum_{j=1}^k(\eta_j^t)^2\sigma_\infty^2\prod_{s=j+1}^k(1 - \eta_s^t\mu). \tag{S30}$$

Thus by induction, for any $t \in [C], K \in [N^t]$:

$$\mathbb{E}\left[\left\|\breve{\boldsymbol{w}}_K^t - \boldsymbol{w}^\star\right\|^2\right] \leq \prod_{k=1}^K(1 - \mu\eta_k^t)\mathbb{E}\left[\left\|\breve{\boldsymbol{w}}_0^t - \boldsymbol{w}^\star\right\|^2\right] + \frac{1}{P}\sigma_\infty^2\sum_{k=1}^K(\eta_k^t)^2\prod_{j=k+1}^K(1 - \mu\eta_j^t)$$

$$+ 2\sigma_\infty^2 M \sum_{k=1}^{K} \eta_k^t \mathbb{E}\left[\left\|\breve{w}_{k-1}^t - w^\star\right\|\right] \sum_{j=1}^{k} (\eta_j^t)^2 \prod_{s=j+1}^{k} (1 - \eta_s^t \mu) \prod_{j=k+1}^{K} (1 - \mu \eta_j^t)$$

$$= \prod_{k=1}^{K} (1 - \mu \eta_k^t) \mathbb{E}\left[\left\|\breve{w}_0^t - w^\star\right\|^2\right] + \frac{1}{P} \sigma_\infty^2 \sum_{k=1}^{K} (\eta_k^t)^2 \prod_{j=k+1}^{K} (1 - \mu \eta_j^t)$$

$$+ 2M \sigma_\infty^2 \sum_{j=1}^{K} (\eta_j^t)^2 \prod_{s=j+1}^{K} (1 - \mu \eta_j^t) \sum_{k=j}^{K} \eta_k^t \mathbb{E}\left[\left\|\breve{w}_{k-1}^t - w^\star\right\|\right]$$

$$= \prod_{k=1}^{K} (1 - \mu \eta_k^t) \mathbb{E}\left[\left\|\breve{w}_0^t - w^\star\right\|^2\right] + C_{P,M,K,t} \frac{\sigma_\infty^2}{P} \sum_{k=1}^{K} (\eta_k^t)^2 \prod_{j=k+1}^{K} (1 - \mu \eta_j^t),$$

with $C_{P,M,K,t} = 1 + MP \sum_{k=1}^{K} \eta_k^t \mathbb{E}\left[\left\|\breve{w}_{k-1}^t - w^\star\right\|\right]$. This concludes the proof.
In the following section, we proved the auxiliary Lemma that was used in the proof.

### D.4.4 Proof of Lemma S23

We now study $\frac{1}{P} \sum_{p=1}^{P} \left\|w_{p,k}^t - \breve{w}_k^t\right\|^2$ as $k$ increases. Note that initially ($k = 0$), this quantity is 0.
For any $k \in [N^t], p \in [P]$:

$$\left\|w_{p,k}^t - \breve{w}_k^t\right\|^2 = \left\|w_{p,k-1}^t - \eta_k^t g_{p,k}^t(w_{p,k-1}^t) - \breve{w}_{k-1}^t + \eta_k^t \frac{1}{P} \sum_{i=1}^{P} g_{i,k}^t(w_{i,k-1}^t)\right\|^2$$

$$= \left\|w_{p,k-1}^t - \breve{w}_{k-1}^t\right\|^2 - 2\eta_k^t \left\langle w_{p,k-1}^t - \breve{w}_{k-1}^t, g_{p,k}^t(w_{p,k-1}^t) - \frac{1}{P} \sum_{i=1}^{P} g_{i,k}^t(w_{i,k-1}^t)\right\rangle$$

$$+ (\eta_k^t)^2 \left\|g_{p,k}^t(w_{p,k-1}^t) - \frac{1}{P} \sum_{i=1}^{P} g_{i,k}^t(w_{i,k-1}^t)\right\|^2.$$

Thus, expanding and using cocoercivity Assumption:

$$\mathbb{E}\left[\left\|w_{p,k}^t - \breve{w}_k^t\right\|^2 |\mathcal{H}_{k-1}^t\right] = \left\|w_{p,k-1}^t - \breve{w}_{k-1}^t\right\|^2$$

$$- 2\eta_k^t \left\langle w_{p,k-1}^t - \breve{w}_{k-1}^t, F'(w_{p,k-1}^t) - \frac{1}{P} \sum_{i=1}^{P} F'(w_{i,k-1}^t)\right\rangle$$

$$+ \mathbb{E}\left[(\eta_k^t)^2 \left\|g_{p,k}^t(w_{p,k-1}^t) - \frac{1}{P} \sum_{i=1}^{P} g_{i,k}^t(w_{i,k-1}^t)\right\|^2 |\mathcal{H}_{k-1}^t\right]$$

$$= \left\|w_{p,k-1}^t - \breve{w}_{k-1}^t\right\|^2 - 2\eta_k^t \left\langle w_{p,k-1}^t - \breve{w}_{k-1}^t, F'(w_{p,k-1}^t) - F'(\breve{w}_{k-1}^t)\right\rangle$$

$$+ 2\eta_k^t \left\langle w_{p,k-1}^t - \breve{w}_{k-1}^t, F'(\breve{w}_{k-1}^t) - \frac{1}{P} \sum_{i=1}^{P} F'(w_{i,k-1}^t)\right\rangle$$

$$+ \mathbb{E}\left[(\eta_k^t)^2 \left\|g_{p,k}^t(w_{p,k-1}^t) - \frac{1}{P} \sum_{i=1}^{P} g_{i,k}^t(w_{i,k-1}^t)\right\|^2 |\mathcal{H}_{k-1}^t\right]$$

$$\leq (1 - 2\eta_k^t \mu(1 - \eta_k^t L)) \left\|w_{p,k-1}^t - \breve{w}_{k-1}^t\right\|^2$$

$$+ 2\eta_k^t \left\langle w_{p,k-1}^t - \breve{w}_{k-1}^t, F'(\breve{w}_{k-1}^t) - \frac{1}{P} \sum_{i=1}^{P} F'(w_{i,k-1}^t)\right\rangle$$

$$+ \mathbb{E}\left[(\eta_k^t)^2 \left\|(g_{p,k}^t - F')(w_{p,k-1}^t) - \frac{1}{P} \sum_{i=1}^{P} (g_{i,k}^t - F')(w_{i,k-1}^t)\right\|^2 |\mathcal{H}_{k-1}^t\right].$$

Summing over $p \in [P]$:

$$\sum_{p=1}^{P} \mathbb{E}\left[\left\|\boldsymbol{w}_{p,k}^t - \breve{\boldsymbol{w}}_k^t\right\|^2 |\mathcal{H}_{k-1}^t\right] \leq (1 - \eta_k^t \mu) \sum_{p=1}^{P} \left\|\boldsymbol{w}_{p,k-1}^t - \breve{\boldsymbol{w}}_{k-1}^t\right\|^2$$

$$+ 2\eta_k^t \underbrace{\left\langle \sum_{p=1}^{P}(\boldsymbol{w}_{p,k-1}^t - \breve{\boldsymbol{w}}_{k-1}^t), F'(\breve{\boldsymbol{w}}_{k-1}^t) - \frac{1}{P}\sum_{i=1}^{P} F'(\boldsymbol{w}_{i,k-1}^t)\right\rangle}_{=0}$$

$$+ \sum_{p=1}^{P} \mathbb{E}\left[(\eta_k^t)^2 \left\|(g_{p,k}^t - F')(\boldsymbol{w}_{p,k-1}^t) - \frac{1}{P}\sum_{i=1}^{P}(g_{i,k}^t - F')(\boldsymbol{w}_{i,k-1}^t)\right\|^2 |\mathcal{H}_{k-1}^t\right].$$

If we denote $\delta_k^t = \frac{1}{P}\sum_{p=1}^{P} \mathbb{E}\left[\left\|\boldsymbol{w}_{p,k}^t - \breve{\boldsymbol{w}}_k^t\right\|^2\right]$, we thus have $\delta_0 = 0$ and

$$\delta_k^t \leq (1 - \eta_k^t \mu)\delta_{k-1}^t + \frac{1}{P}\sum_{p=1}^{P} \mathbb{E}\left[(\eta_k^t)^2 \left\|(g_{p,k}^t - F')(\boldsymbol{w}_{p,k-1}^t) - \frac{1}{P}\sum_{i=1}^{P}(g_{i,k}^t - F')(\boldsymbol{w}_{i,k-1}^t)\right\|^2 |\mathcal{H}_{k-1}^t\right].$$

$$\leq \frac{1}{P}\sum_{p=1}^{P}\sum_{j=1}^{k} \mathbb{E}\left[(\eta_j^t)^2 \left\|(g_{p,j}^t - F')(\boldsymbol{w}_{p,j-1}^t) - \frac{1}{P}\sum_{i=1}^{P}(g_{i,j}^t - F')(\boldsymbol{w}_{i,j-1}^t)\right\|^2\right] \prod_{s=j+1}^{k}(1 - \eta_s^t \mu)$$

$$\leq \sum_{j=1}^{k} \mathbb{E}\left[(\eta_j^t)^2 \left\|(g_{1,j}^t - F')(\boldsymbol{w}_{1,j-1}^t) - \frac{1}{P}\sum_{i=1}^{P}(g_{i,j}^t - F')(\boldsymbol{w}_{i,j-1}^t)\right\|^2\right] \prod_{s=j+1}^{k}(1 - \eta_s^t \mu)$$

$$\leq \sum_{j=1}^{k} \mathbb{E}\left[(\eta_j^t)^2 \left\|(g_{1,j}^t - F')(\boldsymbol{w}_{1,j-1}^t)\right\|^2\right] \prod_{s=j+1}^{k}(1 - \eta_s^t \mu).$$

Note that everything is tight until the last line for $P = 1$ ( then for all $k$, $\delta_k^t = 0$). Under Assumption **A**4, we thus have:

$$\delta_k^t \leq \sum_{j=1}^{k}(\eta_j^t)^2 \sigma_\infty^2 \prod_{s=j+1}^{k}(1 - \eta_s^t \mu).$$

This concludes the proof.

## E    Convergence guaranties for the fourth order moment.

In this section, we prove several Lemmas that allow to control the fourth order moment of the iterate. While controlling the second order moment is sufficient for quadratic functions as no "residual" term appears in Equation (3) (the "residual" corresponds to the rest of a linear expansion of the gradient, which is thus exact for a quadratic function), in the general case, we also need to control the 4th order moment.

We first give guarantees for the inner iterates (within a phase) in Appendix E.1, then in the local SGD framework in Appendix E.2.

### E.1    Inner Iteration Lemmas

Here, we can use the following Lemma from [25], that gives a recursion for the 4th order moment.

**Lemma S24** *Under the Assumptions* **A**1, **A**2, **A**3, **A**5 *for th* $4^{th}$*-order moment, assuming* $\eta \leq \frac{1}{18L}$ *we have,*

$$\mathbb{E}\left[(\|\boldsymbol{w}_{i,k}^t - \boldsymbol{w}^\star\|)^4\right]^{1/2} \leq (1 - \eta\mu)\mathbb{E}\left[\|\boldsymbol{w}_{i,k-1}^t - \boldsymbol{w}^\star\|^4\right]^{1/2} + 20\eta^2\sigma^2$$

$$\mathbb{E}\left[\left\|\boldsymbol{w}_{i,k}^t - \boldsymbol{w}^\star\right\|^4\right]^{1/2} \leq (1-\eta\mu)^k \mathbb{E}\left[\left\|\boldsymbol{w}_{i,0}^t - \boldsymbol{w}^\star\right\|^4\right]^{1/2} + \frac{20\eta\sigma^2}{\mu}.$$

In the mini-batch setting, we have of course the same result with a variance reduction:

**Lemma S25** *Under the Assumptions A1, A2, A3, A5 for th $4^{th}$-order moment for mini-batch averaging we have, assuming $\eta \leq \frac{1}{18L}$ we have,*

$$\mathbb{E}\left[\left\|\hat{\boldsymbol{w}}^t - \boldsymbol{w}^\star\right\|^4\right]^{1/2} \leq \left(1-\eta\mu\right)\mathbb{E}\left[\left\|\hat{\boldsymbol{w}}^{t-1} - \boldsymbol{w}^\star\right\|^4\right]^{1/2} + \frac{20\eta^2}{P}\sigma^2$$

$$\mathbb{E}\left[\left\|\hat{\boldsymbol{w}}^t - \boldsymbol{w}^\star\right\|^4\right]^{1/2} \leq \left(1-\eta\mu\right)^t \left\|\boldsymbol{w}^0 - \boldsymbol{w}^\star\right\|^2 + \frac{20\eta}{P\mu}\sigma^2.$$

Analogous to Lemma S24 we have the following result for fourth order moments,

**Lemma S26** *Under the Assumptions A1, A2, A3, A5 for th $4^{th}$-order moment, assuming $\eta \leq \frac{1}{18L}$ we have,*

$$\mathbb{E}\left[\left\|\boldsymbol{w}_{i,k}^t - \boldsymbol{w}^\star\right\|^4\right]^{1/2} \leq \left(1-\eta_k^t\mu\right)\mathbb{E}\left[\left\|\boldsymbol{w}_{i,k-1}^t - \boldsymbol{w}^\star\right\|^4\right]^{1/2} + 20\eta^2\sigma^2$$

$$\mathbb{E}\left[\left\|\boldsymbol{w}_{i,k}^t - \boldsymbol{w}^\star\right\|^4\right]^{1/2} \leq \prod_{j=1}^{k}(1-\eta_j^t\mu)\left\|\boldsymbol{w}^0 - \boldsymbol{w}^\star\right\|^2 + 20\sigma^2\sum_{j=1}^{k}\prod_{l=j+1}^{k}(1-\mu\eta_l^t)(\eta_j^t)^2.$$

Similarly for mini-batch analogous to Lemma S25,

**Lemma S27** *Under the Assumptions A1, A2, A3, A5 for th $4^{th}$-order moment for mini-batch averaging and decreasing step size we have, assuming $\eta \leq \frac{1}{18L}$ we have,*

$$\mathbb{E}\left[\left\|\hat{\boldsymbol{w}}^t - \boldsymbol{w}^\star\right\|^4\right]^{1/2} \leq \left(1-\eta^t\mu\right)\mathbb{E}\left[\left\|\hat{\boldsymbol{w}}^{t-1} - \boldsymbol{w}^\star\right\|^4\right]^{1/2} + \frac{20\eta^2}{P}\sigma^2$$

$$\mathbb{E}\left[\left\|\hat{\boldsymbol{w}}^t - \boldsymbol{w}^\star\right\|^4\right]^{1/2} \leq \prod_{j=1}^{t}\left(1-\eta^j\mu\right)\left\|\hat{\boldsymbol{w}}^0 - \boldsymbol{w}^\star\right\|^2 + \frac{20\sigma^2}{P}\sum_{j=1}^{t}\prod_{l=j+1}^{t}(1-\mu\eta^l)(\eta^j)^2.$$

The proof is included for completeness and because the same proof technique is used afterwards in Appendix E.2.

**Proof 28** *For $i \in [P]$, $k \in [N_t]$ and $t \in [C]$ we define the notation $\phi_{i,k}^t = \left\|\boldsymbol{w}_{i,k}^t - \boldsymbol{w}^\star\right\|$. We have that,*

$$
\begin{aligned}
(\phi_{i,k}^t)^4 &= \left(\left\|\boldsymbol{w}_{i,k-1}^t - \boldsymbol{w}^\star\right\|^2 - 2\eta\langle\boldsymbol{g}_{i,k}^t(\boldsymbol{w}_{i,k-1}^t), \boldsymbol{w}_{i,k-1}^t - \boldsymbol{w}^\star\rangle + \eta^2\left\|\boldsymbol{g}_{i,k}^t(\boldsymbol{w}_{i,k-1}^t)\right\|^2\right)^2 \\
&= \left((\phi_{i,k-1}^t)^2 - 2\eta\langle\boldsymbol{g}_{i,k}^t(\boldsymbol{w}_{i,k-1}^t), \boldsymbol{w}_{i,k-1}^t - \boldsymbol{w}^\star\rangle + \eta^2\left\|g_{i,k}^t(\boldsymbol{w}_{i,k-1}^t)\right\|^2\right)^2 \\
&= (\phi_{i,k-1}^t)^4 - 4\eta(\phi_{i,k-1}^t)^2\langle\boldsymbol{g}_{i,k}^t(\boldsymbol{w}_{i,k-1}^t), \boldsymbol{w}_{i,k-1}^t - \boldsymbol{w}^\star\rangle \\
&\quad + 4\eta^2\langle g_{i,k}^t(\boldsymbol{w}_{i,k-1}^t), \boldsymbol{w}_{i,k-1}^t - \boldsymbol{w}^\star\rangle^2 + 2\eta^2(\phi_{i,k-1}^t)^2\left\|\boldsymbol{g}_{i,k}^t(\boldsymbol{w}_{i,k-1}^t)\right\|^2 \\
&\quad - 4\eta^3\langle\boldsymbol{g}_{i,k}^t(\boldsymbol{w}_{i,k-1}^t), \boldsymbol{w}_{i,k-1}^t - \boldsymbol{w}^\star\rangle\left\|\boldsymbol{g}_{i,k}^t(\boldsymbol{w}_{i,k-1}^t)\right\|^2 + \eta^4\left\|\boldsymbol{g}_{i,k}^t(\boldsymbol{w}_{i,k-1}^t)\right\|^4.
\end{aligned}
$$

*Moreover,*

$$
\begin{aligned}
\mathbb{E}\left[\left\|\boldsymbol{g}_{i,k}^t(\boldsymbol{w}_{i,k-1}^t)\right\|^p |\mathbb{H}_{k-1}^t\right] &\leq 2^{p-1}\left(\mathbb{E}\left[\left\|\boldsymbol{g}_{i,k}^t(\boldsymbol{w}_{i,k-1}^t) - \boldsymbol{g}_{i,k}^t(\boldsymbol{w}^\star)\right\|^p |\mathbb{H}_{k-1}^t\right] + \mathbb{E}\left[\left\|\boldsymbol{g}_{i,k}^t(\boldsymbol{w}^\star)\right\|^p |\mathbb{H}_{k-1}^t\right]\right) \\
&\leq 2^{p-1}\left(\mathbb{E}\left[\left\|\boldsymbol{g}_{i,k}^t(\boldsymbol{w}_{i,k-1}^t) - \boldsymbol{g}_{i,k}^t(\boldsymbol{w}^\star)\right\|^p\right] + \mathbb{E}\left[\left\|\boldsymbol{g}_{i,k}^t(\boldsymbol{w}^\star)\right\|^p |\mathbb{H}_{k-1}^t\right]\right) \\
&\leq 2^{p-1}\left(\left\|\boldsymbol{g}_{i,k}^t(\boldsymbol{w}_{i,k-1}^t) - \boldsymbol{g}_{i,k}^t(\boldsymbol{w}^\star)\right\|^p + \sigma^p\right), \tag{S31}
\end{aligned}
$$

*Where we have used at the first line Minkowski's inequality and the fact that $x \mapsto x^p$ is convex on $\mathbb{R}^+$ for $p = 1, \ldots, 4$ thus $(x+y)^p \leq 2^{p-1}(x^p + y^p)$, and at the last line the Assumption A5 on the noise : $\mathbb{E}\left[\left\|f_{i,k}^t(\boldsymbol{w}^\star)\right\|^p |\mathbb{H}_{k-1}^t\right] \leq \sigma^p$.*

*This leads to*

$$\blacktriangle := \mathbb{E}\left[(\phi_{i,k}^t)^4|\mathbb{H}_{k-1}^t\right]$$

$$\leq (\phi_{i,k-1}^t)^4 - 4\eta(\phi_{i,k-1}^t)^2\mathbb{E}\left[\langle \boldsymbol{g}_{i,k}^t(\boldsymbol{w}_{i,k-1}^t), \boldsymbol{w}_{i,k-1}^t - \boldsymbol{w}^\star\rangle|\mathbb{H}_{k-1}^t\right]$$

$$+ 4\eta^2\mathbb{E}\left[\langle \boldsymbol{g}_{i,k}^t(\boldsymbol{w}_{i,k-1}^t), \boldsymbol{w}_{i,k-1}^t - \boldsymbol{w}^\star\rangle^2|\mathbb{H}_{k-1}^t\right] + 2\eta^2(\phi_{i,k-1}^t)^2\mathbb{E}\left[\left\|\boldsymbol{g}_{i,k}^t(\boldsymbol{w}_{i,k-1}^t)\right\|^2|\mathbb{H}_{k-1}^t\right]$$

$$- 4\eta^3\mathbb{E}\left[\langle \boldsymbol{g}_{i,k}^t(\boldsymbol{w}_{i,k-1}^t), \boldsymbol{w}_{i,k-1}^t - \boldsymbol{w}^\star\rangle \left\|\boldsymbol{g}_{i,k}^t(\boldsymbol{w}_{i,k-1}^t)\right\|^2|\mathbb{H}_{k-1}^t\right] + \eta^4\mathbb{E}\left[\left\|\boldsymbol{g}_{i,k}^t(\boldsymbol{w}_{i,k-1}^t)\right\|^4|\mathbb{H}_{k-1}^t\right]$$

$$\leq (\phi_{i,k-1}^t)^4 - 4\eta(\phi_{i,k-1}^t)^2\langle F'(\boldsymbol{w}_{i,k-1}^t), \boldsymbol{w}_{i,k-1}^t - \boldsymbol{w}^\star\rangle + 4\eta^2\mathbb{E}\left[\left\|\boldsymbol{g}_{i,k}^t(\boldsymbol{w}_{i,k-1}^t)\right\|^2 (\phi_{i,k-1}^t)^2|\mathbb{H}_{k-1}^t\right]$$

$$+ 2\eta^2(\phi_{i,k-1}^t)^2\mathbb{E}\left[\left\|\boldsymbol{g}_{i,k}^t(\boldsymbol{w}_{i,k-1}^t)\right\|^2|\mathbb{H}_{k-1}^t\right] + 4\eta^3\phi_{i,k-1}^t\mathbb{E}\left[\left\|\boldsymbol{g}_{i,k}^t(\boldsymbol{w}_{i,k-1}^t)\right\|^3|\mathbb{H}_{k-1}^t\right]$$

$$+ \eta^4\mathbb{E}\left[\left\|\boldsymbol{g}_{i,k}^t(\boldsymbol{w}_{i,k-1}^t)\right\|^4|\mathbb{H}_{k-1}^t\right]$$

$$\leq (\phi_{i,k-1}^t)^4 - 4\eta(\phi_{i,k-1}^t)^2\langle F'(\boldsymbol{w}_{i,k-1}^t), \boldsymbol{w}_{i,k-1}^t - \boldsymbol{w}^\star\rangle + 12\eta^2\sigma^2(\phi_{i,k-1}^t)^2 + 16\eta^3\phi_{i,k-1}^t\sigma^3 + 8\eta^4\sigma^4$$

$$+ 12\eta^2(\phi_{i,k-1}^t)^2\mathbb{E}\left[\left\|\boldsymbol{g}_{i,k}^t(\boldsymbol{w}_{i,k-1}^t) - \boldsymbol{g}_{i,k}^t(\boldsymbol{w}^\star)\right\|^2|\mathbb{H}_{k-1}^t\right]$$

$$+ 16\eta^3\phi_{i,k-1}^t\mathbb{E}\left[\left\|\boldsymbol{g}_{i,k}^t(\boldsymbol{w}_{i,k-1}^t) - \boldsymbol{g}_{i,k}^t(\boldsymbol{w}^\star)\right\|^3|\mathbb{H}_{k-1}^t\right] + 8\eta^4\mathbb{E}\left[\left\|\boldsymbol{g}_{i,k}^t(\boldsymbol{w}_{i,k-1}^t) - \boldsymbol{g}_{i,k}^t(\boldsymbol{w}^\star)\right\|^4|\mathbb{H}_{k-1}^t\right].$$

*Above we have used Cauchy Schwartz inequality several times for the second inequality and equation* (S31) *for the third one.*

$$\bigstar := \mathbb{E}\left[(\phi_{i,k}^t)^4|\mathbb{H}_{k-1}^t\right]$$

$$\leq (\phi_{i,k-1}^t)^4 - 4\eta(\phi_{i,k-1}^t)^2\langle F'(\boldsymbol{w}_{i,k-1}^t), \boldsymbol{w}_{i,k-1}^t - \boldsymbol{w}^\star\rangle + 12\eta^2 L(\phi_{i,k-1}^t)^2\langle F'(\boldsymbol{w}_{i,k-1}^t), \boldsymbol{w}_{i,k-1}^t - \boldsymbol{w}^\star\rangle$$

$$+ 16\eta^3 L^2(\phi_{i,k-1}^t)^2\langle F'(\boldsymbol{w}_{i,k-1}^t), \boldsymbol{w}_{i,k-1}^t - \boldsymbol{w}^\star\rangle + 8\eta^4 L^3(\phi_{i,k-1}^t)^2\langle F'(\boldsymbol{w}_{i,k-1}^t), \boldsymbol{w}_{i,k-1}^t - \boldsymbol{w}^\star\rangle$$

$$+ 12\eta\sigma^2(\phi_{i,k-1}^t)^2 + 8\eta^2\sigma^2(\phi_{i,k-1}^t)^2 + 8\eta^4\sigma^4 + 8\eta^4\sigma^4$$

$$= (\phi_{i,k-1}^t)^4 + (-4\eta + 12\eta^2 L + 16\eta^3 L^2 + 8\eta^4 L^3)(\phi_{i,k-1}^t)^2\langle F'(\boldsymbol{w}_{i,k-1}^t), \boldsymbol{w}_{i,k-1}^t - \boldsymbol{w}^\star\rangle$$

$$+ (12\eta^2\sigma^2 + 8\eta^2\sigma^2)(\phi_{i,k-1}^t)^2 + 16\eta^4\sigma^4$$

$$\leq (\phi_{i,k-1}^t)^4 - 4\eta(1 - 9\eta L)(\phi_{i,k-1}^t)^2\langle F'(\boldsymbol{w}_{i,k-1}^t), \boldsymbol{w}_{i,k-1}^t - \boldsymbol{w}^\star\rangle + 20\eta^2\sigma^2(\phi_{i,k-1}^t)^2 + 16\eta^4\sigma^4.$$

*Above we used $\eta L \leq 1$ in the last line. Finally, using strong convexity, we have:*

$$\mathbb{E}\left[(\phi_{i,k}^t)^4|\mathbb{H}_{k-1}^t\right] \leq \left(1 - 4\eta\mu(1 - 9\eta L)\right)(\phi_{i,k-1}^t)^4 + 20\eta^2\sigma^2(\phi_{i,k-1}^t)^2 + 16\eta^4\sigma^4,$$

*Now $\mathbb{E}\left[(\phi_{i,k-1}^t)^2\right] \leq \mathbb{E}\left[(\phi_{i,k-1}^t)^4\right]^{1/2}$ by Jensen's inequality. Also since we assume $\eta \leq \frac{1}{9L}$ and $\frac{\mu}{L} \leq 1$ we can obtain $(1 - 4\eta\mu(1 - 9\eta L))^{1/2} \geq (1 - 4\eta\mu)^{1/2} \geq (1 - \frac{4\mu}{9L})^{1/2} \geq (1 - \frac{4}{9})^{1/2} \geq 1/2$. This finally leads to $20\eta^2\sigma^2\mathbb{E}\left[(\phi_{i,k-1}^t)^2\right] \leq (1 - 4\eta\mu(1 - 9\eta L))^{1/2}\mathbb{E}\left[(\phi_{i,k-1}^t)^4\right]^{1/2} 40\eta^2\sigma^2$, which can be used below to obtain*

$$\mathbb{E}\left[(\phi_{i,k}^t)^4|\mathbb{H}_{k-1}^t\right] \leq \left(1 - 4\eta\mu(1 - 9\eta L)\right)\mathbb{E}\left[(\phi_{i,k-1}^t)^4\right] + 20\eta^2\sigma^2\mathbb{E}\left[(\phi_{i,k-1}^t)^2\right] + 16\eta^4\sigma^4$$

$$\leq \left(\left(1 - 4\eta\mu(1 - 9\eta L)\right)^{1/2}\mathbb{E}\left[(\phi_{i,k-1}^t)^4\right]^{1/2} + 20\eta^2\sigma^2\right)^2$$

$$\mathbb{E}\left[(\phi_{i,k}^t)^4\right]^{1/2} \leq \left(1 - 2\eta\mu(1 - 9\eta L)\right)\mathbb{E}\left[(\phi_{i,k-1}^t)^4\right]^{1/2} + 20\eta^2\sigma^2.$$

*This Concludes the proof.*

## E.2 Proof of Lemma S29

In this section, we prove the following Lemma, which is necessary to conclude the proof for the second set of Assumptions in Theorem 6. Indeed, we need to control the moment of order 4 to be able to control the residual term that arises from linear expansion of the gradient around $\boldsymbol{w}^\star$.

**Lemma S29** *There exist absolute constants $C_4, D_4, E_4$, such that if $\eta_k^t L \leq \frac{1}{C_4}$:*

$$\mathbb{E}\left[\left\|\breve{\boldsymbol{w}}_{k+1}^t - \boldsymbol{w}^\star\right\|^4\right]^{1/2} \leq (1 - \eta_k^t\mu)\mathbb{E}\left[\left\|\breve{\boldsymbol{w}}_k^t - \boldsymbol{w}^\star\right\|^4\right]^{1/2} + D_4(\eta_k^t)^2\frac{\sigma_\infty^2}{P}$$

$$+ E_4 \eta_{k+1}^t \left\| \breve{w}_k^t - w^\star \right\| \left\| F'(\breve{w}_k^t) - \frac{1}{P} \sum_{p=1}^P F'(w_{p,k}^t) \right\|. \tag{S32}$$

In other words, $\mathbb{E}\left[ \left\| \breve{w}_{k+1}^t - w^\star \right\|^4 \right]^{1/2}$ satisfies the same recursion as $\mathbb{E}\left[ \left\| \breve{w}_{k+1}^t - w^\star \right\|^2 \right]$, as this equation is the same as Equation (S30) (up to absolute constants).

**Proof 30** *This proof combines element from the classical bound for the fourth order moment, and from the proof of Lemma S22, which addresses the similar setting but only for the second order moment. We start from the definition of $\breve{w}_{k+1}^t$:*

$$\left\| \breve{w}_{k+1}^t - w^\star \right\|^2 \leq \left\| \breve{w}_k^t - w^\star \right\|^2 - 2\eta_{k+1}^t \langle \breve{w}_k^t - w^\star, \frac{1}{P} \sum_{i=1}^P g_{i,k+1}^t(\breve{w}_k^t) \rangle$$

$$+ (\eta_{k+1}^t)^2 \left\| \frac{1}{P} \sum_{i=1}^P g_{i,k+1}^t(w_{i,k}^t) \right\|^2$$

$$+ 2\eta_{k+1}^t \langle \breve{w}_k^t - w^\star, \frac{1}{P} \sum_{i=1}^P g_{i,k+1}^t(\breve{w}_k^t) - \frac{1}{P} \sum_{p=1}^P F'(w_{p,k}^t) \rangle. \tag{S33}$$

*Thus, squaring this equation we get, denoting $\breve{\phi}_k^t = \left\| \breve{w}_k^t - w^\star \right\|$:*

$$(\breve{\phi}_{k+1}^t)^4 \leq (\breve{\phi}_k^t)^4 - 4(\breve{\phi}_k^t)^2 \eta_{k+1}^t \langle \breve{w}_k^t - w^\star, \frac{1}{P} \sum_{i=1}^P g_{i,k+1}^t(\breve{w}_k^t) \rangle$$

$$+ 2(\breve{\phi}_k^t)^2 (\eta_{k+1}^t)^2 \left\| \frac{1}{P} \sum_{i=1}^P g_{i,k+1}^t(w_{i,k}^t) \right\|^2$$

$$+ 4(\breve{\phi}_k^t)^2 \eta_{k+1}^t \langle \breve{w}_k^t - w^\star, \frac{1}{P} \sum_{i=1}^P g_{i,k+1}^t(\breve{w}_k^t) - \frac{1}{P} \sum_{p=1}^P F'(w_{p,k}^t) \rangle$$

$$+ 3(\eta_{k+1}^t)^2 \langle \breve{w}_k^t - w^\star, \frac{1}{P} \sum_{i=1}^P g_{i,k+1}^t(\breve{w}_k^t) \rangle^2$$

$$+ 3(\eta_{k+1}^t)^4 \left\| \frac{1}{P} \sum_{i=1}^P g_{i,k+1}^t(w_{i,k}^t) \right\|^4$$

$$+ 3(2\eta_{k+1}^t)^2 \langle \breve{w}_k^t - w^\star, \frac{1}{P} \sum_{i=1}^P g_{i,k+1}^t(\breve{w}_k^t) - \frac{1}{P} \sum_{p=1}^P F'(w_{p,k}^t) \rangle^2,$$

*formally, we have used $(a + b + c + d)^2 \leq a^2 + 2ab + 2ac + 2ad + 3b^2 + 3c^2 + 3d^2$.*

*That is, conditioning on the past, and using Assumption A5 (cocoercivity and the fact that $g_k^t$ is a.s. $L$-Lipshitz):*

$$\mathbb{E}\left[ (\breve{\phi}_{k+1}^t)^4 | \mathcal{H}_k^t \right] \leq (\breve{\phi}_k^t)^4 - 4(\breve{\phi}_k^t)^2 \eta_{k+1}^t (1 - \eta_k^t L) \langle \breve{w}_k^t - w^\star, F'(\breve{w}_k^t) \rangle$$

$$+ 2(\breve{\phi}_k^t)^2 (\eta_{k+1}^t)^2 \mathbb{E}\left[ \left\| \frac{1}{P} \sum_{i=1}^P g_{i,k+1}^t(w_{i,k}^t) - F'(w_{i,k}^t) \right\|^2 | \mathcal{H}_k^t \right]$$

$$+ 4(\breve{\phi}_k^t)^2 \eta_{k+1}^t \langle \breve{w}_k^t - w^\star, F'(\breve{w}_k^t) - \frac{1}{P} \sum_{p=1}^P F'(w_{p,k}^t) \rangle$$

$$+ 3(\eta_{k+1}^t)^2 \langle \breve{w}_k^t - w^\star, \frac{1}{P} \sum_{i=1}^P F'(\breve{w}_k^t) \rangle L(\breve{\phi}_k^t)^2$$

$$+ 6(\eta_{k+1}^t)^4 \mathbb{E}\left[\left\|\frac{1}{P}\sum_{i=1}^P g_{i,k+1}^t(\boldsymbol{w}_{i,k}^t) - F'(\boldsymbol{w}_{i,k}^t)\right\|^4 |\mathcal{H}_k^t\right]$$

$$+ 6(\eta_{k+1}^t)^4 L^2(\breve{\phi}_k^t)^2 \langle \breve{\boldsymbol{w}}_k^t - \boldsymbol{w}^\star, F'(\breve{\boldsymbol{w}}_k^t)\rangle$$

$$+ 3(2\eta_{k+1}^t)^2 \langle \breve{\boldsymbol{w}}_k^t - \boldsymbol{w}^\star, \frac{1}{P}\sum_{i=1}^P F'(\breve{\boldsymbol{w}}_k^t) - \frac{1}{P}\sum_{p=1}^P F'(\boldsymbol{w}_{p,k}^t)\rangle^2.$$

*Rearranging terms and using the uniform upper bound on the 4-th moment of the noise A6, we have:*

$$\mathbb{E}\left[(\breve{\phi}_{k+1}^t)^4|\mathcal{H}_k^t\right] \leq (\breve{\phi}_k^t)^4 - 4(\breve{\phi}_k^t)^2\eta_{k+1}^t(1 - \eta_k^t L - 3\eta_k^t L - 6(\eta_{k+1}^t)^4 L^2)\langle \breve{\boldsymbol{w}}_k^t - \boldsymbol{w}^\star, F'(\breve{\boldsymbol{w}}_k^t)\rangle$$

$$+ 2(\breve{\phi}_k^t)^2(\eta_{k+1}^t)^2\frac{\sigma_\infty^2}{P} + 6(\eta_{k+1}^t)^4\frac{\sigma_\infty^4}{P^2}$$

$$+ 4(\breve{\phi}_k^t)^2\eta_{k+1}^t\langle \breve{\boldsymbol{w}}_k^t - \boldsymbol{w}^\star, F'(\breve{\boldsymbol{w}}_k^t) - \frac{1}{P}\sum_{p=1}^P F'(\boldsymbol{w}_{p,k}^t)\rangle$$

$$+ 3(2\eta_{k+1}^t)^2\mathbb{E}\left[\langle \breve{\boldsymbol{w}}_k^t - \boldsymbol{w}^\star, \frac{1}{P}\sum_{i=1}^P g_{i,k+1}^t(\breve{\boldsymbol{w}}_k^t) - \frac{1}{P}\sum_{p=1}^P F'(\boldsymbol{w}_{p,k}^t)\rangle^2 |\mathcal{H}_k^t\right].$$

(S34)

*The first 2 lines of Equation (S34) correspond to the expansion in Equation (S32) (the constants are slightly different because we use a uniform bound on the gradient instead of co-coercivity). The last two lines correspond to the residual term, for which we will use Lemma S23.*

*We have:*

$$4(\breve{\phi}_k^t)^2\eta_{k+1}^t\langle \breve{\boldsymbol{w}}_k^t - \boldsymbol{w}^\star, F'(\breve{\boldsymbol{w}}_k^t) - \frac{1}{P}\sum_{p=1}^P F'(\boldsymbol{w}_{p,k}^t)\rangle$$

$$+ 6(2\eta_{k+1}^t)^2\mathbb{E}\left[\langle \breve{\boldsymbol{w}}_k^t - \boldsymbol{w}^\star, \frac{1}{P}\sum_{i=1}^P g_{i,k+1}^t(\breve{\boldsymbol{w}}_k^t) - \frac{1}{P}\sum_{p=1}^P F'(\boldsymbol{w}_{p,k}^t)\rangle^2 |\mathcal{H}_k^t\right]$$

$$\leq 4(\breve{\phi}_k^t)^3\eta_{k+1}^t\left\|F'(\breve{\boldsymbol{w}}_k^t) - \frac{1}{P}\sum_{p=1}^P F'(\boldsymbol{w}_{p,k}^t)\right\|$$

$$+ 6(2\eta_{k+1}^t)^2 L\left\|\breve{\boldsymbol{w}}_k^t - \boldsymbol{w}^\star\right\|^3\left\|\frac{1}{P}\sum_{i=1}^P F'(\breve{\boldsymbol{w}}_k^t) - \frac{1}{P}\sum_{p=1}^P F'(\boldsymbol{w}_{p,k}^t)\right\|$$

$$= (\breve{\phi}_k^t)^3\eta_{k+1}^t(4 + 24\eta_k^t L)\left\|F'(\breve{\boldsymbol{w}}_k^t) - \frac{1}{P}\sum_{p=1}^P F'(\boldsymbol{w}_{p,k}^t)\right\|.$$

*As a result, there exist absolute constants ("numbers") $C_4, D_4, E_4$, such that if $\eta_k^t L \leq \frac{1}{C_4}$:*

$$\mathbb{E}\left[(\breve{\phi}_{k+1}^t)^4\right]^{1/2} \leq (1 - \eta_k^t\mu)\mathbb{E}\left[(\breve{\phi}_k^t)^4\right]^{1/2} + D_4(\eta_k^t)^2\frac{\sigma_\infty^2}{P}$$

$$+ E_4\eta_{k+1}^t\mathbb{E}\left[(\breve{\phi}_k^t)\left\|F'(\breve{\boldsymbol{w}}_k^t) - \frac{1}{P}\sum_{p=1}^P F'(\boldsymbol{w}_{p,k}^t)\right\|\right].$$

(S35)

*This is the result of the Lemma.*

# F  Main error decomposition

## F.1  General decomposition

In this section, we prove the following decomposition for the on-line setting.

**Lemma S31** *Under the differentiability of A2 we have[7],*

$$F''(\boldsymbol{w}^\star)(\overline{\overline{\boldsymbol{w}}}^C - \boldsymbol{w}^\star) = \frac{P\left(\boldsymbol{w}^0 - \boldsymbol{w}^\star\right)}{T\eta_1^1} - \frac{P\left(\hat{\boldsymbol{w}}^C - \boldsymbol{w}^\star\right)}{T\eta_{N^C+1}^C} - \frac{1}{T}\sum_{t=1}^{C}\sum_{k=1}^{N^t}\sum_{i=1}^{P}\left(\boldsymbol{w}_{i,k}^t - \boldsymbol{w}^\star\right)\left(\frac{1}{\eta_k^t} - \frac{1}{\eta_{k+1}^t}\right)$$

$$+ \frac{1}{T}\sum_{t=1}^{C}\sum_{k=1}^{N^t}\sum_{i=1}^{P}\delta_{i,k}^t + \frac{1}{T}\sum_{t=1}^{C}\sum_{k=1}^{N^t}\sum_{i=1}^{P}\xi_{i,k}^t,$$

*where $\xi_{i,k}^t = F'(\boldsymbol{w}_{i,k-1}^t) - \boldsymbol{g}_{i,k}^t(\boldsymbol{w}_{i,k-1}^t)$ and $\delta_{i,k}^t = F''(\boldsymbol{w}^\star)(\boldsymbol{w}_{i,k-1}^t - \boldsymbol{w}^\star) - F'(\boldsymbol{w}_{i,k-1}^t).$*

**Proof 32** *Below, we have $\boldsymbol{g}_{i,k}^t(\boldsymbol{w}_{i,k-1}^t)$ as the stochastic gradient at step $k$ on machine $i$ for communication phase $t$. After adding and subtracting few quantities and rearranging we have,*

$$\boldsymbol{w}_{i,k}^t = \boldsymbol{w}_{i,k-1}^t - \eta_k^t\boldsymbol{g}_{i,k}^t(\boldsymbol{w}_{i,k-1}^t)$$
$$\boldsymbol{w}_{i,k}^t = \boldsymbol{w}_{i,k-1}^t - \eta_k^t F'(\boldsymbol{w}_{i,k-1}^t) + \eta_k^t\left(F'(\boldsymbol{w}_{i,k-1}^t) - \boldsymbol{g}_{i,k}^t(\boldsymbol{w}_{i,k-1}^t)\right)$$
$$\boldsymbol{w}_{i,k}^t = \boldsymbol{w}_{i,k-1}^t - \eta_k^t F'(\boldsymbol{w}_{i,k-1}^t) + \eta_k^t\delta_{i,k}^t + \eta F''(\boldsymbol{w}^\star)(\boldsymbol{w}_{i,k-1}^t - \boldsymbol{w}^\star) - \eta_k^t F''(\boldsymbol{w}^\star)(\boldsymbol{w}_{i,k-1}^t - \boldsymbol{w}^\star)$$
$$\boldsymbol{w}_{i,k}^t = \boldsymbol{w}_{i,k-1}^t + \eta_k^t\xi_{i,k}^t + \eta_k^t\delta_{i,k}^t - \eta_k^t F''(\boldsymbol{w}^\star)(\boldsymbol{w}_{i,k-1}^t - \boldsymbol{w}^\star).$$

*where $\xi_{i,k}^t$ and $\delta_{i,k}^t$ are respectively terms related to stochastic noise and quadratic residual. Obtaining the horizontal average over all the machines and recalling the definition of the ghost process $\breve{\boldsymbol{w}}_k^t$ as defined above we have,*

$$\frac{1}{P}\sum_{i=1}^{P}F''(\boldsymbol{w}^\star)(\boldsymbol{w}_{i,k-1}^t - \boldsymbol{w}^\star) = \frac{1}{P}\sum_{i=1}^{P}\frac{1}{\eta_k^t}\left(\boldsymbol{w}_{i,k-1}^t - \boldsymbol{w}_{i,k}^t\right) + \frac{1}{P}\sum_{i=1}^{P}\delta_{i,k}^t + \frac{1}{P}\sum_{i=1}^{P}\xi_{i,k}^t$$

$$F''(\boldsymbol{w}^\star)(\breve{\boldsymbol{w}}_{k-1}^t - \boldsymbol{w}^\star) = \frac{\breve{\boldsymbol{w}}_{k-1}^t - \breve{\boldsymbol{w}}_k^t}{\eta_k^t} + \frac{1}{P}\sum_{i=1}^{P}\delta_{i,k}^t + \frac{1}{P}\sum_{i=1}^{P}\xi_{i,k}^t.$$

*Obtaining the vertical average over all the machines first within a communication phase and then among different phases we have,*

$$\frac{1}{N^t}\sum_{k=1}^{N^t}F''(\boldsymbol{w}^\star)(\breve{\boldsymbol{w}}_{k-1}^t - \boldsymbol{w}^\star) = \frac{1}{N^t}\sum_{k=1}^{N^t}\frac{\breve{\boldsymbol{w}}_{k-1}^t - \breve{\boldsymbol{w}}_k^t}{\eta_k^t} + \frac{1}{N^tP}\sum_{k=1}^{N^t}\sum_{i=1}^{P}\delta_{i,k}^t + \frac{1}{N^tP}\sum_{k=1}^{N^t}\sum_{i=1}^{P}\xi_{i,k}^t$$

$$\frac{1}{\sum_{t=1}^{C}N^t}\sum_{t=1}^{C}\sum_{k=1}^{N^t}F''(\boldsymbol{w}^\star)(\breve{\boldsymbol{w}}_{k-1}^t - \boldsymbol{w}^\star) = \frac{1}{\sum_{t=1}^{C}N^t}\sum_{t=1}^{C}\sum_{k=1}^{N^t}\frac{\breve{\boldsymbol{w}}_{k-1}^t - \breve{\boldsymbol{w}}_k^t}{\eta_k^t} + \frac{1}{P\sum_{t=1}^{C}N^t}\sum_{t=1}^{C}\sum_{k=1}^{N^t}\sum_{i=1}^{P}\delta_{i,k}^t$$

$$+ \frac{1}{P\sum_{t=1}^{C}N^t}\sum_{t=1}^{C}\sum_{k=1}^{N^t}\sum_{i=1}^{P}\xi_{i,k}^t.$$

*Now recalling the definitions for the overall iterate $\overline{\overline{\boldsymbol{w}}}^C = \frac{1}{\sum_{t=1}^{C}N^t}\sum_{t=1}^{C}\sum_{k=1}^{N^t}\breve{\boldsymbol{w}}_k^t$, $\hat{\boldsymbol{w}}^t = \breve{\boldsymbol{w}}_{N^t}^t$, the initial point $\hat{\boldsymbol{w}}^0 = \boldsymbol{w}^0$, and the total number of gradients $T = P\sum_{t=1}^{C}N^t$ as we have defined above. After making these changes and on rearranging we obtain,*

$$F''(\boldsymbol{w}^\star)(\overline{\overline{\boldsymbol{w}}}^C - \boldsymbol{w}^\star) = \frac{P}{T}\sum_{t=1}^{C}\sum_{k=1}^{N^t}\frac{\breve{\boldsymbol{w}}_{k-1}^t - \breve{\boldsymbol{w}}_k^t}{\eta_k^t} + \frac{1}{T}\sum_{t=1}^{C}\sum_{k=1}^{N^t}\sum_{i=1}^{P}\delta_{i,k}^t + \frac{1}{T}\sum_{t=1}^{C}\sum_{k=1}^{N^t}\sum_{i=1}^{P}\xi_{i,k}^t$$

$$F''(\boldsymbol{w}^\star)(\overline{\overline{\boldsymbol{w}}}^C - \boldsymbol{w}^\star) = \frac{P\left(\boldsymbol{w}^0 - \boldsymbol{w}^\star\right)}{T\eta_1^1} - \frac{P\left(\hat{\boldsymbol{w}}^C - \boldsymbol{w}^\star\right)}{T\eta_{N^C+1}^C} - \frac{P}{T}\sum_{t=1}^{C}\sum_{k=1}^{N^t}\left(\breve{\boldsymbol{w}}_k^t - \boldsymbol{w}^\star\right)\left(\frac{1}{\eta_k^t} - \frac{1}{\eta_{k+1}^t}\right)$$

$$+ \frac{1}{T}\sum_{t=1}^{C}\sum_{k=1}^{N^t}\sum_{i=1}^{P}\delta_{i,k}^t + \frac{1}{T}\sum_{t=1}^{C}\sum_{k=1}^{N^t}\sum_{i=1}^{P}\xi_{i,k}^t.$$

*Thus we have obtained the required result as,*

$$F''(\boldsymbol{w}^\star)(\overline{\boldsymbol{w}}^C - \boldsymbol{w}^\star) = \frac{P\left(\boldsymbol{w}^0 - \boldsymbol{w}^\star\right)}{T\eta_1^1} - \frac{P\left(\hat{\boldsymbol{w}}^C - \boldsymbol{w}^\star\right)}{T\eta_{N^C+1}^C} - \frac{1}{T}\sum_{t=1}^{C}\sum_{k=1}^{N^t}\sum_{i=1}^{P}\left(\boldsymbol{w}_{i,k}^t - \boldsymbol{w}^\star\right)\left(\frac{1}{\eta_k^t} - \frac{1}{\eta_{k+1}^t}\right)$$

$$+ \frac{1}{T}\sum_{t=1}^{C}\sum_{k=1}^{N^t}\sum_{i=1}^{P}\delta_{i,k}^t + \frac{1}{T}\sum_{t=1}^{C}\sum_{k=1}^{N^t}\sum_{i=1}^{P}\xi_{i,k}^t.$$

### F.2 Bounding the noise term

The stochastic noise term which appears above can be bounded using the following lemma,

**Lemma S33** *Under the Assumptions A3, A5, A6 we have*

$$\mathbb{E}\left[\left\|\xi_{i,k}^t\right\|^2\right] \le 2L^2\mathbb{E}\left[\left\|\boldsymbol{w}_{i,k-1}^t\boldsymbol{w}^\star\right\|^2\right] + 2\sigma^2.$$

**Proof 34** *Using Assumptions A3, A5, A6 respectively we prove the result*

$$\mathbb{E}\left[\left\|\xi_{i,k}^t\right\|^2\right] = \mathbb{E}\left[\left\|F'(\boldsymbol{w}_{i,k-1}^t) - \boldsymbol{g}_{i,k}^t(\boldsymbol{w}_{i,k-1}^t)\right\|^2\right] \le \mathbb{E}\left[\left\|\boldsymbol{g}_{i,k}^t(\boldsymbol{w}_{i,k-1}^t)\right\|^2\right] - \left\|F'(\boldsymbol{w}_{i,k-1}^t)\right\|^2$$

$$\le 2\mathbb{E}\left[\left\|\boldsymbol{g}_{i,k}^t(\boldsymbol{w}_{i,k-1}^t) - \boldsymbol{g}_{i,k}^t(\boldsymbol{w}^\star)\right\|^2\right] + 2\mathbb{E}\left[\left\|\boldsymbol{g}_{i,k}^t(\boldsymbol{w}^\star)\right\|^2\right]$$

$$\le 2L^2\mathbb{E}\left[\left\|\boldsymbol{w}_{i,k-1}^t - \boldsymbol{w}^\star\right\|^2\right] + 2\sigma^2.$$

## G  Proofs for OSA, MBA and Local-SGD in the finite horizon setting

In this Section and Appendix H we prove convergence results for $\mathbb{E}\left[\left\|F''(\boldsymbol{w}^\star)(\overline{\boldsymbol{w}}^C - \boldsymbol{w}^\star)\right\|\right]$. The proof technique is the one proposed by Polyak and Judisky in the original article on averaging [1]. This proof technique has also been used in [10, 15]. We notice here the following differences, that justify including the proofs:

1. Polyak and Judisky were mainly interested in the asymptotic analysis, and the set of assumptions considered was different.

2. In [10], the authors prove comparable bounds in the case of bounded gradients. However, their analysis in the smooth and strongly convex setting is not optimal. Precisely, they use a sub-optimal upper bound when controlling the second order moments, that significantly worsens the subsequent proof. This point was underlined in [25, 47]. The result they provide under our set of assumptions is eventually 1) not optimal, 2) uselessly complex, and 3) only for serial-SGD.

3. In [15], authors prove a result close to us, using a similar approach for one-shot averaging. Their bounds only apply to decaying step size. Moreover, they rely on the following asymptotic upper bound: $\mathbb{E}\left[\left\|\boldsymbol{w}_{i,k}^t - \boldsymbol{w}^\star\right\|^2\right] \le C_1\eta_k^t$: this bound is correct but the constant $C_1$ is "asymptotic" (see for e.g., [34]). On contrary, we use non-asymptotic upper bounds on the second order moment involved. As a consequence, our bounds are both simpler and tighter.

### G.1 Technical Lemmas

**Lemma S35 (Jensen's Inequality)** *For $a_i \in \mathbb{R}^d$, $\left\|\frac{1}{P}\sum_{i=1}^{P}a_i\right\|^2 \le \frac{1}{P}\sum_{i=1}^{P}\|a_i\|^2$.*

**Proof 36** *The result is an application of Jensen's inequality with the convex function $f(.) = \|.\|^2$.*

**Lemma S37 (Minkowski's Inequality)** *For $a_i \in \mathbb{R}^d$, $\mathbb{E}\left[\left\|\sum_{i=1}^{P} a_i\right\|^2\right] \leq \left(\sum_{i=1}^{P} \mathbb{E}\left[\|a_i\|^2\right]^{\frac{1}{2}}\right)^2$*

**Proof 38** *The inequality is an application of Minkowski's inequality (or simply triangle's inequality) with the norm $\|.\|_E = \mathbb{E}\left[\|.\|^2\right]^{\frac{1}{2}}$.*

### G.2 Proof of Proposition 1 (Mini-batch case)

Lemma S8 proves the first part of the proposition. We prove the second part of the proposition here following the approach by [1]. Using Lemma S31, Lemma S24 we can obtain an upper bound on $\mathbb{E}\left[\left\|F''(\boldsymbol{w}^\star)(\overline{\overline{\boldsymbol{w}}}^C - \boldsymbol{w}^\star)\right\|^2\right]$, which is in-fact a tighter quantity when compared to $\mathbb{E}\left[\left\|\overline{\overline{\boldsymbol{w}}}^C - \boldsymbol{w}^\star\right\|^2\right]$. We prove the following lemma,

**Lemma S39** *Under the Assumptions **A**1, **A**2, **A**3, **A**5, **A**6 we have,*

$$\mathbb{E}\left[\left\|\nabla^2 F(\boldsymbol{w}^\star)(\boldsymbol{w} - \boldsymbol{w}^\star)\right\|^2\right] \leq 4 \sum_{i=1}^{5} A_{i,P,C}^2,$$

*where the terms are respectively,*

$$A_{1,P,C}^2 = \frac{P^2}{T^2\eta^2}\left\|\boldsymbol{w}^0 - \boldsymbol{w}^\star\right\|^2, A_{2,P,C}^2 = \frac{P^2}{T^2\eta^2}\left((1-\mu\eta)^C\left\|\boldsymbol{w}^0 - \boldsymbol{w}^\star\right\|^2 + 2\sigma^2\frac{\eta}{\mu P}\right),$$

$$A_{3,P,C}^2 = \frac{P^2 M^2}{T^2\mu^2\eta^2}\left(\left\|\boldsymbol{w}^0 - \boldsymbol{w}^\star\right\|^2 + \frac{C 20\eta^2}{P}\sigma^2\right)^2, A_{4,P,C}^2 = \frac{2\sigma^2}{T},$$

$$A_{3,P,C}^2 = \frac{2L^2 P}{T^2}\left(\frac{1}{\mu\eta}\left\|\boldsymbol{w}^0 - \boldsymbol{w}^\star\right\|^2 + 2\sigma^2\frac{(C\mu\eta - 1 + (1-\mu\eta)^C)}{\mu^2 P}\right).$$

**Proof 40** *In order to upper bound the expectation we need to separately upper bound all the terms that appear in the result for Lemma S31. But before that we can actually simplify the result with constant step size and using $N^t = 1 \, \forall t \in [C]$ as follows,*

$$F''(\boldsymbol{w}^\star)(\overline{\overline{\boldsymbol{w}}}^C - \boldsymbol{w}^\star) = \frac{\boldsymbol{w}^0 - \boldsymbol{w}^\star}{C\eta} - \frac{\hat{\boldsymbol{w}}^C - \boldsymbol{w}^\star}{C\eta} + \frac{1}{T}\sum_{t=1}^{C}\sum_{i=1}^{P}\delta_{i,1}^t + \frac{1}{T}\sum_{t=1}^{C}\sum_{i=1}^{P}\xi_{i,1}^t.$$

*Now we bound each of the terms in the above decomposition one by one. For the first term,*

$$\mathbb{E}\left[\left\|\frac{1}{C\eta}\left(\boldsymbol{w}^0 - \boldsymbol{w}^\star\right)\right\|^2\right] = \frac{P^2}{T^2\eta^2}\left\|\boldsymbol{w}^0 - \boldsymbol{w}^\star\right\|^2 = A_{1,P,C}^2.$$

*For the second term using Lemma S8,*

$$\mathbb{E}\left[\left\|\frac{1}{C\eta}\left(\hat{\boldsymbol{w}}^C - \boldsymbol{w}^\star\right)\right\|^2\right] = \frac{P^2}{T^2\eta^2}\mathbb{E}\left[\left\|\boldsymbol{w}_{MB}^C - \boldsymbol{w}^\star\right\|^2\right]$$

$$\leq \frac{P^2}{T^2\eta^2}\left(\prod_{k=1}^{C}(1-\mu\eta)\mathbb{E}\left[\left\|\boldsymbol{w}^0 - \boldsymbol{w}^\star\right\|^2\right] + 2\sigma^2\frac{1}{P}\sum_{k=1}^{C}\prod_{l=k+1}^{C}(1-\mu\eta)\eta^2\right)$$

$$\leq \frac{P^2}{T^2\eta^2}\left((1-\mu\eta)^C\left\|\boldsymbol{w}^0 - \boldsymbol{w}^\star\right\|^2 + 2\sigma^2\frac{1}{P}\left(\frac{1-(1-\mu\eta)^C}{\mu\eta}\right)\eta^2\right)$$

$$\leq \frac{P^2}{T^2\eta^2}\left((1-\mu\eta)^C\left\|\boldsymbol{w}^0 - \boldsymbol{w}^\star\right\|^2 + 2\sigma^2\frac{\eta}{\mu P}\right) = A_{2,P,C}^2.$$

*For the third term using Lemma S35 and Lemma S37 we get,*

$$\mathbb{E}\left[\left\|\frac{1}{T}\sum_{t=1}^{C}\sum_{i=1}^{P}\delta_{i,1}^{t}\right\|^{2}\right] = \frac{1}{T^{2}}\mathbb{E}\left[\left\|\sum_{t=1}^{C}\sum_{i=1}^{P}\left(F'(\boldsymbol{w}_{i,0}^{t}) - F''(\boldsymbol{w}^{\star})(\boldsymbol{w}_{i,0}^{t} - \boldsymbol{w}^{\star})\right)\right\|^{2}\right]$$

$$\leq \frac{P}{T^{2}}\sum_{i=1}^{P}\mathbb{E}\left[\left\|\sum_{t=1}^{C}\left(F'(\hat{\boldsymbol{w}}^{t-1}) - F''(\boldsymbol{w}^{\star})(\hat{\boldsymbol{w}}^{t-1} - \boldsymbol{w}^{\star})\right)\right\|^{2}\right]$$

$$\leq \frac{P^{2}}{T^{2}}\left(\sum_{t=1}^{C}\sqrt{\mathbb{E}\left[\left\|(F'(\hat{\boldsymbol{w}}^{t-1}) - F''(\boldsymbol{w}^{\star})(\hat{\boldsymbol{w}}^{t-1} - \boldsymbol{w}^{\star}))\right\|^{2}\right]}\right)^{2}.$$

*Now using the upper bound from **A**2 followed by Lemma S25 we get,*

$$\mathbb{E}\left[\left\|\frac{1}{T}\sum_{t=1}^{C}\sum_{i=1}^{P}\delta_{i,1}^{t}\right\|^{2}\right] \leq \frac{P^{2}M^{2}}{T^{2}}\left(\sum_{t=1}^{C}\sqrt{\mathbb{E}\left[\left\|\hat{\boldsymbol{w}}^{t-1} - \boldsymbol{w}^{\star}\right\|^{4}\right]}\right)^{2}$$

$$\leq \frac{P^{2}M^{2}}{T^{2}}\left(\sum_{t=1}^{C}\left((1-\eta\mu)^{t-1}\mathbb{E}\left[(\hat{\boldsymbol{w}}^{0} - \boldsymbol{w}^{\star})^{4}\right]^{1/2} + \frac{20\eta}{P\mu}\sigma^{2}\right)\right)^{2}$$

$$\leq \frac{P^{2}M^{2}}{T^{2}}\left(\frac{1-(1-\eta\mu)^{C}}{\eta\mu}\mathbb{E}\left[(\hat{\boldsymbol{w}}^{0} - \boldsymbol{w}^{\star})^{4}\right]^{1/2} + \frac{20C\eta}{P\mu}\sigma^{2}\right)^{2}$$

$$\leq \frac{P^{2}M^{2}}{T^{2}\mu^{2}\eta^{2}}\left(\left\|\boldsymbol{w}^{0} - \boldsymbol{w}^{\star}\right\|^{2} + \frac{20C\eta^{2}}{P}\sigma^{2}\right)^{2} = A_{3,P,C}^{2}.$$

*For the fourth term, note that we are sampling i.i.d observations and thus the stochastic noise across all machines and iterations is independent and equal to zero in expectation (see **A**3). This implies the first equation below while the second inequality is obtained using Lemma S33,*

$$\mathbb{E}\left[\left\|\frac{1}{T}\sum_{t=1}^{C}\sum_{i=1}^{P}\xi_{i,1}^{t}\right\|^{2}\right] = \frac{1}{T^{2}}\sum_{t=1}^{C}\sum_{i=1}^{P}\mathbb{E}\left[\left\|\xi_{i,1}^{t}\right\|^{2}\right] \leq \frac{1}{T^{2}}\sum_{t=1}^{C}\sum_{i=1}^{P}\left(2L^{2}\mathbb{E}\left[\left\|\boldsymbol{w}_{i,0}^{t} - \boldsymbol{w}^{\star}\right\|^{2}\right] + 2\sigma^{2}\right)$$

$$\leq \frac{2\sigma^{2}}{T} + \frac{2L^{2}P}{T^{2}}\sum_{t=1}^{C}\mathbb{E}\left[\left\|\boldsymbol{w}_{1,0}^{t} - \boldsymbol{w}^{\star}\right\|^{2}\right].$$

*Now using Lemma S8 we have,*

$$\mathbb{E}\left[\left\|\frac{1}{T}\sum_{t=1}^{C}\sum_{i=1}^{P}\xi_{i,1}^{t}\right\|^{2}\right] \leq \frac{2\sigma^{2}}{T} + \frac{2L^{2}P}{T^{2}}\sum_{t=1}^{C}\mathbb{E}\left[\left\|\hat{\boldsymbol{w}}_{MB}^{t-1} - \boldsymbol{w}^{\star}\right\|^{2}\right]$$

$$\leq \frac{2\sigma^{2}}{T} + \frac{2L^{2}P}{T^{2}}\sum_{t=1}^{C}\left((1-\mu\eta)^{t-1}\left\|\boldsymbol{w}^{0} - \boldsymbol{w}^{\star}\right\|^{2} + 2\sigma^{2}\frac{\eta\left(1-(1-\mu\eta)^{C}\right)}{\mu P}\right)$$

$$\leq \frac{2\sigma^{2}}{T} + \frac{2L^{2}P}{T^{2}}\left(\frac{1-(1-\mu\eta)^{C}}{\mu\eta}\left\|\boldsymbol{w}^{0} - \boldsymbol{w}^{\star}\right\|^{2} + 2\sigma^{2}\frac{\left(C\mu\eta - (1-(1-\mu\eta)^{C})\right)}{\mu^{2}P}\right)$$

$$\leq \frac{2\sigma^{2}}{T} + \frac{2L^{2}P}{T^{2}}\left(\frac{1}{\mu\eta}\left\|\boldsymbol{w}^{0} - \boldsymbol{w}^{\star}\right\|^{2} + 2\sigma^{2}\frac{C\eta}{\mu P}\right)$$

$$= A_{4,P,C}^{2} + A_{5,P,C}^{2}.$$

*Now using Lemma S35, we have proved the lemma.*

It can be seen in the above lemma that there are two kinds of terms: one that depend on the history or initialization and second the ones that depend on the variance bound. This implies that it would be possible to restate Lemma S39 as follows,

**Lemma S41** *Under the assumptions A1, A2, A3, A5, A6 we have,*

$$\mathbb{E}\left[\left\|\nabla^2 F(\boldsymbol{w}^\star)(\boldsymbol{w} - \boldsymbol{w}^\star)\right\|^2\right] \leq 4(\hat{A}_{1,P,C}^2 + \hat{A}_{2,P,C}^2)$$

*Where the terms are respectively,*

$$\hat{A}_{1,P,C}^2 = \frac{\left\|\boldsymbol{w}^0 - \boldsymbol{w}^\star\right\|^2}{\eta^2 C^2}\left(1 + (1 - \mu\eta)^C + \frac{2M^2}{\mu^2}\left\|\boldsymbol{w}^0 - \boldsymbol{w}^\star\right\|^2 + \frac{2L^2\eta}{\mu P}\right),$$

$$\hat{A}_{2,P,C}^2 = \frac{2\sigma^2}{T}\left(1 + \frac{P}{T\eta\mu} + \frac{400M^2 C^2\eta^2\sigma^2}{T\mu^2} + \frac{2L^2 C\eta}{T\mu}\right).$$

Ignoring constants the above constants can be upper bounded as follows,

$$\hat{A}_{1,P,C}^2 \leq \frac{\left\|\boldsymbol{w}^0 - \boldsymbol{w}^\star\right\|^2}{\eta^2 C^2}\left(1 + 1 + \frac{2M^2}{\mu^2}\left\|\boldsymbol{w}^0 - \boldsymbol{w}^\star\right\|^2 + \frac{2L^2\eta}{\mu P}\right)$$

$$\leq 2\frac{\left\|\boldsymbol{w}^0 - \boldsymbol{w}^\star\right\|^2}{\eta^2 C^2}\left(1 + \frac{M^2}{\mu^2}\left\|\boldsymbol{w}^0 - \boldsymbol{w}^\star\right\|^2 + \frac{L^2\eta}{\mu P}\right)$$

$$\precsim \frac{\left\|\boldsymbol{w}^0 - \boldsymbol{w}^\star\right\|^2}{\eta^2 C^2}\left(1 + \frac{M^2}{\mu^2}\left\|\boldsymbol{w}^0 - \boldsymbol{w}^\star\right\|^2 + \frac{L^2\eta}{\mu P}\right),$$

$$\hat{A}_{2,P,C}^2 \leq 800\frac{\sigma^2}{T}\left(1 + \frac{P}{T\eta\mu} + \frac{M^2 C^2\eta^2\sigma^2}{T\mu^2} + \frac{L^2 C\eta}{T\mu}\right)$$

$$\precsim \frac{\sigma^2}{T}\left(1 + \frac{P}{T\eta\mu} + \frac{M^2 C^2\eta^2\sigma^2}{T\mu^2} + \frac{L^2 C\eta}{T\mu}\right).$$

Thus, we recover Proposition 1.

### G.3   Proof Proposition 2 (One-shot averaging case)

To prove the proposition we need to prove a bound on second moment of the inner iterations followed by a bound on the final average outer iteration. For inner iterations we follow the result from [53] as the process on a single worker is completely independent of any other worker. We have the following lemma,

**Lemma S42** *Under the Assumptions A1, A2, A3, A5, A6 for constant step size for one shot averaging we have,*

$$\mathbb{E}\left[\left\|F''(\boldsymbol{w}^\star)(\boldsymbol{w}_{i,k}^1 - \boldsymbol{w}^\star)\right\|^2\right] \leq 4\sum_{i=1}^5 B_{i,P,N^1}^2$$

*where the terms are respectively,*

$$B_{1,P,N^1}^2 = \frac{P^2}{T^2\eta^2}\left\|\boldsymbol{w}^0 - \boldsymbol{w}^\star\right\|^2, B_{2,P,N^1}^2 = \frac{P^2}{T^2\eta^2}\left((1 - \mu\eta)^{N^1}\left\|\boldsymbol{w}^0 - \boldsymbol{w}^\star\right\|^2 + \frac{2\sigma^2\eta}{\mu}\right),$$

$$B_{3,P,N^1}^2 = \frac{P^2 M^2}{T^2\mu\eta}\left(\left\|\boldsymbol{w}^0 - \boldsymbol{w}^\star\right\|^2 + 20\eta^2 N^1\sigma^2\right)^2, B_{4,P,N^1}^2 = \frac{2\sigma^2}{T},$$

$$B_{5,P,N^1}^2 = \frac{2L^2 P}{T^2}\left(\frac{1}{\mu\eta}\left\|\boldsymbol{w}^0 - \boldsymbol{w}^\star\right\|^2 + \frac{2\sigma^2 N^1\eta}{\mu}\right).$$

**Proof 43** *We follow the same line of proof as before. We can use the decomposition from Lemma S31 with constant step size and $C = 1$, which results in the following simpler decomposition,*

$$F''(\boldsymbol{w}^\star)(\overline{\overline{\boldsymbol{w}}}^C - \boldsymbol{w}^\star) = \frac{\boldsymbol{w}^0 - \boldsymbol{w}^\star}{N\eta} - \frac{\hat{\boldsymbol{w}}^1 - \boldsymbol{w}^\star}{N^1\eta} + \frac{1}{T}\sum_{k=1}^{N^1}\sum_{i=1}^P \delta_{i,k}^1 + \frac{1}{T}\sum_{k=1}^{N^1}\sum_{i=1}^P \xi_{i,k}^1$$

*For the first term,*

$$\mathbb{E}\left[\left\|\frac{\boldsymbol{w}^0 - \boldsymbol{w}^\star}{N^1 \eta}\right\|^2\right] \leq \frac{P^2}{T^2 \eta^2} \left\|\boldsymbol{w}^0 - \boldsymbol{w}^\star\right\|^2 = B_{1,P,N^1}^2.$$

*For the second term using Lemma S10 and rearranging we have,*

$$\mathbb{E}\left[\left\|\frac{\hat{\boldsymbol{w}}^1 - \boldsymbol{w}^\star}{N^1 \eta}\right\|^2\right] = \mathbb{E}\left[\left\|\frac{1}{PN^1\eta}\sum_{i=1}^{P}\boldsymbol{w}_{i,N^1}^1 - \boldsymbol{w}^\star\right\|^2\right] \leq \frac{P}{T^2\eta^2}\sum_{i=1}^{P}\mathbb{E}\left[\left\|\boldsymbol{w}_{i,N^1}^1 - \boldsymbol{w}^\star\right\|^2\right]$$

$$\leq \frac{P^2}{T^2\eta^2}\left(\prod_{l=1}^{N^1}(1-\mu\eta)\left\|\boldsymbol{w}^0 - \boldsymbol{w}^\star\right\|^2 + 2\sigma^2\sum_{l=1}^{N^1}\prod_{m=l+1}^{N^1}(1-\mu\eta)\eta^2\right)$$

$$\leq \frac{P^2}{T^2\eta^2}\left((1-\mu\eta)^{N^1}\left\|\boldsymbol{w}^0 - \boldsymbol{w}^\star\right\|^2 + 2\sigma^2\frac{1-(1-\mu\eta)^{N^1}}{\mu\eta}\eta^2\right)$$

$$\leq \frac{P^2}{T^2\eta^2}\left((1-\mu\eta)^{N^1}\left\|\boldsymbol{w}^0 - \boldsymbol{w}^\star\right\|^2 + \frac{2\sigma^2\eta}{\mu}\right) = B_{2,P,N^1}^2.$$

*For the third term using Lemma S35 and Lemma S37 we obtain,*

$$\mathbb{E}\left[\left\|\frac{1}{T}\sum_{i=1}^{P}\sum_{k=1}^{N^1}\delta_{i,k}^1\right\|^2\right] = \frac{1}{T^2}\mathbb{E}\left[\left\|\sum_{i=1}^{P}\sum_{k=1}^{N^1}F'(\boldsymbol{w}_{i,k-1}^t) - F''(\boldsymbol{w}^\star)(\boldsymbol{w}_{i,k-1}^t - \boldsymbol{w}^\star)\right\|^2\right]$$

$$\leq \frac{P}{T^2}\sum_{i=1}^{P}\mathbb{E}\left[\left\|\sum_{k=1}^{N^1}F'(\boldsymbol{w}_{i,k-1}^t) - F''(\boldsymbol{w}^\star)(\boldsymbol{w}_{i,k-1}^t - \boldsymbol{w}^\star)\right\|^2\right]$$

$$\leq \frac{P}{T^2}\sum_{i=1}^{P}\left(\sum_{k=1}^{N^1}\sqrt{\mathbb{E}\left[\left\|F'(\boldsymbol{w}_{i,k-1}^1) - F''(\boldsymbol{w}^\star)(\boldsymbol{w}_{i,k-1}^1 - \boldsymbol{w}^\star)\right\|^2\right]}\right)^2$$

*Now first using the upper bound of A2, followed by Lemma S24 and some rearranging we can obtain the following,*

$$\mathbb{E}\left[\left\|\frac{1}{T}\sum_{i=1}^{P}\sum_{k=1}^{N^1}\delta_{i,k}^1\right\|^2\right] \leq \frac{PM^2}{T^2}\sum_{i=1}^{P}\left(\sum_{k=1}^{N^1}\mathbb{E}\left[\left\|\boldsymbol{w}_{i,k-1}^1 - \boldsymbol{w}^\star\right\|^4\right]^{1/2}\right)^2$$

$$\leq \frac{PM^2}{T^2}\sum_{i=1}^{P}\left(\sum_{k=1}^{N^1}\left((1-\mu\eta)^{k-1}\mathbb{E}\left[\left\|\boldsymbol{w}_{i,0}^1 - \boldsymbol{w}^\star\right\|^4\right]^{1/2} + \frac{20\eta\sigma^2}{\mu}\right)\right)^2$$

$$\leq \frac{P^2M^2}{T^2}\left(\sum_{k=1}^{N^1}\left((1-\mu\eta)^{k-1}\left\|\boldsymbol{w}^0 - \boldsymbol{w}^\star\right\|^2 + \frac{20\eta\sigma^2}{\mu}\right)\right)^2$$

$$\leq \frac{P^2M^2}{T^2}\left(\frac{1-(1-\mu\eta)^{N^1}}{\mu\eta}\left\|\boldsymbol{w}^0 - \boldsymbol{w}^\star\right\|^2 + \frac{20\eta N^1\sigma^2}{\mu}\right)^2$$

$$\leq \frac{P^2M^2}{T^2\mu^2\eta^2}\left(\left\|\boldsymbol{w}^0 - \boldsymbol{w}^\star\right\|^2 + 20\eta^2 N^1\sigma^2\right)^2 = B_{3,P,N^1}^2.$$

*For the fourth term, using the fact that on different machines noise of the gradient is i.i.d. over different iterations and zero in expectation (A3) we obtain,*

$$\mathbb{E}\left[\left\|\frac{1}{T}\sum_{i=1}^{P}\sum_{k=1}^{N^1}\xi_{i,k}^1\right\|^2\right] = \frac{1}{T^2}\sum_{i=1}^{P}\sum_{k=1}^{N^1}\mathbb{E}\left[\left\|\xi_{i,k}^1\right\|^2\right].$$

*Now using Lemma S33 we have,*

$$\mathbb{E}\left[\left\|\frac{1}{T}\sum_{i=1}^{P}\sum_{k=1}^{N^1}\xi_{i,k}^1\right\|^2\right] \leq \frac{1}{T^2}\sum_{i=1}^{P}\sum_{k=1}^{N^1}\left(2L^2\mathbb{E}\left[\left\|\boldsymbol{w}_{i,k-1}^1 - \boldsymbol{w}^\star\right\|^2\right] + 2\sigma^2\right)$$

$$\leq \frac{2\sigma^2}{T} + \frac{2L^2}{T^2}\sum_{i=1}^{P}\sum_{k=1}^{N^1}\mathbb{E}\left[\left\|\boldsymbol{w}_{i,k-1}^1 - \boldsymbol{w}^\star\right\|^2\right].$$

*Now using Lemma S10 we have,*

$$\mathbb{E}\left[\left\|\frac{1}{T}\sum_{i=1}^{P}\sum_{k=1}^{N^1}\xi_{i,k}^1\right\|^2\right] \leq \frac{2\sigma^2}{T} + \frac{2L^2P}{T^2}\sum_{k=1}^{N^1}\left(\prod_{l=1}^{k-1}(1-\mu\eta)\left\|\boldsymbol{w}^0-\boldsymbol{w}^\star\right\|^2 + 2\sigma^2\sum_{l=1}^{k-1}\prod_{m=l+1}^{k-1}(1-\mu\eta)\eta^2\right)$$

$$\leq \frac{2\sigma^2}{T} + \frac{2L^2P}{T^2}\sum_{k=1}^{N^1}\left((1-\mu\eta)^{k-1}\left\|\boldsymbol{w}^0-\boldsymbol{w}^\star\right\|^2 + \frac{2\sigma^2\eta}{\mu}\right)$$

$$\leq \frac{2\sigma^2}{T} + \frac{2L^2P}{T^2}\left(\frac{1}{\mu\eta}\left\|\boldsymbol{w}^0-\boldsymbol{w}^\star\right\|^2 + \frac{N^12\sigma^2\eta}{\mu}\right)$$

$$= B_{4,P,N^1}^2 + B_{5,P,N^1}^2.$$

*Finally using Lemma S37, concludes the proof.*

Similar to the mini-batch case, there are two kinds of terms one that depend on the history or initialization and second that depend on the variance bound of the functions. This implies that it would be possible to restate Lemma S42 as follows,

**Lemma S44** *Under the Assumptions **A**3, **A**2, **A**1, **A**5, **A**6 we have,*

$$\mathbb{E}\left[\left\|\nabla^2 F(\boldsymbol{w}^\star)(\boldsymbol{w}-\boldsymbol{w}^\star)\right\|^2\right] \leq 4(\hat{B}_{1,P,N^1}^2 + \hat{B}_{2,P,N^1}^2)$$

*Where the terms are respectively,*

$$\hat{B}_{1,P,N^1}^2 = \frac{\left\|\boldsymbol{w}^0-\boldsymbol{w}^\star\right\|^2}{(N^1)^2\eta^2}\left(1 + (1-\mu\eta)^{N^1} + \frac{2M^2\eta}{\mu}\left\|\boldsymbol{w}^0-\boldsymbol{w}^\star\right\|^2 + \frac{2L^2\eta}{P\mu}\right),$$

$$\hat{B}_{2,P,N^1}^2 = \frac{2\sigma^2}{T}\left(1 + \frac{2L^2\eta}{\mu} + \frac{P^2}{T\mu\eta} + \frac{400M^2\sigma^2\eta^2T}{\mu^2}\right).$$

On upper-bounding the above two terms while ignoring the constants,

$$\hat{B}_{1,P,N^1}^2 \leq \frac{\left\|\boldsymbol{w}^0-\boldsymbol{w}^\star\right\|^2}{(N^1)^2\eta^2}\left(1 + 1 + \frac{2M^2\eta}{\mu}\left\|\boldsymbol{w}^0-\boldsymbol{w}^\star\right\|^2 + \frac{2L^2\eta}{P\mu}\right)$$

$$\leq 2\frac{\left\|\boldsymbol{w}^0-\boldsymbol{w}^\star\right\|^2}{(N^1)^2\eta^2}\left(1 + \frac{M^2\eta}{\mu}\left\|\boldsymbol{w}^0-\boldsymbol{w}^\star\right\|^2 + \frac{L^2\eta}{P\mu}\right)$$

$$\precsim \frac{\left\|\boldsymbol{w}^0-\boldsymbol{w}^\star\right\|^2}{(N^1)^2\eta^2}\left(1 + \frac{M^2\eta}{\mu}\left\|\boldsymbol{w}^0-\boldsymbol{w}^\star\right\|^2 + \frac{L^2\eta}{P\mu}\right),$$

$$\hat{B}_{2,P,N^1}^2 \leq 800\frac{\sigma^2}{T}\left(1 + \frac{L^2\eta}{\mu} + \frac{P^2}{T\mu\eta} + \frac{M^2\sigma^2\eta^2T}{\mu^2}\right)$$

$$\hat{B}_{2,P,N^1}^2 \precsim \frac{\sigma^2}{T}\left(1 + \frac{L^2\eta}{\mu} + \frac{P^2}{T\mu\eta} + \frac{M^2\sigma^2\eta^2T}{\mu^2}\right).$$

Thus we have recovered Proposition 2.

# H Proofs for OSA, MBA and Local-SGD in the online setting

Recall that the step size at iteration $(t, k), \in [C] \times [N^t]$ is defined as $\eta_k^t = \frac{c_\eta}{\left(\sum_{t'=1}^{t-1} N^t + k\right)^\alpha}$ where $\alpha \in (0, 1)$. Though our results can be extended for the entire range of learning rates, we prove results only for $\alpha \in (\frac{1}{2}, 1)$.

## H.1 Technical Lemmas

We first state a few technical results which are helpful in the following proofs.

**Lemma S45** *For $\tilde{\eta}_m = \frac{c_\eta}{m^\alpha}$, $\alpha \in (0, 1)$ we have $\prod_{m=1}^t (1 - \mu \tilde{\eta}_m) \leq \exp\left(-\frac{\mu c_\eta t^{1-\alpha}}{2(1-\alpha)}\right)$.*

**Proof 46** *The proof simply follows from applying the inequality $1 + x \leq \exp(x)$, followed by an integral bound over the series as $\sum_{m=1}^t \frac{1}{m^\alpha} \geq \frac{1}{2} \int_0^t \frac{1}{m^\alpha} dm = \frac{t^{1-\alpha}}{1-\alpha}$. Note that it is possible to consider $\alpha = 1$ but the integral bound changes. For brevity we don't include it here.*

**Lemma S47** *For $\tilde{\eta}_m = \frac{c_\eta}{m^\alpha}$, $\alpha \in (0, 1)$ we have*

$$\sum_{m=1}^t (\tilde{\eta}_m)^2 \prod_{l=m+1}^t (1 - \mu \tilde{\eta}_l) \leq \exp\left(-\frac{\mu c_\eta t^{1-\alpha}}{2(1-\alpha)}\left(1 - \frac{1}{2^{1-\alpha}}\right)\right) c_\eta^2 \left(1 + \frac{t^{1-2\alpha} - 1}{1 - 2\alpha}\right) + \frac{2c_\eta}{t^\alpha \mu}.$$

*Further if $\alpha \in (\frac{1}{2}, 1)$, then for large $t$, $\sum_{m=1}^t (\tilde{\eta}_m)^2 \prod_{l=m+1}^t (1 - \mu \tilde{\eta}_l) \leq \exp\left(-\frac{\mu c_\eta t^{1-\alpha}}{2(1-\alpha)}\left(1 - \frac{1}{2^{1-\alpha}}\right)\right) \frac{2\alpha c_\eta^2}{2\alpha - 1} + \frac{2c_\eta}{t^\alpha \mu}$.*

**Proof 48** *First we decompose the term, then use $1 + x \leq \exp(x)$, followed by a series of integral bounds like Lemma S45,*

$$\sum_{m=1}^t \tilde{\eta}_m^2 \prod_{l=m+1}^t (1 - \mu \tilde{\eta}_l) \leq \sum_{m=1}^{\frac{t}{2}} (\tilde{\eta}_m)^2 \prod_{l=m+1}^t (1 - \mu \tilde{\eta}_l) + \sum_{m=\frac{t}{2}}^t (\tilde{\eta}_m)^2 \prod_{l=m+1}^t (1 - \mu \tilde{\eta}_l)$$

$$\leq \prod_{l=\frac{t}{2}+1}^t (1 - \mu \tilde{\eta}_l) \sum_{m=1}^{\frac{t}{2}} (\tilde{\eta}_m)^2 + \sum_{m=\frac{t}{2}}^t \frac{\tilde{\eta}_m}{\mu} \left(\prod_{l=m+1}^t (1 - \mu \tilde{\eta}_l) - \prod_{l=m}^t (1 - \mu \tilde{\eta}_l)\right)$$

$$\leq \exp\left(-\mu \sum_{l=\frac{t}{2}+1}^t \tilde{\eta}_l\right) \sum_{m=1}^t (\tilde{\eta}_m)^2 + \frac{\tilde{\eta}_{\frac{t}{2}}}{\mu} \sum_{m=\frac{t}{2}}^t \left(\prod_{l=m+1}^t (1 - \mu \tilde{\eta}_l) - \prod_{l=m}^t (1 - \mu \tilde{\eta}_l)\right)$$

$$\leq \exp\left(-\mu c_\eta \frac{t^{1-\alpha} - \left(\frac{t}{2}\right)^{1-\alpha}}{2(1-\alpha)}\right) \sum_{m=1}^t \frac{c_\eta^2}{m^{2\alpha}} + \frac{\tilde{\eta}_{\frac{t}{2}}}{\mu} \left(1 - \prod_{l=\frac{t}{2}+1}^t (1 - \mu \tilde{\eta}_l)\right)$$

$$\leq \exp\left(-\frac{\mu c_\eta t^{1-\alpha}}{2(1-\alpha)}\left(1 - \frac{1}{2^{1-\alpha}}\right)\right) c_\eta^2 \left(1 + \frac{t^{1-2\alpha} - 1}{1 - 2\alpha}\right) + \frac{2c_\eta}{t^\alpha \mu}.$$

*The additional condition on $\alpha$ is obtained by simply taking the limiting case for $t \to \infty$. Also note that this upper bound is tight up to constants (for both terms), especially one could easily show $\sum_{m=1}^t (\tilde{\eta}_m)^2 \prod_{l=m+1}^t (1 - \mu \tilde{\eta}_l) \geq \frac{c_\eta}{2t^\alpha \mu}$.*

**Lemma S49** *For the gamma function $\Gamma(s) = \int_0^\infty y^{s-1} \exp(-y) dy$ we have, $\sum_{t=1}^C \exp\left(-at^b\right) \leq \frac{1}{ba^{1/b}} \Gamma(\frac{1}{b})$.*

**Proof 50** *First we use an integral bound as $\sum_{t=1}^C \exp\left(-at^b\right) \leq \int_0^\infty \exp\left(-az^b\right) dz$, followed by the integral substitution $u = az^b$ after which the proof follows from the definition of the gamma function.*

**Lemma S51** *For the gamma function $\Gamma(s) = \int_0^\infty y^{s-1}\exp(-y)dy$ we have, $\sum_{t=1}^C \frac{\exp(-at^b)}{t^c} \leq \frac{1}{ba^{(1-c)/b}}\Gamma(\frac{1-c}{b})$.*

**Proof 52** *First we use an integral bound as $\sum_{t=1}^C \frac{\exp(-at^b)}{t^c} \leq \int_0^\infty \frac{\exp(-az^b)}{z^c}dz$, followed by the integral substitution $u = az^b$ after which the proof follows from the definition of the gamma function.*

**Lemma S53** *For $a \in (0,1)$, $\sum_{t=1}^C \frac{1}{t^{1-a}} \leq \frac{C^a}{a}$.*

**Proof 54** *It is a simple application of the integral bound on a decreasing function, $\sum_{t=1}^C \frac{1}{t^{1-a}} \leq \int_0^C x^{a-1}dx = \frac{C^a}{a}$.*

**Lemma S55 (Weighted Minkowski)** *For $b_i \in \mathbb{R}$ and $a_i \in \mathbb{R}^d$, we have $\mathbb{E}\left[\left\|\sum_{i=1}^P a_i b_i\right\|^2\right] \leq \left(\sum_{i=1}^P b_i \sqrt{\mathbb{E}\left[\|a_i\|^2\right]}\right)^2$.*

**Proof 56** *We consider again the norm $\|.\|_E = \mathbb{E}\left[\|.\|^2\right]^{\frac{1}{2}}$. Now the above result follows by first applying triangle inequality as $\left\|\sum_{i=1}^P a_i b_i\right\|_E \leq \sum_{i=1}^P \|a_i b_i\|_E$, followed by Holder's inequality to give $\sum_{i=1}^P b_i \|a_i\|_E$.*

### H.2   Proof of Proposition S7 (Mini-batch Averaging Case)

We have the following lemma for mini-batch averaging for the decreasing step-size case,

**Lemma S57** *Under the Assumptions A1, A2, A3, A5, A6 we have for mini-batch averaging,*

$$\mathbb{E}\left[\|\nabla^2 F(\boldsymbol{w}^\star)(\boldsymbol{w} - \boldsymbol{w}^\star)\|^2\right] \leq 5\sum_{i=1}^6 C_{i,P,C}^2.$$

*Where the terms are,*

$$C_{1,P,C}^2 = \frac{1}{C^2 c_\eta^2}\|\boldsymbol{w}^0 - \boldsymbol{w}^\star\|^2,$$

$$C_{2,P,C}^2 = \frac{4}{C^{2-2\alpha}c_\eta^2}\left(\exp\left(-\frac{\mu c_\eta C^{1-\alpha}}{2(1-\alpha)}\right)\|\boldsymbol{w}^0 - \boldsymbol{w}^\star\|^2\right.$$
$$\left. + \frac{2\sigma^2}{P}\left(\exp\left(-\frac{\mu C^{1-\alpha}}{2(1-\alpha)}\left(1 - \frac{1}{2^{1-\alpha}}\right)\right)\frac{2\alpha c_\eta^2}{2\alpha-1} + \frac{2c_\eta}{C^\alpha \mu}\right)\right),$$

$$C_{3,P,C}^2 = \frac{P^2\alpha^2}{T^2 c_\eta^2}\left(\beta_1\|\boldsymbol{w}^0 - \boldsymbol{w}^\star\|^2 + \beta_2\frac{\sigma^2}{P} + \beta_3\frac{\sigma^2 C^\alpha}{P}\right),$$

$$C_{4,P,C}^2 = \frac{P^2 M^2}{T^2}\left(2\beta_1^2\|\boldsymbol{w}^0 - \boldsymbol{w}^\star\|^4 + 2\frac{400\sigma^4}{P^2}\left(\beta_2^2 + \beta_3^2 C^{2-2\alpha}\right)\right),$$

$$C_{5,P,C}^2 = \frac{2\sigma^2}{T} + \frac{2L^2 P}{T^2}\left(\beta_1\|\boldsymbol{w}^0 - \boldsymbol{w}^\star\|^2 + \beta_2\frac{\sigma^2}{P} + \beta_3\frac{\sigma^2 C^{1-\alpha}}{P}\right).$$

*And the constants are,*

$$\beta_1 = \frac{2^{\frac{1+3\alpha}{1-\alpha}}(1-\alpha)^{\frac{4\alpha-2}{1-\alpha}}}{(\mu c_\eta)^{\frac{2\alpha}{1-\alpha}}}\Gamma(\frac{\alpha}{1-\alpha})^2, \beta_2 = \frac{4^{\frac{1+2\alpha-\alpha^2}{(1-\alpha)}}(1-\alpha)^{\frac{2\alpha-1}{(1-\alpha)}}c_\eta^2}{(2\alpha-1)(\mu c_\eta(2^{1-\alpha}-1))^{\frac{2\alpha}{(1-\alpha)}}}\Gamma\left(\frac{\alpha}{1-\alpha}\right)^2, \beta_3 = \frac{32c_\eta}{\alpha^2\mu},$$

$$\beta_4 = \frac{2^{\frac{1}{1-\alpha}}(1-\alpha)^{\frac{\alpha}{1-\alpha}}}{(\mu c_\eta)^{\frac{1}{1-\alpha}}}\Gamma\left(\frac{1}{1-\alpha}\right), \beta_5 = \frac{2^{\frac{3-2\alpha}{1-\alpha}}(1-\alpha)^{\frac{\alpha}{1-\alpha}}\alpha c_\eta^2}{(2\alpha-1)(\mu c_\eta(2^{1-\alpha}-1))^{\frac{1}{1-\alpha}}}\Gamma\left(\frac{1}{1-\alpha}\right), \beta_6 = \frac{2c_\eta}{(1-\alpha)\mu}.$$

**Proof 58** *Using again the decomposition in Lemma S31, we can obtain the following simpler version for mini-batch averaging,*

$$F''(\boldsymbol{w}^\star)(\overline{\overline{\boldsymbol{w}}}^C - \boldsymbol{w}^\star) = \frac{\boldsymbol{w}^0 - \boldsymbol{w}^\star}{C\eta_1^1} - \frac{\hat{\boldsymbol{w}}^C - \boldsymbol{w}^\star}{C\eta_2^C} - \frac{1}{T}\sum_{t=1}^{C}\sum_{i=1}^{P}\left(\boldsymbol{w}_{i,1}^t - \boldsymbol{w}^\star\right)\left(\frac{1}{\eta_1^t} - \frac{1}{\eta_2^t}\right)$$

$$+ \frac{1}{T}\sum_{t=1}^{C}\sum_{i=1}^{P}\delta_{i,1}^t + \frac{1}{T}\sum_{t=1}^{C}\sum_{i=1}^{P}\xi_{i,1}^t.$$

*Note again that we assume $\alpha \in (\frac{1}{2}, 1)$, just for the sake of brevity. For the first term,*

$$\mathbb{E}\left[\left\|\frac{\boldsymbol{w}^0 - \boldsymbol{w}^\star}{C\eta_1^1}\right\|^2\right] = \frac{1}{C^2 c_\eta^2}\|\boldsymbol{w}^0 - \boldsymbol{w}^\star\|^2 = C_{1,P,C}^2.$$

*For the second term using Lemma S9, followed by Lemma S45 and Lemma S47 we obtain,*

$$\mathbb{E}\left[\left\|\frac{\hat{\boldsymbol{w}}^C - \boldsymbol{w}^\star}{C\eta_2^C}\right\|^2\right] = \frac{(C+1)^{2\alpha}}{C^2 c_\eta^2}\mathbb{E}\left[\|\boldsymbol{w}_{MB}^C - \boldsymbol{w}^\star\|^2\right]$$

$$\leq \frac{2^{2\alpha}}{C^{2-2\alpha}c_\eta^2}\left(\prod_{m=1}^{C}(1-\mu\tilde{\eta}_m)\mathbb{E}\left[\|\boldsymbol{w}^0 - \boldsymbol{w}^\star\|^2\right] + 2\sigma^2\frac{1}{P}\sum_{m=1}^{C}(\tilde{\eta}_m)^2\prod_{l=m+1}^{C}(1-\mu\tilde{\eta}_l)\right)$$

$$\leq \frac{4}{C^{2-2\alpha}c_\eta^2}\left(\exp\left(-\frac{\mu c_\eta C^{1-\alpha}}{2(1-\alpha)}\right)\|\boldsymbol{w}^0 - \boldsymbol{w}^\star\|^2\right.$$

$$\left. + \frac{2\sigma^2}{P}\left(\exp\left(-\frac{\mu C^{1-\alpha}}{2(1-\alpha)}\left(1 - \frac{1}{2^{1-\alpha}}\right)\right)\frac{2\alpha c_\eta^2}{2\alpha - 1} + \frac{2c_\eta}{C^\alpha \mu}\right)\right) = C_{2,P,C}^2$$

*For the third term using Lemma S55 and $(t+1)^\alpha - t^\alpha \leq \alpha t^{\alpha-1}$,*

$$\mathbb{E}\left[\left\|\frac{1}{T}\sum_{t=1}^{C}\sum_{i=1}^{P}\left(\boldsymbol{w}_{i,1}^t - \boldsymbol{w}^\star\right)\left(\frac{1}{\eta_1^t} - \frac{1}{\eta_2^t}\right)\right\|^2\right]$$

$$\leq \frac{1}{T^2 c_\eta^2}\mathbb{E}\left[\left\|\sum_{t=1}^{C}\sum_{i=1}^{P}\left(\boldsymbol{w}_{i,1}^t - \boldsymbol{w}^\star\right)\left((t+1)^\alpha - t^\alpha\right)\right\|^2\right]$$

$$\leq \frac{P^2\alpha^2}{T^2 c_\eta^2}\left(\sum_{t=1}^{C}\left((t+1)^\alpha - t^\alpha\right)\sqrt{\mathbb{E}\left[\left\|\sum_{i=1}^{P}\left(\boldsymbol{w}_{i,1}^t - \boldsymbol{w}^\star\right)\right\|^2\right]}\right)^2$$

$$\leq \frac{P^2\alpha^2}{T^2 c_\eta^2}\left(\sum_{t=1}^{C}t^{\alpha-1}\sqrt{\mathbb{E}\left[\|\boldsymbol{w}_{MB}^t - \boldsymbol{w}^\star\|^2\right]}\right)^2.$$

*Now using Lemma S9, Lemma S45, Lemma S47 and $\sqrt{a+b} \leq \sqrt{a} + \sqrt{b}$ we get,*

$$\mathbb{E}\left[\left\|\frac{1}{T}\sum_{t=1}^{C}\sum_{i=1}^{P}\left(\boldsymbol{w}_{i,1}^t - \boldsymbol{w}^\star\right)\left(\frac{1}{\eta_1^t} - \frac{1}{\eta_2^t}\right)\right\|^2\right]$$

$$\leq \frac{P^2\alpha^2}{T^2 c_\eta^2}\left(\sum_{t=1}^{C}t^{\alpha-1}\sqrt{\prod_{m=1}^{t}(1-\mu\tilde{\eta}_m)\|\boldsymbol{w}^0 - \boldsymbol{w}^\star\|^2 + 2\sigma^2\frac{1}{P}\sum_{m=1}^{t}(\tilde{\eta}_m)^2\prod_{l=m+1}^{t}(1-\mu\tilde{\eta}_l)}\right)^2$$

$$\leq \frac{P^2\alpha^2}{T^2 c_\eta^2}\left(\sum_{t=1}^{C}t^{\alpha-1}\sqrt{\exp\left(-\frac{\mu c_\eta t^{1-\alpha}}{2(1-\alpha)}\right)\|\boldsymbol{w}^0 - \boldsymbol{w}^\star\|^2 + \frac{2\sigma^2}{P}\left(\exp\left(-\frac{\mu c_\eta t^{1-\alpha}}{2(1-\alpha)}\left(1 - \frac{1}{2^{1-\alpha}}\right)\right)\frac{2\alpha c_\eta^2}{2\alpha - 1} + \frac{2c_\eta}{t^\alpha \mu}\right)}\right)^2$$

$$\leq \frac{P^2\alpha^2}{T^2c_\eta^2}\left(\sum_{t=1}^{C}t^{\alpha-1}\left(\exp\left(-\frac{\mu c_\eta t^{1-\alpha}}{4(1-\alpha)}\right)\left\|\boldsymbol{w}^0-\boldsymbol{w}^\star\right\|+\sqrt{\frac{2\sigma^2}{P}\exp\left(-\frac{\mu c_\eta t^{1-\alpha}}{2(1-\alpha)}\left(1-\frac{1}{2^{1-\alpha}}\right)\right)\frac{2\alpha c_\eta^2}{2\alpha-1}}\right.\right.$$

$$\left.\left.+\sqrt{\frac{4c_\eta\sigma^2}{Pt^\alpha\mu}}\right)\right)^2$$

$$\leq \frac{P^2\alpha^2}{T^2c_\eta^2}\left(\sum_{t=1}^{C}t^{\alpha-1}\exp\left(-\frac{\mu c_\eta t^{1-\alpha}}{4(1-\alpha)}\right)\left\|\boldsymbol{w}^0-\boldsymbol{w}^\star\right\|+\sum_{t=1}^{C}t^{\alpha-1}\sqrt{\frac{2\sigma^2 c_\eta^2}{P(2\alpha-1)}\exp\left(-\frac{\mu c_\eta t^{1-\alpha}}{2(1-\alpha)}\left(1-\frac{1}{2^{1-\alpha}}\right)\right)}\right.$$

$$\left.+\sum_{t=1}^{C}t^{\frac{\alpha}{2}-1}\sqrt{\frac{4c_\eta\sigma^2}{P\mu}}\right)^2$$

$$\leq \frac{P^2\alpha^2}{T^2c_\eta^2}\left(\sum_{t=1}^{C}t^{\alpha-1}\exp\left(-\frac{\mu c_\eta t^{1-\alpha}}{4(1-\alpha)}\right)\left\|\boldsymbol{w}^0-\boldsymbol{w}^\star\right\|+\sqrt{\frac{2\sigma^2 c_\eta^2}{P(2\alpha-1)}}\sum_{t=1}^{C}t^{\alpha-1}\exp\left(-\frac{\mu c_\eta t^{1-\alpha}}{4(1-\alpha)}\left(1-\frac{1}{2^{1-\alpha}}\right)\right)\right.$$

$$\left.+\sqrt{\frac{4c_\eta\sigma^2}{P\mu}}\sum_{t=1}^{C}\frac{1}{t^{1-\frac{\alpha}{2}}}\right)^2.$$

*Now using Lemma S51 (with $b=1-\alpha$, $c=1-\alpha$ and $a=\frac{\mu c_\eta}{4(1-\alpha)}$), followed by using Lemma S51 again (with $a=\frac{\mu c_\eta}{4(1-\alpha)}\left(1-\frac{1}{2^{1-\alpha}}\right)$, $b=1-\alpha$ and $c=1-\alpha$) and Lemma S53 (with $a=\frac{\alpha}{2}$) we get,*

$$\mathbb{E}\left[\left\|\frac{1}{T}\sum_{t=1}^{C}\sum_{i=1}^{P}\left(\boldsymbol{w}_{i,1}^t-\boldsymbol{w}^\star\right)\left(\frac{1}{\eta_1^t}-\frac{1}{\eta_2^t}\right)\right\|^2\right]$$

$$\leq \frac{P^2\alpha^2}{T^2c_\eta^2}\left(\frac{4^{\frac{\alpha}{1-\alpha}}(1-\alpha)^{\frac{2\alpha-1}{1-\alpha}}}{(\mu c_\eta)^{\frac{\alpha}{1-\alpha}}}\Gamma(\frac{\alpha}{1-\alpha})\left\|\boldsymbol{w}^0-\boldsymbol{w}^\star\right\|+\sqrt{\frac{2\sigma^2 c_\eta^2}{P(2\alpha-1)}}\frac{2^{\frac{\alpha(3-\alpha)}{1-\alpha}}(1-\alpha)^{\frac{2\alpha-1}{1-\alpha}}}{(\mu c_\eta(2^{1-\alpha}-1))^{\frac{\alpha}{1-\alpha}}}\Gamma(\frac{\alpha}{1-\alpha})\right.$$

$$\left.+\sqrt{\frac{4c_\eta\sigma^2}{P\mu}}\frac{2C^{\frac{\alpha}{2}}}{\alpha}\right)^2.$$

*Finally using Lemma S35 and re-organizing with constants defined as above,*

$$\mathbb{E}\left[\left\|\frac{1}{T}\sum_{t=1}^{C}\sum_{i=1}^{P}\left(\boldsymbol{w}_{i,1}^t-\boldsymbol{w}^\star\right)\left(\frac{1}{\eta_1^t}-\frac{1}{\eta_2^t}\right)\right\|^2\right]$$

$$\leq \frac{P^2\alpha^2}{T^2c_\eta^2}\left(2\frac{4^{\frac{2\alpha}{1-\alpha}}(1-\alpha)^{\frac{4\alpha-2}{1-\alpha}}}{(\mu c_\eta)^{\frac{2\alpha}{1-\alpha}}}\Gamma(\frac{\alpha}{1-\alpha})^2\left\|\boldsymbol{w}^0-\boldsymbol{w}^\star\right\|^2+2\frac{2\sigma^2 c_\eta^2}{P(2\alpha-1)}\frac{4^{\frac{\alpha(3-\alpha)}{(1-\alpha)}}(1-\alpha)^{\frac{2\alpha-1}{(1-\alpha)}}}{(\mu c_\eta(2^{1-\alpha}-1))^{\frac{2\alpha}{(1-\alpha)}}}\Gamma\left(\frac{\alpha}{1-\alpha}\right)^2\right.$$

$$\left.+2\frac{4c_\eta\sigma^2}{P\mu}\frac{4C^\alpha}{\alpha^2}\right)$$

$$\leq \frac{P^2\alpha^2}{T^2c_\eta^2}\left(\beta_1\left\|\boldsymbol{w}^0-\boldsymbol{w}^\star\right\|^2+\beta_2\frac{\sigma^2}{P}+\beta_3\frac{\sigma^2 C^\alpha}{P}\right)=C_{3,P,C}^2.$$

*For the fourth term first proceeding as in Lemma S39 with Lemma S35 and Lemma S37 we can obtain,*

$$\mathbb{E}\left[\left\|\frac{1}{T}\sum_{t=1}^{C}\sum_{i=1}^{P}\delta_{i,1}^t\right\|^2\right]=\frac{1}{T^2}\mathbb{E}\left[\left\|\sum_{t=1}^{C}\sum_{i=1}^{P}\left(F'(\boldsymbol{w}_{i,0}^t)-F''(\boldsymbol{w}^\star)(\boldsymbol{w}_{i,0}^t-\boldsymbol{w}^\star)\right)\right\|^2\right]$$

$$\leq \frac{P}{T^2}\sum_{i=1}^{P}\mathbb{E}\left[\left\|\sum_{t=1}^{C}\left(F'(\hat{\boldsymbol{w}}^{t-1})-F''(\boldsymbol{w}^\star)(\hat{\boldsymbol{w}}^{t-1}-\boldsymbol{w}^\star)\right)\right\|^2\right]$$

$$\leq \frac{P}{T^2}\sum_{i=1}^{P}\left(\sum_{t=1}^{C}\sqrt{\mathbb{E}\left[\left\|(F'(\hat{\boldsymbol{w}}^{t-1})-F''(\boldsymbol{w}^\star)(\hat{\boldsymbol{w}}^{t-1}-\boldsymbol{w}^\star))\right\|^2\right]}\right)^2$$

$$\leq \frac{PM^2}{T^2} \sum_{i=1}^{P} \left( \sum_{t=1}^{C} \sqrt{\mathbb{E}\left[(\hat{\boldsymbol{w}}^{t-1} - \boldsymbol{w}^\star)^4\right]} \right)^2$$

$$\leq \frac{P^2 M^2}{T^2} \left( \sum_{t=1}^{C} \sqrt{\mathbb{E}\left[(\boldsymbol{w}_{MB}^{t-1} - \boldsymbol{w}^\star)^4\right]} \right)^2.$$

*Now using Lemma S26, followed by Lemma S45 and Lemma S47 we get[8],*

$$\mathbb{E}\left[\left\|\frac{1}{T}\sum_{t=1}^{C}\sum_{i=1}^{P}\delta_{i,1}^{t}\right\|^2\right] \leq \frac{P^2 M^2}{T^2}\left( \sum_{t=1}^{C}\left( \prod_{j=1}^{t-1}(1 - \tilde{\eta}_j \mu) \left\|\boldsymbol{w}^0 - \boldsymbol{w}^\star\right\|^2 + \frac{20\sigma^2}{P}\sum_{j=1}^{t-1}(\tilde{\eta}_j)^2\prod_{l=j+1}^{t-1}(1 - \mu\tilde{\eta}_l)\right)\right)^2$$

$$\leq \frac{P^2 M^2}{T^2}\Bigg( \sum_{t=1}^{C}\exp\left(-\frac{\mu c_\eta(t-1)^{1-\alpha}}{2(1-\alpha)}\right)\left\|\boldsymbol{w}^0 - \boldsymbol{w}^\star\right\|^2$$

$$+ \sum_{t=2}^{C}\frac{20\sigma^2}{P}\left(\exp\left(-\frac{\mu c_\eta(t-1)^{1-\alpha}}{2(1-\alpha)}\left(1 - \frac{1}{2^{1-\alpha}}\right)\right)\frac{2\alpha c_\eta^2}{2\alpha - 1} + \frac{2c_\eta}{(t-1)^\alpha \mu}\right)\Bigg)^2$$

$$\leq \frac{P^2 M^2}{T^2}\Bigg( \sum_{t=1}^{C}\exp\left(-\frac{\mu c_\eta(t-1)^{1-\alpha}}{2(1-\alpha)}\right)\left\|\boldsymbol{w}^0 - \boldsymbol{w}^\star\right\|^2$$

$$+ \sum_{t=1}^{C}\frac{20\sigma^2}{P}\left(\exp\left(-\frac{\mu c_\eta t^{1-\alpha}}{2(1-\alpha)}\left(1 - \frac{1}{2^{1-\alpha}}\right)\right)\frac{2\alpha c_\eta^2}{2\alpha - 1} + \sum_{t=1}^{C}\frac{2c_\eta}{t^\alpha \mu}\right)\Bigg)^2.$$

*Now using Lemma S49 (with $b = 1 - \alpha$ and $a = \frac{\mu c_\eta}{2(1-\alpha)}$), followed by Lemma S49 again (with $a = \frac{\mu c_\eta}{2(1-\alpha)}\left(1 - \frac{1}{2^{1-\alpha}}\right)$ and $b = 1 - \alpha$), followed by Lemma S53 (with $a = 1 - \alpha$) and Lemma S35 we get,*

$$\mathbb{E}\left[\left\|\frac{1}{T}\sum_{t=1}^{C}\sum_{i=1}^{P}\delta_{i,1}^{t}\right\|^2\right]$$

$$\leq \frac{P^2 M^2}{T^2}\Bigg( \frac{2^{\frac{1}{1-\alpha}}(1-\alpha)^{\frac{\alpha}{1-\alpha}}}{(\mu c_\eta)^{\frac{1}{1-\alpha}}}\Gamma\left(\frac{1}{1-\alpha}\right)\left\|\boldsymbol{w}^0 - \boldsymbol{w}^\star\right\|^2$$

$$+ \frac{20\sigma^2}{P}\left(\frac{2^{\frac{2-\alpha}{1-\alpha}}(1-\alpha)^{\frac{\alpha}{1-\alpha}}}{(\mu c_\eta(2^{1-\alpha}-1))^{\frac{1}{1-\alpha}}}\Gamma\left(\frac{1}{1-\alpha}\right)\frac{2\alpha c_\eta^2}{2\alpha - 1} + \frac{2c_\eta C^{1-\alpha}}{(1-\alpha)\mu}\right)\Bigg)^2$$

$$\leq \frac{P^2 M^2}{T^2}\Bigg( 2\frac{2^{\frac{2}{1-\alpha}}(1-\alpha)^{\frac{2\alpha}{1-\alpha}}}{(\mu c_\eta)^{\frac{2}{1-\alpha}}}\Gamma\left(\frac{1}{1-\alpha}\right)^2\left\|\boldsymbol{w}^0 - \boldsymbol{w}^\star\right\|^4$$

$$+ 2\frac{400\sigma^4}{P^2}\left(\frac{2^{\frac{4-2\alpha}{1-\alpha}}(1-\alpha)^{\frac{2\alpha}{1-\alpha}}}{(\mu c_\eta(2^{1-\alpha}-1))^{\frac{2}{1-\alpha}}}\Gamma\left(\frac{1}{1-\alpha}\right)^2\frac{4\alpha^2 c_\eta^4}{(2\alpha - 1)^2} + \frac{4c_\eta^2 C^{2-2\alpha}}{(1-\alpha)^2\mu^2}\right)\Bigg)$$

*Bounding again with the constants defined above,*

$$\mathbb{E}\left[\left\|\frac{1}{T}\sum_{t=1}^{C}\sum_{i=1}^{P}\delta_{i,1}^{t}\right\|^2\right] \leq \frac{P^2 M^2}{T^2}\left(2\beta_4^2\left\|\boldsymbol{w}^0 - \boldsymbol{w}^\star\right\|^4 + 2\frac{400\sigma^4}{P^2}\left(\beta_5^2 + \beta_6^2 C^{2-2\alpha}\right)\right) = C_{4,P,C}^2.$$

*For the fifth term, proceeding as in Lemma S39,*

$$\mathbb{E}\left[\left\|\frac{1}{T}\sum_{t=1}^{C}\sum_{i=1}^{P}\xi_{i,1}^{t}\right\|^2\right] = \frac{1}{T^2}\sum_{t=1}^{C}\sum_{i=1}^{P}\left(2L^2\mathbb{E}\left[\left\|\boldsymbol{w}_{i,0}^{t} - \boldsymbol{w}^\star\right\|^2\right] + 2\sigma^2\right)$$

$$\leq \frac{2\sigma^2}{T} + \frac{2L^2 P}{T^2} \sum_{t=1}^{C} \mathbb{E}\left[\left\|\boldsymbol{w}_{1,0}^t - \boldsymbol{w}^\star\right\|^2\right]$$

$$\leq \frac{2\sigma^2}{T} + \frac{2L^2 P}{T^2} \sum_{t=1}^{C} \mathbb{E}\left[\left\|\hat{\boldsymbol{w}}_{MB}^{t-1} - \boldsymbol{w}^\star\right\|^2\right].$$

*Now using Lemma S9, Lemma S45 and Lemma S47 like before,*

$$\mathbb{E}\left[\left\|\frac{1}{T}\sum_{t=1}^{C}\sum_{i=1}^{P}\xi_{i,1}^t\right\|^2\right] \leq \frac{2\sigma^2}{T} + \frac{2L^2 P}{T^2} \sum_{t=1}^{C}\left(\exp\left(-\frac{\mu c_\eta}{2(1-\alpha)}t^{1-\alpha}\right)\left\|\boldsymbol{w}^0 - \boldsymbol{w}^\star\right\|^2\right.$$
$$\left. + \frac{2\sigma^2}{P}\exp\left(-\frac{\mu c_\eta t^{1-\alpha}}{2(1-\alpha)}\left(1 - \frac{1}{2^{1-\alpha}}\right)\right)\frac{2\alpha c_\eta^2}{2\alpha - 1} + \frac{4\sigma^2 c_\eta}{P t^\alpha \mu}\right).$$

*Further using Lemma S49 (with $b = 1 - \alpha$ and $a = \frac{\mu c_\eta}{2(1-\alpha)}$), followed by Lemma S49 again (with $a = \frac{\mu c_\eta}{2(1-\alpha)}\left(1 - \frac{1}{2^{1-\alpha}}\right)$ and $b = 1 - \alpha$), followed by Lemma S53 (with $a = 1 - \alpha$) and the constants as used above we get,*

$$\mathbb{E}\left[\left\|\frac{1}{T}\sum_{t=1}^{C}\sum_{i=1}^{P}\xi_{i,1}^t\right\|^2\right] \leq \frac{2\sigma^2}{T} + \frac{2L^2 P}{T^2}\left(\frac{2^{\frac{1}{1-\alpha}}(1-\alpha)^{\frac{\alpha}{1-\alpha}}}{(\mu c_\eta)^{\frac{1}{1-\alpha}}}\Gamma\left(\frac{1}{1-\alpha}\right)\left\|\boldsymbol{w}^0 - \boldsymbol{w}^\star\right\|^2\right.$$
$$\left. + \frac{2^{\frac{2-\alpha}{1-\alpha}}(1-\alpha)^{\frac{\alpha}{1-\alpha}}}{(\mu c_\eta(2^{1-\alpha}-1))^{\frac{1}{1-\alpha}}}\Gamma\left(\frac{1}{1-\alpha}\right)\frac{2\alpha c_\eta^2}{2\alpha - 1} + \frac{2c_\eta C^{1-\alpha}}{(1-\alpha)\mu}\right)$$
$$\leq \frac{2\sigma^2}{T} + \frac{2L^2 P}{T^2}\left(\beta_4 \left\|\boldsymbol{w}^0 - \boldsymbol{w}^\star\right\|^2 + \beta_5 \frac{\sigma^2}{P} + \beta_6 \frac{\sigma^2 C^{1-\alpha}}{P}\right) = C_{5,P,C}^2.$$

*Finally using Lemma S35 we have proved the lemma.*

The following lemma separates the terms above into bias and variance terms, following which we can easily prove Proposition S7,

**Lemma S59** *Under the Assumptions A1, A2, A3, A5, A6 we have for mini-batch averaging,*

$$\mathbb{E}\left[\left\|\nabla^2 F(\boldsymbol{w}^\star)(\boldsymbol{w} - \boldsymbol{w}^\star)\right\|^2\right] \leq 5\left(\hat{C}_{1,P,C}^2 + \hat{C}_{2,P,C}^2\right)$$

*Where for constants defined as above the terms are,*

$$\hat{C}_{1,P,C}^2 = \frac{\left\|\boldsymbol{w}^0 - \boldsymbol{w}^\star\right\|^2}{C^2 c_\eta^2}\left(1 + 4C^{2\alpha}\exp\left(-\frac{\mu c_\eta C^{1-\alpha}}{2(1-\alpha)}\right) + \alpha^2\beta_1 + 2M^2 c_\eta^2 \beta_1^2 \left\|\boldsymbol{w}^0 - \boldsymbol{w}^\star\right\|^2 + \frac{2L^2\beta_1 c_\eta^2}{P}\right),$$

$$\hat{C}_{2,P,C}^2 = \frac{2\sigma^2}{T}\left(1 + \frac{8\alpha C^{2\alpha-1}}{2\alpha - 1}\exp\left(-\frac{\mu C^{1-\alpha}}{2(1-\alpha)}\left(1 - \frac{1}{2^{1-\alpha}}\right)\right) + \frac{8}{C^{1-\alpha}c_\eta\mu} + \frac{\alpha^2\beta_2}{2Cc_\eta^2} + \frac{\alpha^2\beta_3}{2C^{1-\alpha}c_\eta^2}\right.$$
$$\left. + \frac{400M^2\sigma^2}{T}\left(\beta_2^2 + \beta_3^2 C^{2-2\alpha}\right) + \frac{L^2}{T}\left(\beta_2 + \beta_3 C^{1-\alpha}\right)\right).$$

To get Proposition S7, we upper bound every term up to constants depending only on $\alpha$. Specifically, we use $\beta_1 \precsim (\mu c_\eta)^{-\frac{1}{1-\alpha}}$, $\beta_2 \precsim (\mu c_\eta)^{-\frac{\alpha}{1-\alpha}}$, and $\beta_3 \precsim \frac{c_\eta}{\mu}$.

## H.3 Proof of Proposition S7 (One-shot Averaging case)

The analysis for the one-shot case is very similar to the mini-batch case, just like the constant step-size case. In fact at many place the communications $C$ of MBA get replaced by $N^1$ and the form of the bound remains the same. This intuitive conversion strengthens our analysis, which smoothly extends to both the extreme cases.

**Lemma S60** *Under the Assumptions  A1, A2, A3,  A5, A6 for decreasing step size, for one shot averaging we have,*

$$\mathbb{E}\left[\left\|\nabla^2 F(\boldsymbol{w}^\star)(\boldsymbol{w}^1_{i,k} - \boldsymbol{w}^\star)\right\|^2\right] \leq 5\sum_{i=1}^{6} D^2_{i,P,C}$$

*where the terms are,*

$$D^2_{1,P,N^1} = \frac{P^2}{T^2 c_\eta^2}\left\|\boldsymbol{w}^0 - \boldsymbol{w}^\star\right\|^2, D^2_{2,P,N^1} = \frac{4}{(N^1)^{2-2\alpha}c_\eta^2}\left(\exp\left(-\frac{\mu c_\eta (N^1)^{1-\alpha}}{1-\alpha}\right)\left\|\boldsymbol{w}^0 - \boldsymbol{w}^\star\right\|^2 + \frac{2\sigma^2 c_\eta}{\mu}\right),$$

$$D^2_{3,P,N^1} = \frac{P^2 \alpha^2}{T^2 c_\eta^2}\left(4\beta^2\left\|\boldsymbol{w}^0 - \boldsymbol{w}^\star\right\|^2 + \frac{2\sigma^2 (N^1)^{2\alpha} c_\eta}{\mu\alpha^2}\right), D^2_{4,P,N^1} = \frac{P^2 M^2}{T^2}\left(\beta\left\|\boldsymbol{w}^0 - \boldsymbol{w}^\star\right\|^2 + \frac{20\sigma^2 N^1 c_\eta}{\mu}\right)^2,$$

$$D^2_{5,P,N^1} = \frac{2\sigma^2}{T}, D^2_{6,P,N^1} = \frac{2L^2 P}{T^2}\left(\beta\left\|\boldsymbol{w}^0 - \boldsymbol{w}^\star\right\|^2 + \frac{2\sigma^2 N^1 c_\eta}{\mu}\right).$$

*And the constants are* $\beta_1 = 1 + \left(\frac{(1-\alpha)^\alpha}{\mu c_\eta}\right)^{\frac{1}{1-\alpha}}\Gamma\left(\frac{1}{1-\alpha}\right)$ *and* $\beta_2 = \left(2^\alpha \frac{(1-\alpha)^{2\alpha-1}}{(\mu c_\eta)^\alpha}\right)^{\frac{1}{1-\alpha}}\Gamma\left(\frac{\alpha}{1-\alpha}\right).$

**Proof 61** *We follow an analysis similar to [15]. We can simplify the decomposition from Lemma S31 for one outer phase as follows,*

$$F''(\boldsymbol{w}^\star)(\overline{\overline{\boldsymbol{w}}}^C - \boldsymbol{w}^\star) = \frac{\boldsymbol{w}^0 - \boldsymbol{w}^\star}{N^1 \eta_1^1} - \frac{\hat{\boldsymbol{w}}^1 - \boldsymbol{w}^\star}{N^1 \eta_{N^1+1}^1} - \frac{1}{T}\sum_{i=1}^{P}\sum_{k=1}^{N^1}(\boldsymbol{w}^1_{i,k} - \boldsymbol{w}^\star)\left(\frac{1}{\eta_k^1} - \frac{1}{\eta_{k+1}^1}\right)$$

$$+ \frac{1}{T}\sum_{k=1}^{N^1}\sum_{i=1}^{P}\delta^1_{i,k} + \frac{1}{T}\sum_{k=1}^{N^1}\sum_{i=1}^{P}\xi^1_{i,k}.$$

*For the first term,*

$$\mathbb{E}\left[\left\|\frac{\boldsymbol{w}^0 - \boldsymbol{w}^\star}{N^1 \eta_1^1}\right\|^2\right] \leq \frac{P^2}{T^2 c_\eta^2}\left\|\boldsymbol{w}^0 - \boldsymbol{w}^\star\right\|^2 = D^2_{1,P,N^1}.$$

*For the second term note that the inner iterate bound is independent for different machines using Lemma S11 for say machine* 1, *followed by Lemma S45 and Lemma S47 we get,*

$$\mathbb{E}\left[\left\|\frac{\hat{\boldsymbol{w}}^1 - \boldsymbol{w}^\star}{N^1 \eta_{N^1+1}^1}\right\|^2\right] \leq \frac{(N^1+1)^{2\alpha}}{(N^1)^2 c_\eta^2}\mathbb{E}\left[\left\|\frac{1}{P}\sum_{i=1}^{P}(\boldsymbol{w}^1_{i,N^1} - \boldsymbol{w}^\star)\right\|^2\right]$$

$$\leq \frac{2^{2\alpha}}{(N^1)^{2-2\alpha}c_\eta^2}\mathbb{E}\left[\left\|\boldsymbol{w}^1_{1,N^1} - \boldsymbol{w}^\star\right\|^2\right]$$

$$\leq \frac{4}{(N^1)^{2-2\alpha}c_\eta^2}\left(\prod_{m=1}^{N^1}(1-\mu\eta_m^1)\left\|\boldsymbol{w}^0 - \boldsymbol{w}^\star\right\|^2 + 2\sigma^2\sum_{m=1}^{N^1}(\eta_m^1)^2\prod_{l=m+1}^{N^1}(1-\mu\eta_l^1)\right)$$

$$\leq \frac{4}{(N^1)^{2-2\alpha}c_\eta^2}\left(\exp\left(-\frac{\mu c_\eta (N^1)^{1-\alpha}}{1-\alpha}\right)\left\|\boldsymbol{w}^0 - \boldsymbol{w}^\star\right\|^2 + \frac{2\sigma^2 c_\eta}{\mu}\right) = D^2_{2,P,N^1}.$$

*For the third term using* $(k+1)^\alpha - k^\alpha \leq \alpha k^{\alpha-1}$, *Lemma S55, and noting that the individual bounds on inner iterates for different machines are the same, thus using machine* 1 *for brevity we can obtain,*

$$\mathbb{E}\left[\left\|\frac{1}{T}\sum_{i=1}^{P}\sum_{k=1}^{N^1}(\boldsymbol{w}^1_{i,k} - \boldsymbol{w}^\star)\left(\frac{1}{\eta_k^1} - \frac{1}{\eta_{k+1}^1}\right)\right\|^2\right] \leq \frac{P^2 \alpha^2}{T^2 c_\eta^2}\mathbb{E}\left[\left\|\sum_{k=1}^{N^1}k^{\alpha-1}(\boldsymbol{w}^1_{1,k} - \boldsymbol{w}^\star)\right\|^2\right]$$

$$\leq \frac{P^2 \alpha^2}{T^2 c_\eta^2}\left(\sum_{k=1}^{N^1}k^{\alpha-1}\sqrt{\mathbb{E}\left[\left\|\boldsymbol{w}^1_{1,k} - \boldsymbol{w}^\star\right\|^2\right]}\right)^2.$$

*Now using Lemma S11, Lemma S45, Lemma S47 and $\sqrt{a+b} \leq \sqrt{a} + \sqrt{b}$ we get,*

$$\mathbb{E}\left[\left\|\frac{1}{T}\sum_{k=1}^{N^1}\sum_{i=1}^{P}\left(\boldsymbol{w}_{i,k}^1 - \boldsymbol{w}^\star\right)\left(\frac{1}{\eta_k^1} - \frac{1}{\eta_{k+1}^1}\right)\right\|^2\right]$$

$$\leq \frac{P^2\alpha^2}{T^2 c_\eta^2}\left(\sum_{k=1}^{N^1} k^{\alpha-1}\sqrt{\mathbb{E}\left[\prod_{m=1}^k (1-\mu\tilde{\eta}_m)\left\|\boldsymbol{w}^0 - \boldsymbol{w}^\star\right\|^2 + 2\sigma^2\sum_{m=1}^k(\tilde{\eta}_m)^2\prod_{l=m+1}^k(1-\mu\tilde{\eta}_l)\right]}\right)^2$$

$$\leq \frac{P^2\alpha^2}{T^2 c_\eta^2}\left(\sum_{k=1}^{N^1} k^{\alpha-1}\sqrt{\exp\left(-\frac{\mu c_\eta k^{1-\alpha}}{1-\alpha}\right)\left\|\boldsymbol{w}^0 - \boldsymbol{w}^\star\right\|^2 + \frac{2\sigma^2 c_\eta}{\mu}}\right)^2$$

$$\leq \frac{P^2\alpha^2}{T^2 c_\eta^2}\left(\sum_{k=1}^{N^1} k^{\alpha-1}\left(\exp\left(-\frac{\mu c_\eta k^{1-\alpha}}{2(1-\alpha)}\right)\left\|\boldsymbol{w}^0 - \boldsymbol{w}^\star\right\| + \sqrt{\frac{2\sigma^2 c_\eta}{\mu}}\right)\right)^2.$$

*Now using Lemma S51 again with $b = 1-\alpha$ and $a = \frac{\mu c_\eta}{2(1-\alpha)}$ with $\beta_2$ defined as above and Lemma S53 we get,*

$$\mathbb{E}\left[\left\|\frac{1}{T}\sum_{k=1}^{N^1}\sum_{i=1}^{P}\left(\boldsymbol{w}_{i,k}^1 - \boldsymbol{w}^\star\right)\left(\frac{1}{\eta_k^1} - \frac{1}{\eta_{k+1}^1}\right)\right\|^2\right]$$

$$\leq \frac{P^2\alpha^2}{T^2 c_\eta^2}\left(\left(2^\alpha\frac{(1-\alpha)^{2\alpha-1}}{(\mu c_\eta)^\alpha}\right)^{\frac{1}{1-\alpha}}\Gamma\left(\frac{\alpha}{1-\alpha}\right)\left\|\boldsymbol{w}^0 - \boldsymbol{w}^\star\right\| + \sqrt{\frac{2\sigma^2(N^1)^{2\alpha}c_\eta}{\mu\alpha^2}}\right)^2$$

$$\leq \frac{P^2\alpha^2}{T^2 c_\eta^2}\left(\beta_2\left\|\boldsymbol{w}^0 - \boldsymbol{w}^\star\right\| + \sqrt{\frac{2\sigma^2(N^1)^{2\alpha}c_\eta}{P\mu\alpha^2}}\right)^2$$

$$\leq \frac{P^2\alpha^2}{T^2 c_\eta^2}\left(2\beta_2^2\left\|\boldsymbol{w}^0 - \boldsymbol{w}^\star\right\|^2 + \frac{4\sigma^2(N^1)^{2\alpha}c_\eta}{\mu\alpha^2}\right) = D_{3,P,N^1}^2.$$

*Now for the fourth term proceeding as in Lemma S42 with Lemma S35 and Lemma S37 we can obtain ,*

$$\mathbb{E}\left[\left\|\frac{1}{T}\sum_{i=1}^{P}\sum_{k=1}^{N^1}\delta_{i,k}^1\right\|^2\right] = \frac{1}{T^2}\mathbb{E}\left[\left\|\sum_{i=1}^{P}\sum_{k=1}^{N^1}F'(\boldsymbol{w}_{i,k-1}^t) - F''(\boldsymbol{w}^\star)(\boldsymbol{w}_{i,k-1}^t - \boldsymbol{w}^\star)\right\|^2\right]$$

$$\leq \frac{P}{T^2}\sum_{i=1}^{P}\mathbb{E}\left[\left\|\sum_{k=1}^{N^1}F'(\boldsymbol{w}_{i,k-1}^t) - F''(\boldsymbol{w}^\star)(\boldsymbol{w}_{i,k-1}^t - \boldsymbol{w}^\star)\right\|^2\right]$$

$$\leq \frac{P}{T^2}\sum_{i=1}^{P}\left(\sum_{k=1}^{N^1}\sqrt{\mathbb{E}\left[\left\|F'(\boldsymbol{w}_{i,k-1}^1) - F''(\boldsymbol{w}^\star)(\boldsymbol{w}_{i,k-1}^1 - \boldsymbol{w}^\star)\right\|^2\right]}\right)^2$$

*Now first using the upper bound of A2, followed by Lemma S26, Lemma S45, Lemma S47 and Lemma S49 we can obtain the following,*

$$\mathbb{E}\left[\left\|\frac{1}{T}\sum_{i=1}^{P}\sum_{k=1}^{N^1}\delta_{i,k}^1\right\|^2\right] \leq \frac{PM^2}{T^2}\sum_{i=1}^{P}\left(\sum_{k=1}^{N^1}\mathbb{E}\left[\left\|\boldsymbol{w}_{i,k-1}^1 - \boldsymbol{w}^\star\right\|^4\right]^{1/2}\right)^2$$

$$\leq \frac{P^2 M^2}{T^2} \left( \sum_{k=1}^{N^1} \left( \prod_{j=1}^{k-1} (1 - \eta_j^1 \mu) \left\| \boldsymbol{w}^0 - \boldsymbol{w}^\star \right\|^2 + 20\sigma^2 \sum_{j=1}^{k-1} \prod_{l=j+1}^{k-1} (1 - \mu\eta_l^1)(\eta_j^1)^2 \right) \right)^2$$

$$\leq \frac{P^2 M^2}{T^2} \left( \sum_{k=1}^{N^1} \left( \exp\left( -\frac{\mu c_\eta (k-1)^{1-\alpha}}{1-\alpha} \right) \left\| \boldsymbol{w}^0 - \boldsymbol{w}^\star \right\|^2 + \frac{20\sigma^2 c_\eta}{\mu} \right) \right)^2$$

$$\leq \frac{P^2 M^2}{T^2} \left( \left( 1 + \left( \frac{(1-\alpha)^\alpha}{\mu c_\eta} \right)^{\frac{1}{1-\alpha}} \Gamma\left( \frac{1}{1-\alpha} \right) \right) \left\| \boldsymbol{w}^0 - \boldsymbol{w}^\star \right\|^2 + \frac{20\sigma^2 N^1 c_\eta}{\mu} \right)^2$$

$$\leq \frac{P^2 M^2}{T^2} \left( \beta_1 \left\| \boldsymbol{w}^0 - \boldsymbol{w}^\star \right\|^2 + \frac{20\sigma^2 N^1 c_\eta}{\mu} \right)^2 = D_{4,P,N^1}^2.$$

*For the fifth term, using the fact that for different machines noise is independent, zero in expectation (A3) we obtain,*

$$\mathbb{E}\left[ \left\| \frac{1}{T} \sum_{i=1}^{P} \sum_{k=1}^{N^1} \xi_{i,k}^1 \right\|^2 \right] = \frac{1}{T^2} \sum_{i=1}^{P} \sum_{k=1}^{N^1} \mathbb{E}\left[ \left\| \xi_{i,k}^1 \right\|^2 \right].$$

*Now using Lemma S33 we have,*

$$\mathbb{E}\left[ \left\| \frac{1}{T} \sum_{i=1}^{P} \sum_{k=1}^{N^1} \xi_{i,k}^1 \right\|^2 \right] \leq \frac{1}{T^2} \sum_{i=1}^{P} \sum_{k=1}^{N^1} \left( 2L^2 \mathbb{E}\left[ \left\| \boldsymbol{w}_{i,k-1}^1 - \boldsymbol{w}^\star \right\|^2 \right] + 2\sigma^2 \right)$$

$$\leq \frac{2\sigma^2}{T} + \frac{2L^2}{T^2} \sum_{i=1}^{P} \sum_{k=1}^{N^1} \mathbb{E}\left[ \left\| \boldsymbol{w}_{i,k-1}^1 - \boldsymbol{w}^\star \right\|^2 \right].$$

*Now using Lemma S11, followed by Lemma S45, Lemma S47 and Lemma S49 with definition of $\beta$ as before, and we have,*

$$\mathbb{E}\left[ \left\| \frac{1}{T} \sum_{i=1}^{P} \sum_{k=1}^{N^1} \xi_{i,k}^1 \right\|^2 \right] \leq \frac{2\sigma^2}{T} + \frac{2L^2 P}{T^2} \sum_{k=1}^{N^1} \left( \prod_{m=1}^{k-1} (1 - \mu\eta_m^1) \left\| \boldsymbol{w}^0 - \boldsymbol{w}^\star \right\|^2 + 2\sigma^2 \sum_{m=1}^{k-1} (\eta_m^1)^2 \prod_{l=m+1}^{k-1} (1 - \mu\eta_l^1) \right)$$

$$\leq \frac{2\sigma^2}{T} + \frac{2L^2 P}{T^2} \sum_{k=1}^{N^1} \left( \exp\left( -\frac{\mu c_\eta (k-1)^{1-\alpha}}{1-\alpha} \right) \left\| \boldsymbol{w}^0 - \boldsymbol{w}^\star \right\|^2 + \frac{2\sigma^2 c_\eta}{\mu} \right)$$

$$\leq \frac{2\sigma^2}{T} + \frac{2L^2 P}{T^2} \left( \left( 1 + \left( \frac{(1-\alpha)^\alpha}{\mu c_\eta} \right)^{\frac{1}{1-\alpha}} \Gamma\left( \frac{1}{1-\alpha} \right) \right) \left\| \boldsymbol{w}^0 - \boldsymbol{w}^\star \right\|^2 + \frac{2\sigma^2 N^1 c_\eta}{\mu} \right)$$

$$\leq \frac{2\sigma^2}{T} + \frac{2L^2 P}{T^2} \left( \left( 1 + \left( \frac{(1-\alpha)^\alpha}{\mu c_\eta} \right)^{\frac{1}{1-\alpha}} \Gamma\left( \frac{1}{1-\alpha} \right) \right) \left\| \boldsymbol{w}^0 - \boldsymbol{w}^\star \right\|^2 + \frac{2\sigma^2 N^1 c_\eta}{\mu} \right)$$

$$\leq \frac{2\sigma^2}{T} + \frac{2L^2 P}{T^2} \left( \beta \left\| \boldsymbol{w}^0 - \boldsymbol{w}^\star \right\|^2 + \frac{2\sigma^2 N c_\eta}{\mu} \right) = D_{5,P,N^1}^2 + D_{6,P,N^1}^2.$$

*Thus using Lemma S35 we have proved the lemma.*

We can get the following lemma combining the bias and variance terms separately,

**Lemma S62** *Under the Assumptions A1, A2, A3, A5, A6 for decreasing step size, for one shot averaging we have,*

$$\mathbb{E}\left[ \left\| \nabla^2 F(\boldsymbol{w}^\star)(\boldsymbol{w} - \boldsymbol{w}^\star) \right\|^2 \right] \leq 5 \left( \hat{D}_{1,P,N^1}^2 + \hat{D}_{2,P,N^1}^2 \right)$$

*Where for constants defined as above the terms are,*

$$\hat{D}_{1,P,N^1}^2 = \frac{\|\boldsymbol{w}^0 - \boldsymbol{w}^\star\|^2}{(N^1)^2 c_\eta^2} \left( 1 + 4(N^1)^{2\alpha} \exp\left( -\frac{\mu c_\eta (N^1)^{1-\alpha}}{2(1-\alpha)} \right) + \alpha^2 \beta_1 + 2M^2 c_\eta^2 \beta_1^2 \|\boldsymbol{w}^0 - \boldsymbol{w}^\star\|^2 + \frac{2L^2 \beta_1 c_\eta^2}{P} \right),$$

$$\hat{D}_{2,P,N^1}^2 = \frac{2\sigma^2}{T} \left( 1 + \frac{8\alpha P (N^1)^{2\alpha-1}}{2\alpha - 1} \exp\left( -\frac{\mu (N^1)^{1-\alpha}}{2(1-\alpha)} \left( 1 - \frac{1}{2^{1-\alpha}} \right) \right) + \frac{8P}{(N^1)^{1-\alpha} c_\eta \mu} + \frac{\alpha^2 P \beta_2}{2N^1 c_\eta^2} + \frac{\alpha^2 P \beta_3}{2(N^1)^{1-\alpha} c_\eta^2} \right.$$

$$\left. + \frac{400 M^2 P \sigma^2}{N^1} \left( \beta_2^2 + \beta_3^2 (N^1)^{2-2\alpha} \right) + \frac{L^2}{N^1} \left( \beta_2 + \beta_3 (N^1)^{1-\alpha} \right) \right).$$

# I   Brief overview of distributed optimization

The above three schemes (OSA, MBA, Local-SGD) are the most studied synchronous parallel schemes. However, communication latencies often make it difficult to use these algorithms for large-scale problems. Thus many alternative parallelization schemes which minimize communication or perform better have been studied. The major problem with some of these variants is that they are often difficult to tune, are not as stable and don't scale well to non-convex optimization problems. Result-wise, most of the machine learning packages use centralized mini-batch synchronous SGD.

**Asynchronous SGD:** These techniques are characterized by avoiding a centralized synchronization, using delayed updates, maintaining parameter server estimates and being fault tolerant. Some of the notable references in a chronological order are [46, 52, 54–70].

**Federated optimization:** This setting is characterized by a huge number of mobile user devices, which run their local model in a decentralized manner with often unbalanced data, but aim to train jointly. Many research questions still remain open but the direction is very relevant for distributed AI. Some references are [71–73].

**Compressed Communication:** A common strategy to combat the communication overhead is to introduce lossless or lossy compression of exchanged information, often the gradients. Some of the work in this direction can be found in [74–81].

**Non-SGD methods:** Many other optimization algorithms (coordinate descent, quasi newton, etc.) have also been studied in the parallel setting, owing to their better distributivity or convergence for some applications compared to the SGD algorithm. Some of them are [82] (ADMM), [83] (DANE), [84] (DiSCO), [85] (AIDE), [86–88] (COCOA) and some of the references therein. Recently [89] gave provably optimal algorithms for the strongly convex and smooth functions for both synchronous and asynchronous cases. More broadly speaking, variance reduction methods are often the methods of choice in better understood, convex optimization problems [add reference]. Yet, their usage in the deep learning community has been relatively scarce, and often they are more difficult to parallelize [add reference]. Some of the works for instance are [61, 90–92]. Among second order methods, quasi newton methods like distributed L-BFGS [93, 94] are also widely popular among the machine learning community.

**Communication Lower Bounds:** On a broader level our work is related to communication lower bounds which arise from information and learning-theoretic considerations. Unfortunately, these bounds are difficult to match for convex optimization as they are provided in [95]. Similar bounds have also been provided for the generally easier statistical estimation setting in [96–98].

**Feature Distribution:** As clearly evident training data is not the only element of our optimization scheme which can be parallelized. Often in many problems in natural language processing and linear estimation, the features number in hundreds of thousands, and it might be of some interest to distribute the features alongside or beside training data. Some relevant references are [87, 99–102].

There has also been work in parallelizing stochastic optimization algorithms for specific problems (like PCA) in the past, for e.g., [31, 32, 103–107].

| Reference | Setting | Limitations |
|---|---|---|
| Zhang et. al. [33] | OSA | Small learning rates $\frac{c}{\mu t}$; $\mu$ often unknown; Non-asymptotic bound on single worker convergence rate is used ([34]); |
| Jain et. al. [20] | OSA, MBA | Results for least square regression (LSR) in finite horizon setting only; |
| Godichon et. al. [108] | OSA | Uses uniform gradient bound **A**4 and thus not usable for LSR; Non-asymptotic result ([34]) is used; |
| Stich [40] | Local SGD | Small learning rates $\frac{c}{\mu t}$; $\mu$ often unknown; Uses uniform gradient bound **A**4 and thus not usable for LSR; Doesn't capture the need for an adaptive communication frequency [21]; Doesn't extend to one-shot averaging, implying it is not tight enough; |

Table S3: Limitations of the previously existing results.

We also provide a brief overview of some other techniques in distributed optimization in Appendix I.