[Reviews · NeurIPS 2019]

Reviewer 1



This paper seeks to address the existing ambiguity in distributed SGD and knowing how and when to aggregate the solution. They start by analyzing two extreme aggregate policies (every time and at the end) and show that the more frequent communication is by far better. Their closest work is [21] that assumes uniformly bounded gradients and [40] that also assumes bounded gradients and small step-sizes. They have clear notation and assumptions that make the paper concise and easier to follow. Their main result is in showing they can decompose the results into three terms that they then show bounds for. They bounds help show where communication comes into play to distinguish OSA and MBA. They make the interesting point that though the asymptotic equivalence exists, the pre-asympotic analysis is what distinguish the results in practice. This is interesting and practical and hopefully more papers consider pre-asymptotics.

Reviewer 2



The paper is densely packed with valuable novel analysis of a very popular algorithm: distributed SGD. It makes a substantial contribution to alleviating the communication cost of this algorithm, which is a major bottleneck in its practical application. It is a challenging read, but this is mostly due to the inherent complexity of the subject matter of analysing a distributed stochastic optimisation process. However, a little improvement seems possible by adding a paragraph in the beginning on the underlying optimisation problem and the kind of bounds one is interested in (see also below). Beyond that the paper is breaking down the complexity in an exemplary manner, giving interpretations of all formal results and a systematic listing of all the various assumptions that have been traditionally used in the formal analysis of (distributed) SGD. Some suggestions: - The paper mentions that it is focused on analysing the Malahanobis distance of the parameter vector from the optimum as the “natural quantity in this setting”. However, it also mentions that one “eventually aims to minimise the excess risk”, i.e., the expected difference in function value from the optimum. This is plausible as we are ultimately interested in optimising F. It is mentioned that the two quantities are related but it is also hinted that using straightforward relations does not necessarily lead to a tight analysis. The paper could benefit from a more consistent and explicit discussion of this issue; especially, how translation of parameter distance relates to F differences under the various assumptions listed in Section 2.3. Right now the respective paragraph in Section 3.1 feels more like an afterthought that is not well integrated into the main narrative of the paper. - The paper hints at a couple of places that it would be interesting to investigate adaptive communication schedules (as opposed to statically predefined just based on the number of workers and the properties of F). Such a generalisation has been proposed for the online case in Kamp et al., Communication-Efficient Distributed Online Prediction by Dynamic Model Synchronization, ECMLPKDD, 2013, which uses a form of local SGD combined with actively monitoring an upper bound to the variance of the local parameters. While the focus there is not on the convergence of the Polyak Rupert iterate but on the in-place online loss, the idea seems transferable, and the goals of the papers appear so similar that it might be worth to mention. - Assumption A3 seems more technically than what I would usually expect in an ML conference and, more importantly, could rather be considered a part of the problem definition. Perhaps some valuable space could be reclaimed in the main paper by mentioning in an initial paragraph about the problem that the g’s for a specific weight vector are i.i.d estimates of the gradient (and refer to the supplementary appendix for a full generalisation)? - When mentioning the convexity parameter mu for the first time in Section 2.2 its definition is only implicit (the symbol is defined only later in Assumption A1) - Example 5 should be Theorem or Corollary 5

Reviewer 3



Update: I've revised the overall score based on the response from the authors. This is another step to better understandings to distributed SGD approach for deep learning model training. However, the key limitation is on the strong assumptions over the model, which makes the results not meaningful to any practice. Q1 in Section 2.3 strongly claims that the mapping from weights w to the loss function is a quadratic function. This will never happen in real world. Moreover, batch normalization is commonly applied in real world deep learning tasks. Obviously, the convergence results do not consider it, although it is out of scope of a theoretical study in this submission.

[Author Response · NeurIPS 2019]

We thank the reviewers for their time and feedback. We address the main concerns raised by different reviewers below.

**Reviewer 1**

We will provide more illustrations on simple data-sets. As our first goal was to provide a tight analysis of local-sgd from a theoretical perspective, we did not highlight the experiments, some are provided in the Supplementary Material. We will add an illustration of the optimal behavior described on lines 343-357, and a direct illustration of Corollary 4.

While not directly illustrative of our theory, experiments in related papers like [1], [2] (which focus on practical aspects) are consistent with our findings, and underline similar observations.

**Reviewer 2**

Thank you for your detailed feedback and precise comments!

We will further clarify the discussion about functional convergence and also present it earlier. We will also add the suggested references and minor changes to the final draft.

Thank you for the suggestion regarding Assumption A3, which is indeed a bit technical: we will simplify its description as you suggested!

**Reviewer 3**

We would like to point out that the reviewer's major criticism is already tackled in our paper. We offer results both with and without the quadratic assumption (Q1): while Theorem 3 uses the Q1, Proposition 1,2 and Theorem 6.2 **do not require** this assumption.

Especially, Theorem 6.2 which is one of the main contributions of the paper, does not use Q1, but only the more common smoothness assumption. The propositions leading up to this theorem (placed in the appendix due to lack of space), as well as the online setting's results (Proposition 7) also do not make this assumption.

We chose to first provide and describe results under Q1 in the main paper only for the sake of brevity, and because they convey a very similar message more elegantly, and help to understand our approach.

We hope this clarifies the reviewer's main concern and that the score will be updated to reflect this.

**References**

[1] T. Lin, S. U. Stich, and M. Jaggi. Don't Use Large Mini-Batches, Use Local SGD. 2018.

[2] J. Zhang, C. De Sa, I. Mitliagkas, and C. Ré. Parallel SGD: When does averaging help? 2016.


[Meta-Review · NeurIPS 2019]

This paper presents an analysis of distributed SGD with large stepsize, which provides useful insights.